# The transition state and regulation of γ-TuRC-mediated microtubule nucleation revealed by single molecule microscopy

Akanksha Thawani[1], Michael J Rale[2], Nicolas Coudray[3], Gira Bhabha[3], Howard A Stone[4], Joshua W Shaevitz[5,6], Sabine Petry[2]*

[1]Department of Chemical and Biological Engineering, Princeton University, Princeton, United States; [2]Department of Molecular Biology, Princeton University, Princeton, United States; [3]Department of Cell Biology, New York University School of Medicine, New York, United States; [4]Department of Mechanical and Aerospace Engineering, Princeton University, Princeton, United States; [5]Lewis-Sigler Institute for Integrative Genomics, Princeton, United States; [6]Department of Physics, Princeton University, Princeton, United States

**Abstract** Determining how microtubules (MTs) are nucleated is essential for understanding how the cytoskeleton assembles. While the MT nucleator, γ-tubulin ring complex (γ-TuRC) has been identified, precisely how γ-TuRC nucleates a MT remains poorly understood. Here, we developed a single molecule assay to directly visualize nucleation of a MT from purified *Xenopus laevis* γ-TuRC. We reveal a high γ-/αβ-tubulin affinity, which facilitates assembly of a MT from γ-TuRC. Whereas spontaneous nucleation requires assembly of 8 αβ-tubulins, nucleation from γ-TuRC occurs efficiently with a cooperativity of 4 αβ-tubulin dimers. This is distinct from pre-assembled MT seeds, where a single dimer is sufficient to initiate growth. A computational model predicts our kinetic measurements and reveals the rate-limiting transition where laterally associated αβ-tubulins drive γ-TuRC into a closed conformation. NME7, TPX2, and the putative activation domain of CDK5RAP2 do not enhance γ-TuRC-mediated nucleation, while XMAP215 drastically increases the nucleation efficiency by strengthening the longitudinal γ-/αβ-tubulin interaction.

**\*For correspondence:**
spetry@Princeton.EDU

**Competing interests:** The authors declare that no competing interests exist.

## Introduction

Microtubules (MTs) enable cell division, motility, intracellular organization and transport. Half a century ago, MTs were found to be composed of αβ-tubulin dimers, yet how MTs are nucleated in the cell to assemble the cellular structures remains poorly understood (*Petry, 2016*; *Wu and Akhmanova, 2017*). The universal nucleator, γ-tubulin efficiently nucleates MTs in vivo (*Oakley and Oakley, 1989*; *Hannak et al., 2002*; *Groen et al., 2009*) by forming a 2.2 megadalton, ring-shaped complex with γ-tubulin complex proteins (GCPs), known as the γ-Tubulin Ring Complex (γ-TuRC) (*Moritz et al., 1995*; *Moritz et al., 1998*; *Zheng et al., 1995*; *Kollman et al., 2010*; *Kollman et al., 2015*; *Oegema et al., 1999*). Structural studies (*Kollman et al., 2010*; *Moritz et al., 2000*; *Liu et al., 2020*; *Wieczorek et al., 2020*; *Consolati et al., 2020*) have revealed that γ-TuRC positions a lateral array of 13 γ-tubulin molecules that are thought to template MT assembly by binding αβ-tubulin dimers and promoting their lateral interaction to result in nucleation of a MT (*Kollman et al., 2010*; *Moritz et al., 2000*; *Keating and Borisy, 2000*; *Wiese and Zheng, 2000*; *Kollman et al., 2011*). Despite this model being widely accepted, MT nucleation from γ-TuRC molecules has not been directly visualized in real time and the dynamics of nucleation of a MT from αβ-tubulin dimers remains to be characterized. In particular, determining the critical nucleus, that is the

rate-limiting transition state, for γ-TuRC nucleation is of tremendous interest, as it has important implications for how MT nucleation is spatiotemporally regulated in the cell (*Figure 1A*).

In the absence of γ-TuRC, MTs can also nucleate spontaneously from high concentrations of αβ-tubulin in vitro. In this process, which displays a nucleation barrier, the assembly of many αβ-tubulin dimers is thought to occur to form lateral and longitudinal contacts (*Voter and Erickson, 1984*; *Flyvbjerg et al., 1996*; *Portran et al., 2017*; *Roostalu and Surrey, 2017*). It has long been speculated whether γ-TuRC-mediated nucleation occurs similarly, or follows a distinct reaction pathway (*Kollman et al., 2011*; *Roostalu and Surrey, 2017*; *Rice et al., 2019*; *Wiese and Zheng, 2006*; *Wieczorek et al., 2015*). Moreover, the structure of native γ-TuRC shows an *open* conformation where adjacent γ-tubulin do not form a lateral interaction (*Liu et al., 2020*; *Wieczorek et al., 2020*; *Consolati et al., 2020*), raising further questions on how the conformational mismatch impacts γ-

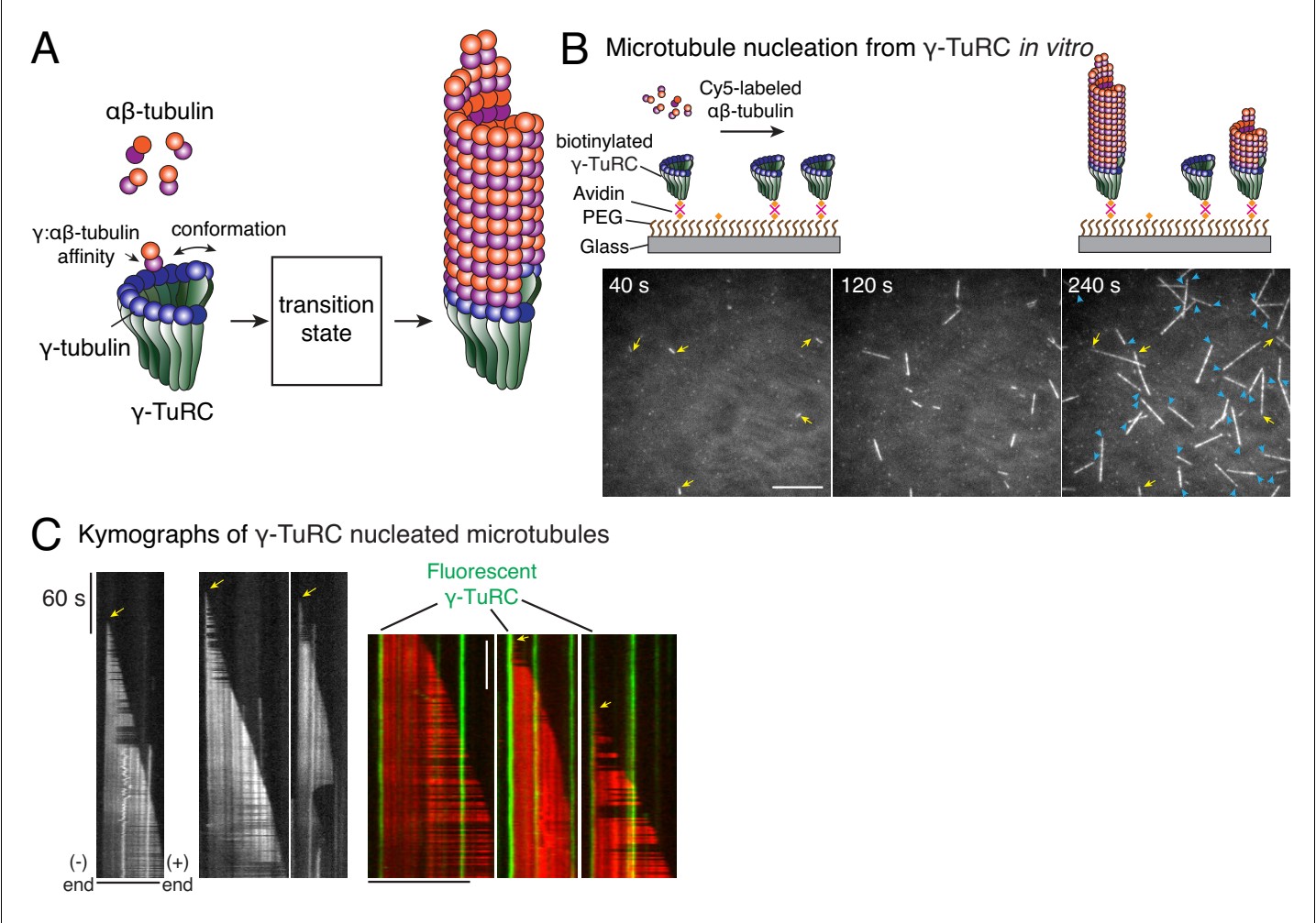

**Figure 1.** Single molecule microscopy of microtubule nucleation from γ-TuRC. (**A**) Schematic for microtubule nucleation from γ-TuRC. Biochemical features of γ-TuRC including the γ-/αβ-tubulin interaction affinity and conformation of γ-TuRC determine to MT nucleation activity and transition state. (**B**) Purified, biotinylated γ-TuRC molecules were attached, incubated with 14 μM αβ-tubulin and time-lapse of MT nucleation after is shown. MTs already nucleated in the first frame are marked with yellow arrow, while new MT nucleation events between the first and last frame with blue arrows. (**C**) Three representative kymographs of (left) unlabeled γ-TuRC nucleating MTs colored in grayscale, or (right) fluorescently-labeled γ-TuRC, pseudo-colored in green, nucleating MTs, pseudo-colored in red. Arrows point to nucleation sites. The experiments with unlabeled γ-TuRC were repeated more than 10 times with independent γ-TuRC preparations, while those with fluorescent γ-TuRC repeated were repeated six times with three independent γ-TuRC preparations. See *Figure 1—figure supplement 1* and *Videos 1–2*.

The online version of this article includes the following figure supplement(s) for figure 1:

**Figure supplement 1.** Controls for γ-TuRC-mediated microtubule nucleation.

TuRC's nucleation activity (*Figure 1A*). It has been widely proposed that γ-TuRC may transition to a *closed* conformation during MT assembly to match the geometry of αβ-tubulin dimers arranged laterally in the MT lattice (*Kollman et al., 2015*; *Liu et al., 2020*). This transition could further provide a mode of regulation through several putative MT-associated proteins (MAPs) that have been proposed to promote a closed conformation of γ-TuRC's (*Kollman et al., 2015*; *Liu et al., 2020*; *Kollman et al., 2011*) and regulate γ-TuRC's nucleation activity (*Kollman et al., 2010*; *Kollman et al., 2015*; *Liu et al., 2020*; *Choi et al., 2010*; *Liu et al., 2014*; *Lynch et al., 2014*). Finally, the interaction affinity between γ-tubulin and αβ-tubulin and its role on MT nucleation remain unknown (*Kollman et al., 2011*; *Rice et al., 2019*; *Figure 1A*).

Investigating the molecular biophysics of MT nucleation by γ-TuRC at the single-molecule level and with computational modeling have the potential to address these questions. By identifying transition states and reaction intermediates during the γ-TuRC-mediated nucleation reaction, important insights into the dynamics of MT nucleation can be revealed. Yet, technical challenges in both purifying γ-TuRC at high yield, as well as the inability to visualize MT nucleation events from individual γ-TuRC molecules in real time and at high resolution, have posed limitations. In this work, we overcome these longstanding challenges to reconstitute MT nucleation from γ-TuRC and visualize the reaction live at the resolution of single molecules. We use computational models to gain further mechanistic insights into MT nucleation and to identify the molecular composition and arrangement of the rate-limiting transition state in γ-TuRC. Finally, we examine the roles of various MAPs, particularly the co-nucleation factor XMAP215, in γ-TuRC-mediated MT nucleation and comprehensively examine how specific biomolecular features govern how MT nucleation from γ-TuRC occurs.

## Results

### Visualizing microtubule nucleation from γ-TuRC with single molecule microscopy

To study how γ-TuRC nucleates a MT, we purified endogenous γ-TuRC from *Xenopus* egg extracts and biotinylated the complexes to immobilize them on functionalized glass (*Figure 1—figure supplement 1A–B*). Upon perfusing fluorescent αβ-tubulin, we visualized MT nucleation live with total internal reflection fluorescence microscopy (TIRFM) (*Figure 1B*). Strikingly, MT nucleation events occurred specifically from γ-TuRC molecules that were either unlabeled (*Figure 1B* and *Video 1*) or fluorescently labeled during the purification (*Figure 1—figure supplement 1C* and *Video 2*). Kymographs revealed that single, attached γ-TuRC molecules assembled αβ-tubulin into a MT de novo starting from zero length within the diffraction limit of light microscopy (*Figure 1C*), ruling out an alternative model where MTs first spontaneously nucleate and then become stabilized via γ-TuRC. By observing fiduciary marks on the MT lattice (*Figure 1C*) and generating polarity-marked MTs from attached γ-TuRC (*Figure 1—figure supplement 1D*), we show that γ-TuRC caps the MT minus-end while only the plus-end polymerizes, as supported by previous works (*Keating and Borisy, 2000*; *Wiese and Zheng, 2000*). Notably, the detachment of γ-TuRC molecules and re-growth of the MT minus-ends were not observed, and γ-TuRC persists on the MT minus-end for the duration of our experiments. Altogether, our results demonstrate that γ-TuRC directly nucleates a MT.

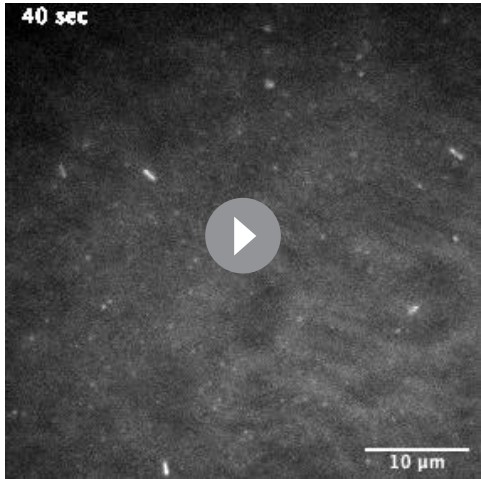

**Video 1.** Microtubule nucleation from γ-TuRC complexes. γ-TuRC was attached to functionalized coverslips and MT nucleation was observed upon introducing fluorescent αβ-tubulin (gray). MTs nucleated from individual γ-TuRC molecules from zero length at 14 μM αβ-tubulin and the plus-end of nucleated MTs polymerized, but not its minus-end. Elapsed time is shown in seconds, where time-point zero represents the start of reaction. Scale bar, 10 μm.
https://elifesciences.org/articles/54253#video1

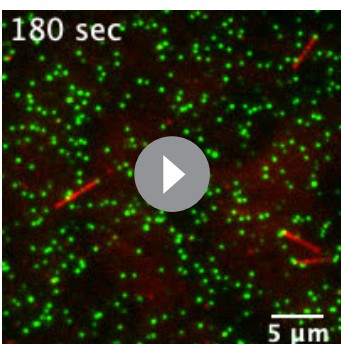

**Video 2.** Microtubule nucleation from fluorescent, single γ-TuRC molecules. Dual Alexa-568 and biotin-labeled γ-TuRC (green) was attached to functionalized coverslips and MT nucleation was observed upon introducing fluorescent αβ-tubulin (red). MTs nucleated from single γ-TuRC molecules at 10.5 µM αβ-tubulin. Elapsed time is shown in seconds, where time-point zero represents the start of reaction. Scale bar, 10 µm.
https://elifesciences.org/articles/54253#video2

## Molecular composition of the transition state during γ-TuRC-mediated nucleation

To determine how γ-TuRC nucleates a MT, we measured the kinetics of MT nucleation for a constant density of γ-TuRC molecules and increasing αβ-tubulin concentrations (*Figure 2A* and *Video 3*). γ-TuRCs nucleated MTs starting from 7 µM tubulin (*Figure 2A–B*), which is higher than the minimum tubulin concentration ($C^*$) needed for growth from a pre-formed MT plus-end ($C^*$=1.4 µM, *Figure 2B*). Furthermore, the number of MTs nucleated from γ-TuRC increased non-linearly with αβ-tubulin concentration as opposed to the linear increase in MT's growth speed with tubulin concentration (*Figure 2B*). By measuring the number of MTs nucleated over time with varying αβ-tubulin concentration (*Figure 2C*), we calculated the rate of MT nucleation. The power-law dependence on αβ-tubulin concentration (*Figure 2D*) yields the number of tubulin dimers, 3.9 ± 0.5, that compose the rate-limiting, transition state during MT assembly from γ-TuRC (*Figure 2D*). Thus, the cooperative assembly of nearly 4 αβ-tubulin subunits on γ-TuRC represents the most critical, rate-limiting step in MT nucleation.

## γ-TuRC-mediated nucleation is more efficient than spontaneous nucleation

Based on the traditional assay where MTs are nucleated, fixed and visualized, a large variability in γ-TuRC's MT nucleation activity has been observed. With this setup, γ-TuRC has often been reported to be a poor nucleator with a similar activity as spontaneous MT nucleation (*Moritz et al., 1995*; *Zheng et al., 1995*; *Kollman et al., 2010*; *Kollman et al., 2015*; *Oegema et al., 1999*; *Liu et al., 2020*; *Wieczorek et al., 2020*; *Choi et al., 2010*; *Thawani et al., 2018*). With our live TIRFM assay, we aimed to quantitatively compare the efficiency of γ-TuRC-mediated MT nucleation with spontaneous MT nucleation (*Figure 3A*). In contrast to γ-TuRC-mediated nucleation, a higher concentration of 14 µM tubulin was required for any spontaneous assembly of MTs, after which both the plus- and minus-ends polymerize (*Figure 3B*, *Figure 3—figure supplement 1A* and *Video 4*). The number of MTs assembled as a function of the αβ-tubulin concentration displayed a power-law dependence with an even larger exponent of 8.1 ± 0.9 (*Figure 3C*), indicating a highly cooperative process that requires 8 αβ-tubulin dimers in a rate-limiting intermediate, in agreement with previous reports (*Voter and Erickson, 1984*; *Flyvbjerg et al., 1996*). Further, direct comparison and measurement of spontaneous MT assembly with γ-TuRC-mediated nucleation (*Figure 3—figure supplement 1B–C*) clearly demonstrates that γ-TuRC nucleates MTs significantly more efficiently. Notably, specific attachment of γ-TuRC to coverslips is also required to observe the nucleation activity (*Figure 3—figure supplement 1C*). In sum, γ-TuRC-mediated nucleation occurs efficiently and its critical nucleus requires less than half the number of αβ-tubulin dimers compared to spontaneous assembly.

## Contribution of end architecture of γ-TuRC to microtubule nucleation

The MT plus-end architecture, which ranges from blunt to tapered, is critical for MT polymerization dynamics (*Gardner et al., 2014*; *Mickolajczyk et al., 2019*; *Brouhard and Rice, 2018*), and was recently proposed to be critical for MT nucleation (*Wieczorek et al., 2015*). To investigate how the blunt-end geometry of γ-TuRC contributes to its nucleation kinetics and transition state, we generated Alexa-568 labeled, stable MT seeds with blunt ends as described previously (*Wieczorek et al., 2015*) and compared MT assembly from seeds upon addition of Cy5-labelled αβ-tubulin dimers (*Figure 3C*) side-by-side with γ-TuRC-mediated nucleation. At a minimum concentration of 2.45 µM,

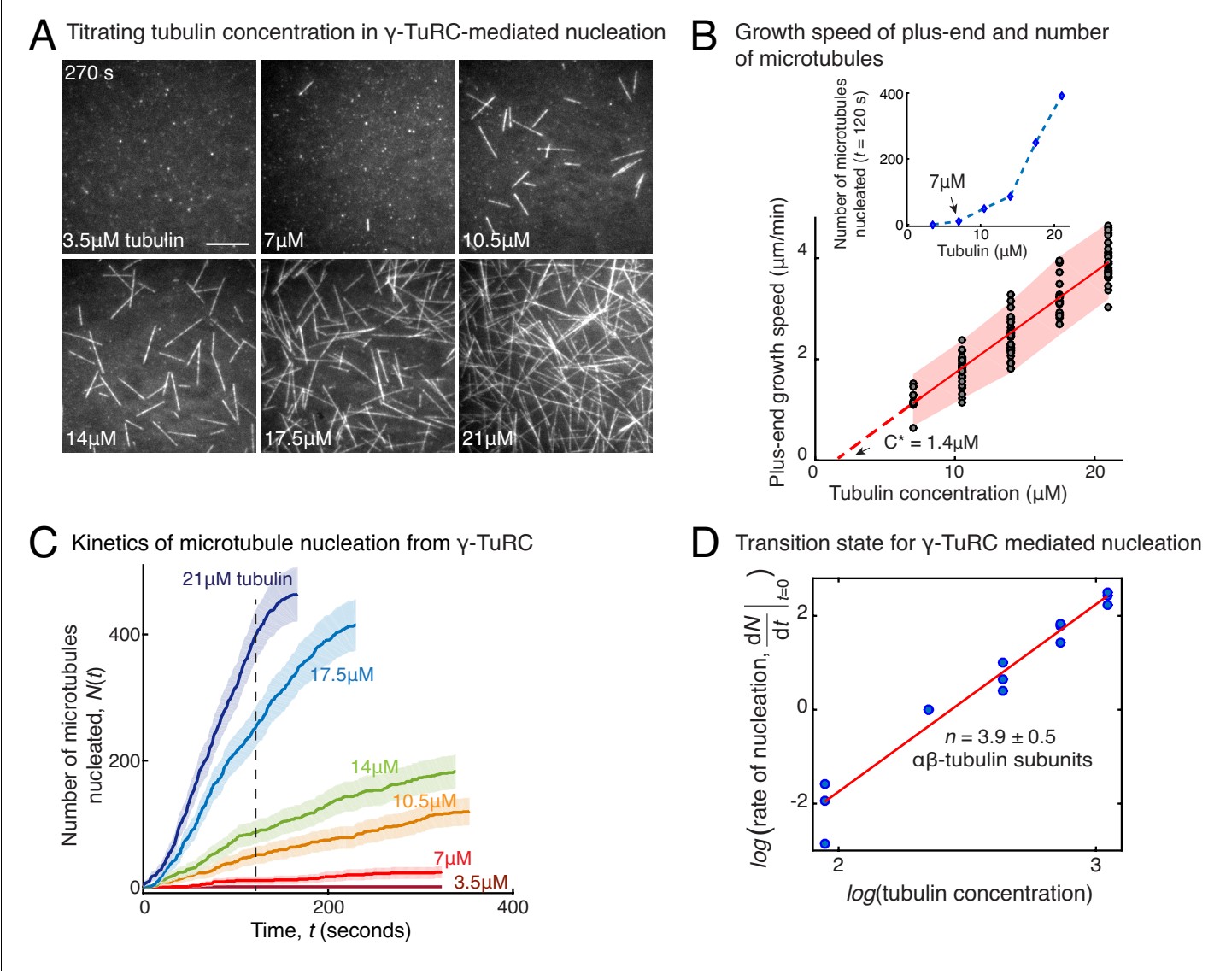

**Figure 2.** Molecular composition of transition state in γ-TuRC-mediated nucleation. (**A**) Titrating tubulin concentration with constant the density of γ-TuRC. MT nucleation from γ-TuRC begins at 7µM tubulin. (**B**) MT plus-end growth speed increases linearly with tubulin concentration. Individual data points are plotted, and linear fit (red line) with shaded mean±2std (95% confidence interval) is displayed. Critical concentration for polymerization as $C^*$ = 1.4 µM. Inset: Number of MTs nucleated by γ-TuRCs within 120 s varies non-linearly with tubulin concentration. (**C**) Number of MTs nucleated ($N(t)$) over time (**t**) is plotted for varying tubulin concentration to obtain rate of nucleation as the slope of the initial part of the curves. Shaded regions represent 95% confidence interval ($n \pm 2\sqrt{n}$) in the number of nucleated MTs (**n**) assuming a Poisson distribution as described in Materials and methods. (**D**) Number of tubulin dimers (**n**) in the critical nucleus on γ-TuRC was obtained as 3.9±0.5 from the equation $\frac{dN}{dt}\big|_{t\to 0} = kC_{tub}^n$ displayed on a log-log axis as detailed in Materials and methods. The rate of nucleation at 10.5 µM was set to 1 to normalize differences in γ-TuRC concentration from individual experiments. The experiments and analyses in (**A–D**) were repeated identically three times with independent γ-TuRC preparations. MT nucleation data, prior to normalization, from one representative dataset is displayed in (**B–C**). Analyses from all repeats was pooled and normalized as described above, and data points from 15 nucleation-time curves are plotted in (**D**). See **Video 3**.

The online version of this article includes the following source data for figure 2:

**Source data 1.** Source data for **Figure 2B–D**.

approaching the critical concentration needed for polymerization of a MT plus-end, a large proportion of pre-formed MT seeds assemble MTs (**Figure 3C–D**, **Figure 3—figure supplement 1D** and **Video 5**). At 7 µM tubulin, the rate of assembly of MTs from the blunt seeds increased to reach the maximum rate that could be temporally resolved, that is all of the MT seeds immediately assembled

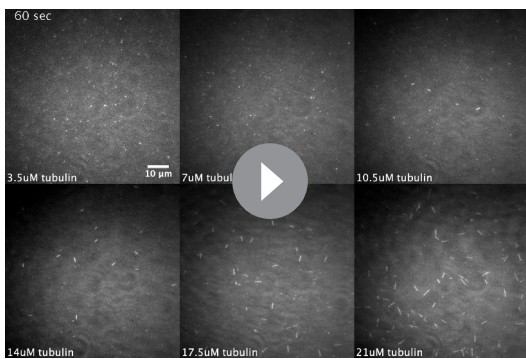

**Video 3.** γ-TuRC molecules nucleate microtubules efficiently. Constant density of γ-TuRC was attached while concentration of fluorescent αβ-tubulin was titrated (3.5–21 µM) and MT nucleation was observed. γ-TuRC molecules nucleated MTs starting from 7 µM tubulin and MT nucleation increased non-linearly with increasing tubulin concentration. Elapsed time is shown in seconds, where time-point zero represents the start of reaction. Scale bar, 10 µm.

https://elifesciences.org/articles/54253#video3

a MT (*Figure 4D*). This is in contrast to the kinetics of γ-TuRC-mediated nucleation at 7 µM tubulin concentration, where minimal nucleation activity was observed (*Figure 2C–D*). The measured reaction kinetics as a function of the αβ-tubulin concentration (*Figure 3D*) was used to obtain the power-law of the nucleation rate, 1 ± 0.3 (*Figure 3E*). This suggests that in our assay condition, blunt MT ends assemble tubulin dimers into the MT lattice non-cooperatively. In other words, the addition of a single αβ-tubulin dimer suffices to overcome the rate-limiting barrier, which also occurs during the polymerization phase of MT dynamics. Notably, when this experiment was replicated with the coverslip preparation and assay conditions reported previously (*Wieczorek et al., 2015*), a high concentration of tubulin was necessary for seeds to assemble MTs in agreement with the previous work (*Wieczorek et al., 2015*). However, our assay conditions, that were used to compare seed-templated MT assembly with γ-TuRC-mediated nucleation side-by-side, result in a low, minimal tubulin concentration that is needed for seed-mediated MT assembly. To conclude, while the γ-TuRC positions a blunt plus-end of γ-tubulins, the contribution of this specific end architecture in defining the kinetics of nucleation from γ-TuRC and its transition state is minimal.

In summary, because γ-TuRC positions an array of γ-tubulins at its nucleation interface that are thought to stabilize intrinsically weak, lateral αβ/αβ-tubulin interaction (*Kollman et al., 2010*; *Kollman et al., 2015*; *Liu et al., 2020*; *Wieczorek et al., 2020*; *Kollman et al., 2011*; *Roostalu and Surrey, 2017*; *Rice et al., 2019*), MT nucleation by γ-TuRC has been proposed to function similar to polymerization of a MT end. Here, we show several lines of evidence that γ-TuRC-mediated nucleation has distinct characteristics from MT polymerization and assembly from blunt MT seeds. While growth speed of MTs nucleated from γ-TuRC or templated from MT seeds is similar (*Figure 3—figure supplement 1D*), γ-TuRC molecules do not nucleate MTs at low tubulin concentration where MT polymerization can occur. Further increasing tubulin concentration results in a non-linear increase in the number of γ-TuRCs molecules that nucleate MT, as opposed to a linear increase in rate of assembly from seeds. At the highest tubulin concentrations tested, approximately 10–15% of γ-TuRCs nucleate MTs in the TIRF assays (see Materials and Methods). While these results were obtained with endogenous γ-TuRCs purified from cytosol, it remains possible that specific factors at MTOCs can modulate γ-TuRC's conformation and kinetics. In summary, the rate-limiting transition state on γ-TuRC is composed of four αβ-tubulin dimers in contrast with MT polymerization where one tubulin dimer suffices to overcome the slowest step.

## γ-tubulin has a high affinity for αβ-tubulin

Consequently, specific biochemical features of γ-TuRC must govern its nucleation activity and the composition of the transition state during nucleation. To address this, we first measured the interaction affinity between γ-tubulin and αβ-tubulin, which could provide insight into γ-TuRC's nucleation interface and its role in MT nucleation. To begin, we performed size-exclusion chromatography where γ-tubulin alone elutes as a broad peak in fractions I-N (*Figure 4A (i)*, pseudo-colored profile in green) at low concentration. Interestingly, in the presence of either 10 µM (low) or 35 µM (high) concentrations of αβ-tubulin, the γ-tubulin binds to αβ-tubulin (pseudo-colored profile in cyan) and elutes earlier, specifically in fraction H (*Figure 4A (ii-iii)*, yellow arrow). Further, the overall elution profile of γ-tubulin is altered to follow αβ-tubulin, showing that γ-tubulin binds to αβ-tubulin at both the low and high concentrations we tested. To compare this with αβ-/αβ-tubulin's longitudinal interaction, we performed chromatography of αβ-tubulin alone (*Figure 4—figure supplement 1A*). At a

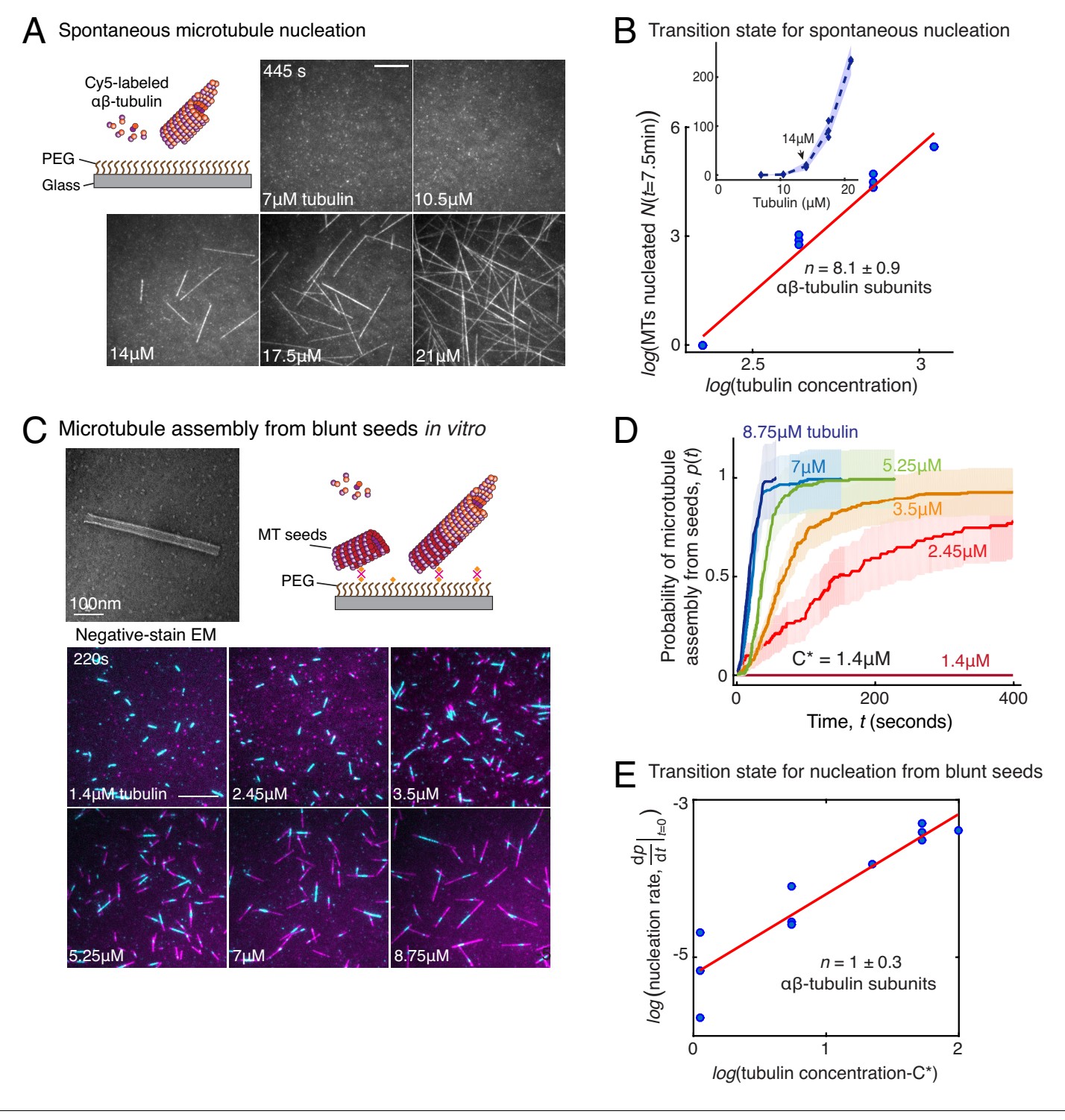

**Figure 3.** Comparison of γ-TuRC-mediated, spontaneous and seed-templated nucleation. (**A**) Spontaneous MT nucleation (schematized) was measured with increasing tubulin concentration and high concentrations. 14 μM tubulin is required. (**B**) Number of MTs ($N(t=\tau)$) nucleated spontaneously were plotted against tubulin concentration. Power-law curve was fit as $N(t=\tau) = kC^n$ on a log-log axis, and linear scale in the inset. Tubulin cooperativity (exponent) of $n = 8.1\pm0.9$ was obtained as detailed in Methods. Experiments and analyses in (**A–B**) were repeated thrice independently, all data were pooled and data points from 11 nucleation curves are plotted in (**B**). In the inset, data is represented in linear plot, where shaded regions represent 95% confidence interval ($n \pm 2\sqrt{n}$) in the number of nucleated MTs ($n$) assuming a Poisson distribution as described in Materials and methods. Scale bars, 10μm. (**C**) Schematic and an example micrograph of blunt, stabilized MT seeds is shown and MT assembly from them was observed (bottom) with

*Figure 3 continued on next page*

*Figure 3 continued*

varying tubulin concentration. (D) Cumulative probability of MT assembly from seeds ($p(t)$) over time (**t**) is plotted and rate of nucleation was obtained as the slope from initial part of the curves. Shaded regions represent 95% confidence interval ($n \pm 2\sqrt{n}$) in the number of MTs assembled ($n$) from seeds as described in Materials and methods. (**E**) As described in Methods, the measurements fit well to equation $\frac{dp}{dt}\big|_{t\to 0} = k(C - C^*)^n$ displayed on a log-log plot. $n = 1\pm 0.3$ was obtained showing nearly non-cooperative assembly of tubulin dimers. The experiments and analyses in (**C–E**) were repeated three times independently. MT nucleation data, prior to normalization, from one representative dataset is displayed in (**C–D**). Analyses from all experiments was pooled, and data points from a total of 11 nucleation-time curves are reported in (**E**). See *Figure 3—figure supplement 1* and *Videos 4* and *5*.

The online version of this article includes the following source data and figure supplement(s) for figure 3:

**Source data 1.** Source data for *Figure 3* panels B, D, E and *Figure 3—figure supplement 1C–D*.
**Figure supplement 1.** Controls for spontaneous and blunt seed-mediated microtubule nucleation.

---

lower concentration (10 μM), αβ-tubulin elutes only as a single subunit in fractions H-K (*Figure 4—figure supplement 1A (i)*). Only at high αβ-tubulin concentration (35 μM) did we detect a small population of αβ-tubulin bound to another αβ-tubulin (*Figure 4—figure supplement 1A (i)*, fractions B-C denoted with red arrows). This suggests that the heterogeneous γ-/αβ-tubulin affinity is higher than the αβ-/αβ-tubulin affinity.

To further investigate how γ-tubulin and αβ-tubulin interact, we turned to single molecule microscopy. We attached biotinylated αβ-tubulin dimers to a coverslip, added either fluorescently labeled αβ-tubulin (*Figure 4B (i)*) or γ-tubulin (*Figure 4B (ii)*) to the solution, and visualized the binding of single fluorescent molecules to αβ-tubulin molecules on the coverslip. While both fluorescent αβ-tubulin and γ-tubulin specifically bind to surface-attached αβ-tubulin, 15-fold more γ-tubulin molecules were bound than αβ-tubulin molecules (*Figure 4C*), further supporting a stronger γ-/αβ-tubulin interaction. Finally, these results were confirmed with a biolayer interferometry assay, where lower concentrations of γ-tubulin were detected to interact with probe-bound αβ-tubulin, while a much higher concentration of αβ-tubulin was necessary to measure an interaction between αβ-/αβ-tubulin dimers (*Figure 4—figure supplement 1B*). These results are congruent with in vivo γ-/αβ-tubulin affinity measurements made in yeast cells (*Erlemann et al., 2012*).

In performing the above experiments, we unexpectedly found that purified γ-tubulin on its own, at high concentrations and at 33°C, efficiently nucleated MTs from αβ-tubulin subunits (*Figure 4—figure supplement 2A*) and capped MT minus-ends while allowing plus-ends to polymerize (*Figure 4—figure supplement 2B*). Besides its ability to form higher order oligomers in a physiological buffer (*Thawani et al., 2018*), γ-tubulin at high concentrations also forms filaments in vitro of variable widths (*Figure 4—figure supplement 2C*; Moritz and Agard, unpublished results) as assayed by negative stain electron microscopy (EM). The formation of filaments in vitro is consistent with the previous in vivo observations where γ-tubulin was over-expressed and immunoprecipitated (*Lindström and Alvarado-Kristensson, 2018*; *Chumová et al., 2018*; *Pouchucq et al., 2018*). To understand the nature of these filaments, we generated 3D reconstructions, which revealed that γ-tubulins self-assemble into lateral arrays with a repeating unit of approximately 54 Å (*Figure 4—figure supplement 2D–E*). This closely matches the lateral tubulin repeats in the MT lattice (PDB:6DPU [*Zhang et al., 2018*; *Zhang and Nogales, 2018*]) and in γ-tubulin crystal contacts

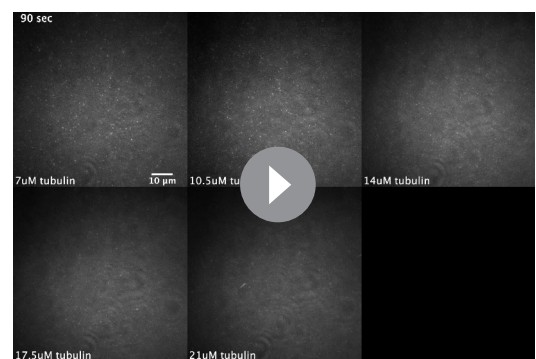

**Video 4.** Spontaneous microtubule nucleation occurs at high tubulin concentration. Concentration of fluorescent αβ-tubulin was titrated (7–21 μM) and spontaneous MT nucleation was assayed. MTs nucleated spontaneously starting from high concentration of 14 μM tubulin and MT nucleation increased non-linearly with tubulin concentration. Both plus- and minus-ends of the assembled MTs polymerize. Elapsed time is shown in seconds, where time-point zero represents the start of reaction. Scale bar, 10 μm.
https://elifesciences.org/articles/54253#video4

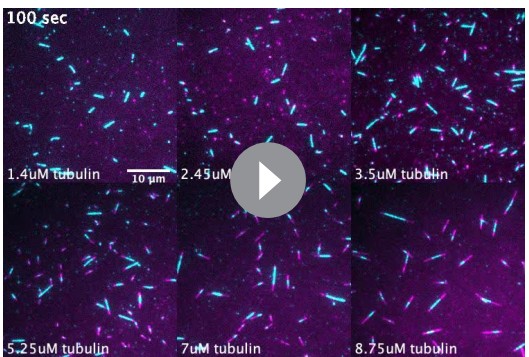

**Video 5.** Microtubule assembly from blunt plus-ends resembles polymerization. MTs with blunt ends (seeds, cyan) were generated and attached to functionalized coverslips. Varying concentration of fluorescent αβ-tubulin was added (1.4–8.7 μM, pseudo-colored as magenta) and MT assembly from seeds was assayed. MTs assembled at concentration above 1.4 μM tubulin, which is the minimum concentration needed for polymerization of MT plus-ends (C*). Elapsed time is shown in seconds, where time-point zero represents the start of reaction. Scale bar, 10 μm.
https://elifesciences.org/articles/54253#video5

(52 Å, PDB:1Z5W [*Aldaz et al., 2005a*; *Aldaz et al., 2005b*]), but not the longitudinal αβ-tubulin repeat (40 Å). This suggests that laterally associated γ-tubulin are sufficient to efficiently nucleate MTs.

In sum, at the nucleation interface of γ-TuRC, γ-tubulin has a higher longitudinal affinity for αβ-tubulin compared to αβ-tubulin's affinity for itself, which promotes MT nucleation from γ-TuRC.

## Monte Carlo simulations recapitulate the dynamics microtubule nucleation from γ-TuRC

To further probe the dynamics of MT nucleation, we developed Monte Carlo simulations to model MT nucleation from γ-TuRC. Our model was based on one previously developed for the plus-end dynamics of a MT (*Mickolajczyk et al., 2019*; *VanBuren et al., 2002*; *Ayaz et al., 2014*). A 13-protofilament geometry for the MT lattice and γ-TuRC were used with a pitch of 3 tubulins (*Figure 5A*). αβ-tubulin dimers arrive with a constant on rate, $k_{on}$ (μM$^{-1}$s$^{-1}$) on each protofilament. The interactions between αβ-tubulins was assumed to occur with longitudinal and lateral bond energies, $G_{Long,αβ-αβ}$ and $G_{Lat,αβ-αβ}$, respectively, similar to previous literature (*Mickolajczyk et al., 2019*; *VanBuren et al., 2002*; *Ayaz et al., 2014*). The longitudinal bond energy between γ-/αβ-tubulin, $G_{Long,γ-αβ}$ determines the dwell time of αβ-tubulin dimers on γ-TuRC. An open conformation of native γ-TuRC was assumed, as observed in recent structural work (*Liu et al., 2020*; *Wieczorek et al., 2020*), where lateral interactions between tubulins on neighboring sites were not allowed. A thermodynamic barrier, $G_{γTuRC-conf}$ and a pre-factor rate constant $k_{γTuRC-conf}$ (s$^{-1}$) determine the transition from this open to *closed* γ-TuRC conformation where lateral tubulin interactions can occur (*Figure 5A*). As αβ-tubulin dimers assemble on γ-TuRC, the free energy of this transition decreases by the total energy of all $n$ lateral bonds that can be formed, $G_{γTuRC-conf} - nG_{Lat,αβ-αβ}$.

MT growth parameters were determined by fitting to experimental growth speed curves (*Figure 5—figure supplement 1A*) and were found to be similar to previous estimates (*Mickolajczyk et al., 2019*; *VanBuren et al., 2002*). Based on our biochemical measurement (*Figure 5B*), $G_{Long,γ-αβ}$ was estimated to be higher than $G_{Long,αβ-αβ}$, while a wide range was explored for the other parameters. The resulting model produces a sharp transition from zero-MT length to a continuously growing MT upon γ-TuRC closure (*Figure 5B*) that occurs at variable time points for each realization of the model (*Figure 5B* and *Figure 5—figure supplement 1B-C(i)*). This qualitatively recapitulates the dynamics of γ-TuRC-mediated nucleation events observed experimentally.

Nucleation kinetics and the power-law dependence on αβ-tubulin concentration was obtained by simulating hundreds of model realizations. While $k_{γTuRC-conf}$ and $ΔG_{Long,γ-αβ}$ do not alter the power-law exponent significantly, they set the rate of nucleation at a specific αβ-tubulin concentration (*Figure 5—figure supplement 1B-C*). The thermodynamic barrier, $ΔG_{γTuRC-conf}$ instead determines the power-law exponent and the number of αβ-tubulins in the rate-limiting, transition state (*Figure 5C*). At $ΔG_{γTuRC-conf}<2.5k_BT$, cooperative assembly of 1-2 αβ-tubulins suffice to nucleate MTs, while at high $ΔG_{γTuRC-conf}>20k_BT$, more than 5 αβ-tubulins assemble cooperatively for successful MT nucleation. At an intermediate $ΔG_{γTuRC-conf} = 10k_BT$, MT nucleation kinetics and its power-law dependence recapitulates our experimental measurements (compare *Figure 5—figure supplement 2A* with *Figure 2C-D*). Here, γ-TuRCs minimally nucleate MTs at 7μM tubulin, MT nucleation increases

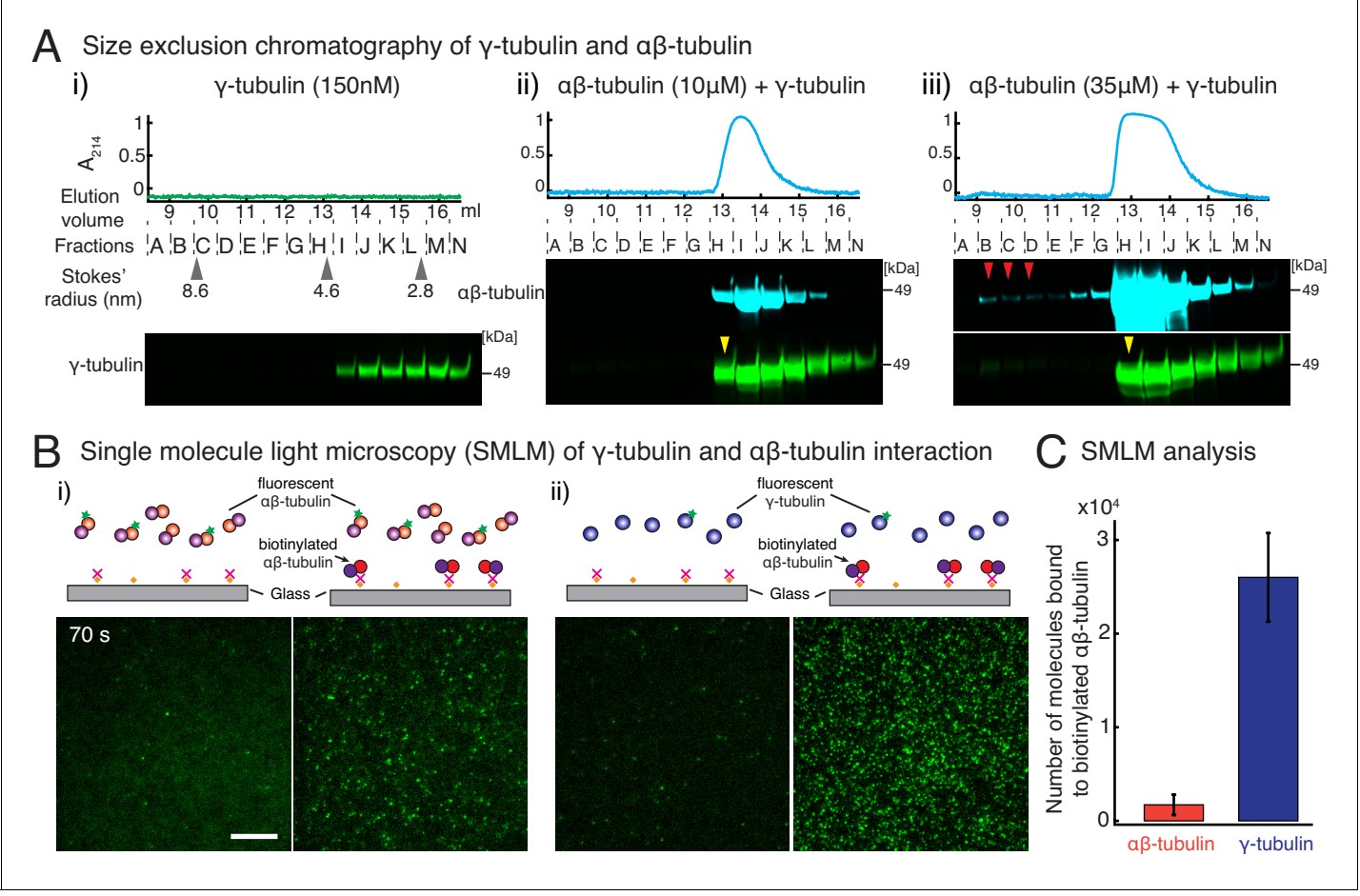

**Figure 4.** γ-tubulin binds to αβ-tubulin with a high affinity. (**A**) Size-exclusion chromatography was performed with 150 nM of γ-tubulin alone (**i**) and with 35 μM and 10 μM αβ-tubulin in (**ii**) and (**iii**), respectively. Gel filtration fractions were analyzed via SDS–PAGE followed by immunoblot with γ-tubulin and αβ-tubulin antibodies. A shift in the γ-tubulin elution to fraction H was observed with both 35 μM and 10 μM αβ-tubulin, denoting complex formation with αβ-tubulin. See *Figure 4—figure supplement 1A*. Stokes' radii of reference proteins: thyroglobulin (8.6 nm), aldolase (4.6 nm) and ovalbumin (2.8 nm), are marked at their elution peak. Size exclusion runs were repeated three times, with the exception of 10 μM αβ-tubulin run that was performed twice. (**B**) Single molecule microscopy was performed with γ-tubulin and αβ-tubulin. Control buffer (left panels, (**i**) and (**ii**)) or biotinylated αβ-tubulin (right panels, (**i**) and (**ii**)) was attached to coverslips, incubated with fluorescent αβ-tubulin (**i**) or γ-tubulin (**ii**) molecules, set as 0 s, and their binding at 60–90 s. (**C**) Number of bound molecules were analyzed for the first 15 s of observation described in Materials and methods. Experiments and analyses in (**B–C**) were repeated identically two times, pooled and reported. n = 56 data points each were displayed as mean ± std in the bar graph in (**C**). Further confirmed with a third supporting experimental set where the observation began later at 180 s and was therefore, not pooled. See also *Figure 4—figure supplements 1–2*.

The online version of this article includes the following source data and figure supplement(s) for figure 4:

**Source data 1.** Source data for *Figure 4C*.
**Figure supplement 1.** Purified γ-tubulin binds αβ-tubulin with high affinity in vitro.
**Figure supplement 2.** Purified γ-tubulin nucleates microtubules and assembles laterally into filaments.

non-linearly with tubulin concentration, and 4 ± 0.4 αβ-tubulins compose the transition state (*Figure 5—figure supplement 2A* and *Figure 5C*, green curve highlighted with an asterisk).

As a further validation of our model, we simulated the dynamics of MT nucleation from blunt MT seeds. Here, we assumed that MT assembly begins from a closed γ-TuRC geometry where all longitudinal bond energies were set equal to $G_{Long,αβ−αβ}$ (*Figure 5—figure supplement 2B*). The simulations predict near complete MT assembly at minimal αβ-tubulin concentration of 2μM and transition state of 1.1 ± 0.1 αβ-tubulins (*Figure 5—figure supplement 2A*), in agreement with MT assembly from blunt seeds that we measured experimentally (*Figure 3D-E*). Thus, our Monte Carlo simulations accurately capture the detailed dynamics of MT nucleation from γ-TuRC.

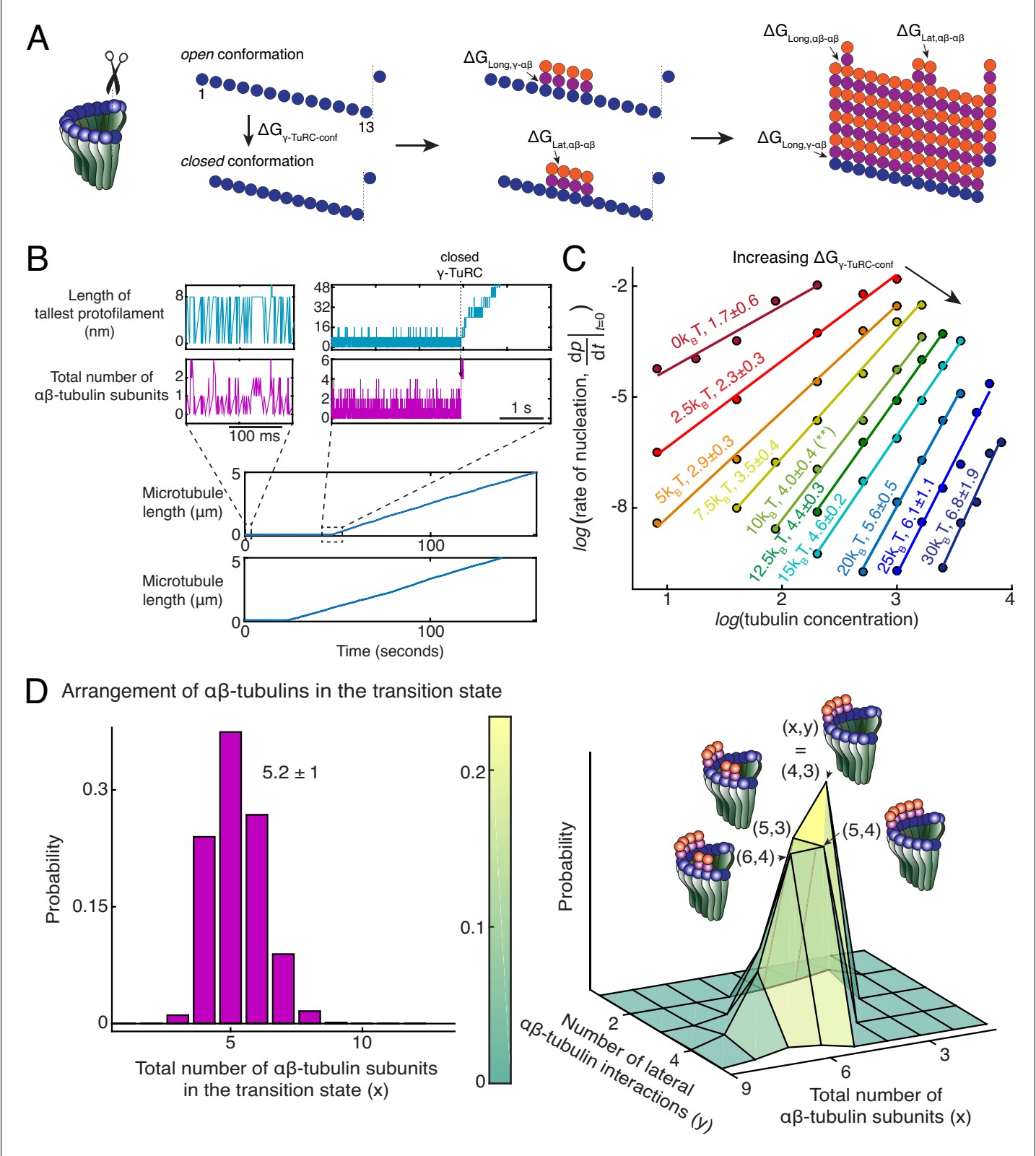

**Figure 5.** Monte Carlo simulations of microtubule nucleation from γ-TuRC. (**A**) Kinetic Monte Carlo simulations of MT nucleation were performed. Helical MT lattice was simulated with 13 protofilaments and a pitch of 3 tubulin monomers across the seam. Native γ-TuRC was simulated in an open conformation and was allowed transition into a closed conformation with a thermodynamic penalty of $G_{\gamma TuRC-conf}$. αβ-tubulin dimers form longitudinal bonds with energies, $G_{Long,\gamma-\alpha\beta}$ and $G_{Long,\alpha\beta-\alpha\beta}$ to γ-tubulin and other αβ-tubulins, respectively, and lateral bond with energy, $G_{Lat,\alpha\beta-\alpha\beta}$ with

*Figure 5 continued on next page*

Figure 5 continued

neighboring αβ-tubulin dimers. (B) MT length (μm) versus time (seconds) traces of two independent simulations are presented (bottom). MT nucleation occurs are variable time points for each model realization. Zoomed-in insets of the first simulation show the length of the tallest protofilament (nm) and total number of αβ-tubulin dimers assembled in the first 200 ms and 5 s near the transition state of the simulation. (C) Simulations were performed with $k_{on} = 1.3 \times 10^6 \ (M^{-1}s^{-1}pf^{-1})\Delta G_{Long,\alpha\beta-\alpha\beta} = -7.2k_BT$, $\Delta G_{Lat,\alpha\beta-\alpha\beta} = -6.5k_BT$, $\Delta G_{Long,\gamma-\alpha\beta} = 1.1\Delta G_{Long,\alpha\beta-\alpha\beta}$, $k_{\gamma TuRC-conf} = 0.01s^{-1}$. $\Delta G_{\gamma TuRC-conf}$ was varied from $+(0 - 30)k_BT$. Tubulin concentration was varied from 2.5 to 50 μM. 200 simulations were performed for a given tubulin concentration at every parameter set, except for $\Delta G_{\gamma TuRC-conf} = 10k_BT$ where 500 simulations were performed. From probability of MT nucleation ($p(t)$) versus time (t) curves, the initial rate of nucleation $\frac{dp}{dt}\big|_{t\to 0}$ was measured and plotted against concentration on a log-log axis as detailed in Materials and methods. (D) With the parameters defined above and $\Delta G_{\gamma TuRC-conf} = 10k_BT$, the transition state at the time of γ-TuRC's conformational change was recorded for $n=2119$ simulations. Normalized histogram of total number of αβ-tubulin dimers is plotted (left). Three-dimensional probability distribution of total number of αβ-tubulin dimers (x) and number of lateral αβ-tubulin interactions (y) is plotted (right). The most populated transition states are denoted with coordinates (x,y) and schematized. See also *Figure 5—figure supplements 1*, *2*.

The online version of this article includes the following figure supplement(s) for figure 5:

**Figure supplement 1.** Parameter variation in Monte Carlo simulations of γ-TuRC-mediated nucleation.
**Figure supplement 2.** Simulations of microtubule nucleation from γ-TuRC and from blunt seeds.

## Arrangement of αβ-tubulin dimers in transition state for γ-TuRC-mediated microtubule nucleation

We next characterized the dynamics of αβ-tubulins during MT nucleation from γ-TuRC by examining the time traces from individual model simulations. First, prior to MT nucleation, we observe longitudinal association of individual αβ-tubulins either to the γ-tubulin sites on the open γ-TuRC or, less frequently, with existing αβ-tubulin in a protofilament (*Figure 5B*, left insets). These αβ-tubulins dissociate rapidly in the absence of additional lateral bond energy. Once a MT lattice is assembled, persistence of αβ-tubulin dimers with both longitudinal and lateral contacts drive the growth of plus-end. Analogous observations during growth of a MT plus-end also show rapid dissociation of αβ-tubulin that form only a longitudinal contact, while ones with additional lateral contacts persist (*Mickolajczyk et al., 2019*). At the sharp transition prior to MT assembly (*Figure 5B*, right insets), we find that many αβ-tubulin dimers stochastically assemble on neighboring sites on γ-TuRC. Favorable Gibbs free energy from the lateral interaction between these αβ-tubulin dimers overcomes the energy penalty of the conformational change and transitions γ-TuRC into a closed state.

Finally, we characterize the arrangement of αβ-tubulin dimers in the rate-limiting, transition state that results in a closed γ-TuRC conformation prior to MT polymerization. A variable total number of αβ-tubulin dimers with an average of 5.2 ± 1 (*n* = 2119 simulations) were present on γ-TuRC at the transition state (*Figure 5D*, left). To our surprise, αβ-tubulin subunits in the transition state assemble on neighboring sites into laterally arranged groupings (*Figure 5D*, right). The most probable transition state is composed of four αβ-tubulin arranged on neighboring sites that form three lateral bonds when the γ-TuRC conformation changes to a closed one. The other probable states have 5 αβ-tubulins arranged laterally in two groups of 2 and 3 dimers each, or in two groups of 1 and 4 dimers each, and 6 αβ-tubulins arranged in two groups of 2 and 4 dimers, or in two groups of 3 dimers each. Most importantly, in these transition states, the free energy gained from the lateral bonds between αβ-tubulins compensates for the thermodynamic barrier posed by γ-TuRC's open conformation to allow for MT nucleation. Notably, the laterally-arranged group of 4 αβ-tubulin dimers physically represents the power-law exponent measured from the average nucleation kinetics (*Figure 2D*).

## Role of putative activation factors in γ-TuRC-mediated nucleation

Next, we investigated how accessory factors regulate γ-TuRC-dependent MT nucleation. While several activation factors (*Choi et al., 2010*; *Liu et al., 2014*; *Alfaro-Aco et al., 2017*) have been proposed to enhance the MT nucleation activity of γ-TuRC, the function of these putative activation factors remains to be tested with a sensitive and direct assay. We incubated the purified γ-TuRC activation domain (γ-TuNA) (*Choi et al., 2010*) from *Xenopus laevis* protein CDK5RAP2 with γ-TuRC at high concentrations to maximally saturate the binding sites on γ-TuRC (*Figure 6A*), and further supplemented additional γ-TuNA with αβ-tubulin used during the nucleation assay. Measurement of nucleation activity revealed that CDK5RAP2's γ-TuNA domain increases γ-TuRC-mediated nucleation

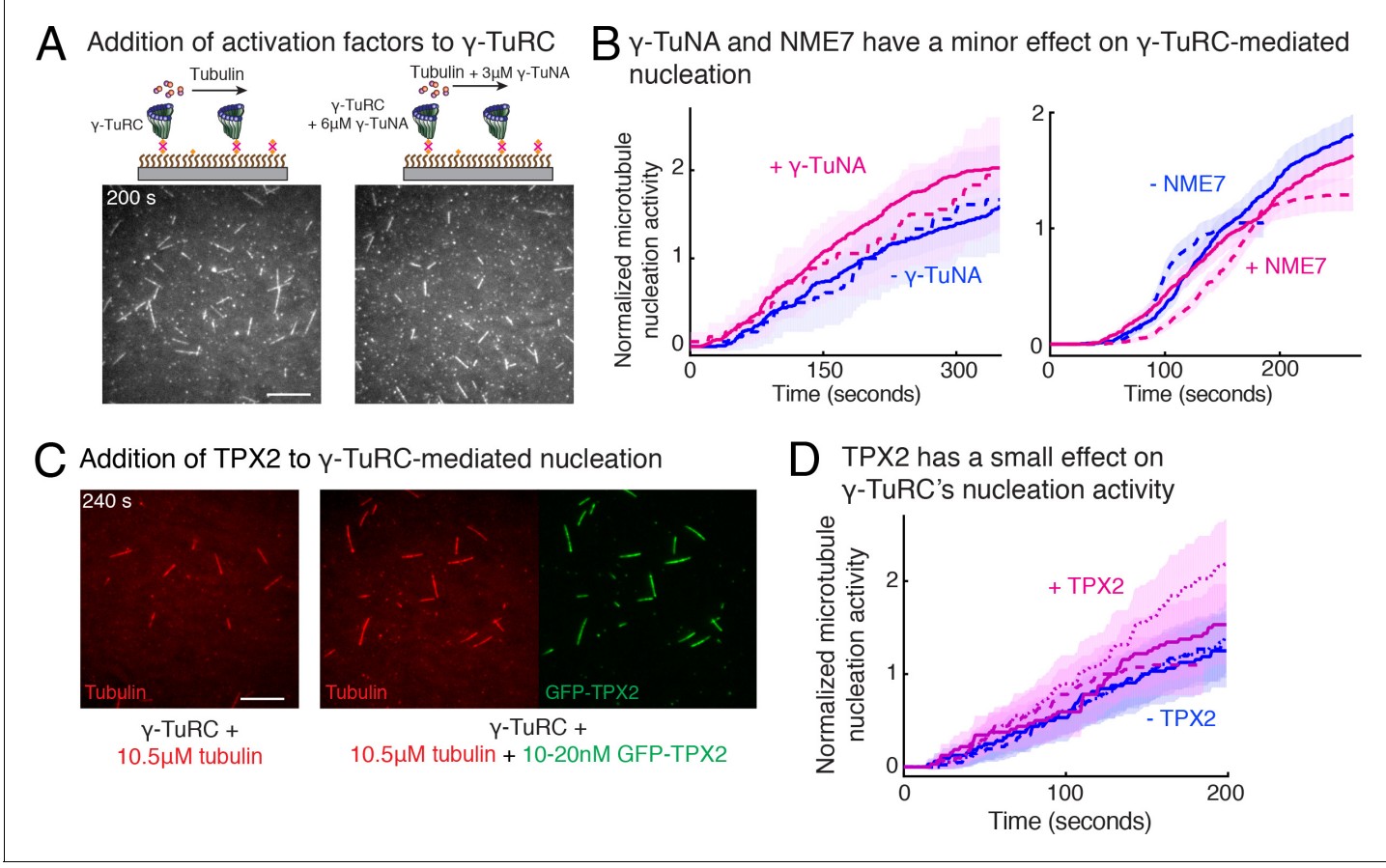

**Figure 6.** Regulation of γ-TuRC-mediated nucleation by putative activation factors. (**A**) A constant density of γ-TuRC molecules were attached without (left) and with (right) 6μM CDK5RAP2's γ-TuNA motif and 10.5μM tubulin ± 3μM additional γ-TuNA was added. Scale bar, 10 μm. (**B**) MTs nucleated from γ-TuRC molecules were analyzed and 3-6 μM CDK5RAP2's γ-TuNA motif (left) or 1-6 μM NME7 (right). Experiments and analyses in (**A–B**) were individually repeated twice on different days of experimentation with independent or same γ-TuRC preparations. Number of MTs nucleated in control reactions at 200 s for γ-TuNA, and at 150 s for NME7 was set to 1 to account for variable γ-TuRC concentration across purifications, all data were pooled and reported. Individual datasets with ±γ-TuNA and ±NME7 is represented with solid or dashed curves. Shaded regions represent 95% confidence interval ($n \pm 2\sqrt{n}$) from each dataset in the number of nucleated MTs ($n$) assuming a Poisson distribution as described in Materials and methods. (**C–D**) A constant density of γ-TuRC molecules were attached and 10.5 μM tubulin ± 10-20 nM GFP-TPX2 was added. Experiments and analyses were repeated thrice with independent γ-TuRC preparations. To account for the variable γ-TuRC concentration across purifications, the number of MTs nucleated in control reactions at 150 s was set to 1. All data were pooled and reported. Individual dataset with ±TPX2 is represented with solid, dashed or dotted curves. Shaded regions represent 95% confidence interval ($n \pm 2\sqrt{n}$) from each dataset in the number of nucleated MTs ($n$) assuming a Poisson distribution as described in Methods. See *Fiure 6—figure supplement 1* and *Video 6*.

The online version of this article includes the following source data and figure supplement(s) for figure 6:

**Source data 1.** Source data for *Figure 6* panels B, D.

**Figure supplement 1.** Effect of putative activation factor NME7 on γ-TuRC-mediated nucleation.

---

only by 1.4 (±0.02) -fold (mean ± std, *n* = 2) at *t* = 180 s, falling within the 95% confidence intervals of the control reactions (*Figure 6A–B* and *Video 6*). Another putative activator, NME7 (*Liu et al., 2014*), when added to γ-TuRC at saturating concentrations (*Wühr et al., 2014*; *Figure 6—figure supplement 1* and *Video 6*), did not increase γ-TuRC's nucleation activity (*Figure 6B*). Finally, we assessed the protein TPX2 that not only contains a split γ-TuNA and overlapping SPM (*Alfaro-Aco et al., 2017*), but also functions as an anti-catastrophe factor in vitro (*Wieczorek et al., 2015*; *Roostalu et al., 2015*) and was proposed to stimulate γ-TuRC-mediated nucleation (*Alfaro-Aco et al., 2017*; *Tovey and Conduit, 2018*; *Zhang et al., 2017*). TPX2 also had a small increase on the nucleation activity of γ-TuRC by 1.2 (±0.3) -fold (mean ± std, *n* = 3) at *t* = 180 s, but bound strongly along the MT lattice (*Figure 6C–D* and *Video 6*). While high concentration of TPX2 forms

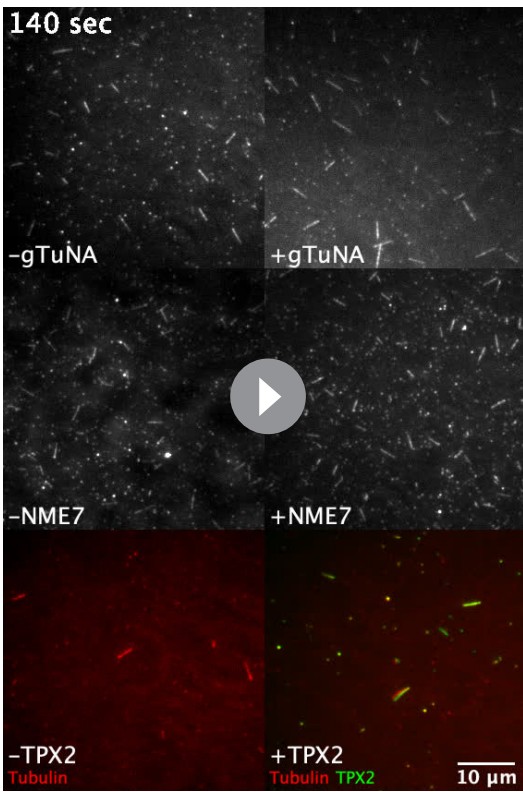

**Video 6.** γ-TuNA motif from CDK5RAP2, NME7 and TPX2 do not significantly increase γ-TuRC-mediated microtubule nucleation. Top panels: γ-TuRC was immobilized on coverslips with control buffer (left) or with 6 μM γ-TuNA motif from CDK5RAP2 (right) and MT nucleation was observed upon introducing fluorescent 10.5 μM αβ-tubulin (gray) without or with 3 μM γ-TuNA, respectively. Middle panels: γ-TuRC was immobilized on coverslips with control buffer (left) or with 6 μM NME7 (right) and MT nucleation was observed upon introducing fluorescent 10.5 μM αβ-tubulin (gray) without or with 1 μM NME7, respectively. Bottom panels: γ-TuRC was immobilized on coverslips and MT nucleation was observed upon introducing fluorescent 10.5 μM αβ-tubulin (pseudo-colored as red) without or with 10 nM GFP-TPX2 (right, labeled as green). TPX2 bound along the nucleated MTs but did not significantly increase the MT nucleation activity of γ-TuRC molecules. Elapsed time is shown in seconds, where time-point zero represents the start of reaction. Scale bar, 10 μm.
https://elifesciences.org/articles/54253#video6

condensates with αβ-tubulin and promotes spontaneous MT nucleation (*Roostalu et al., 2015*; *King and Petry, 2020*), near its endogenous concentration of TPX2 (*Thawani et al., 2019*) used here, TPX2 is able to saturates the MT lattice, yet it does not significantly increase γ-TuRC-mediated nucleation, in agreement with the physiological observations (*Alfaro-Aco et al., 2017*). Thus, the putative activation motif of CDK5RAP2, full-length NME7 or TPX2 all have minor effects on γ-TuRC's MT nucleation activity.

## XMAP215 promotes microtubule nucleation by strengthening the longitudinal bond energy between γ-TuRC and αβ-tubulin

Recently, XMAP215 was discovered to be a nucleation factor that synergizes with γ-TuRC in *X. laevis* and *S. cerevisiae* (*Thawani et al., 2018*; *Gunzelmann et al., 2018*), or works in an additive manner with γ-tubulin (*King et al., 2020*). To investigate how XMAP215 participates in MT nucleation, we performed single molecule experiments with XMAP215 and γ-TuRC. At low tubulin concentrations of 3.5 μM and 7 μM, where none or little MT nucleation occurs from γ-TuRCs alone (*Figure 7A* and *Figure 7—figure supplement 1A*), as shown earlier. Strikingly, the addition of XMAP215 induced many surface-attached γ-TuRCs to nucleate MTs, resulting in a drastic increase in number of nucleated MTs by 25 (±9) - fold (mean ±std, n = 3) within t = 120 s (*Figure 7A–B*, Fig. *Figure 7—figure supplement 1B* and *Video 7*). By directly visualizing γ-TuRC and XMAP215 molecules during the nucleation reaction (*Figure 7C*), we found that XMAP215 and γ-TuRC molecules first form a complex from which a MT was then nucleated (*Figure 7C* and *Video 8*). For 76% of the events (n = 56), XMAP215 visibly persisted between three to ≥300 s on γ-TuRC before MT nucleation. After MT nucleation, XMAP215 molecules polymerize and track with the MT plus-end. For 50% of nucleation events (n = 58), some XMAP215 molecules remained on the minus-end together with γ-TuRC, while for the other 50% of events, XMAP215 was not observed on the minus-end after nucleation. This suggests that XMAP215 molecules nucleate with γ-TuRC and then con-

tinue polymerization of the plus-end.

How does XMAP215 enable MT nucleation from γ-TuRC? We titrated αβ-tubulin at constant γ-TuRC and XMAP215 concentrations and measured the kinetics of nucleation (*Figure 7—figure supplement 1C* and *Figure 7D*). XMAP215 effectively decreases the minimal tubulin concentration necessary for MT nucleation from γ-TuRC to 1.6 μM (*Figure 7—figure supplement 1C*), very close to the minimal concentration for plus-end polymerization. As before, we calculated the composition of the transition state by measuring the power-law dependence between the MT nucleation rate and

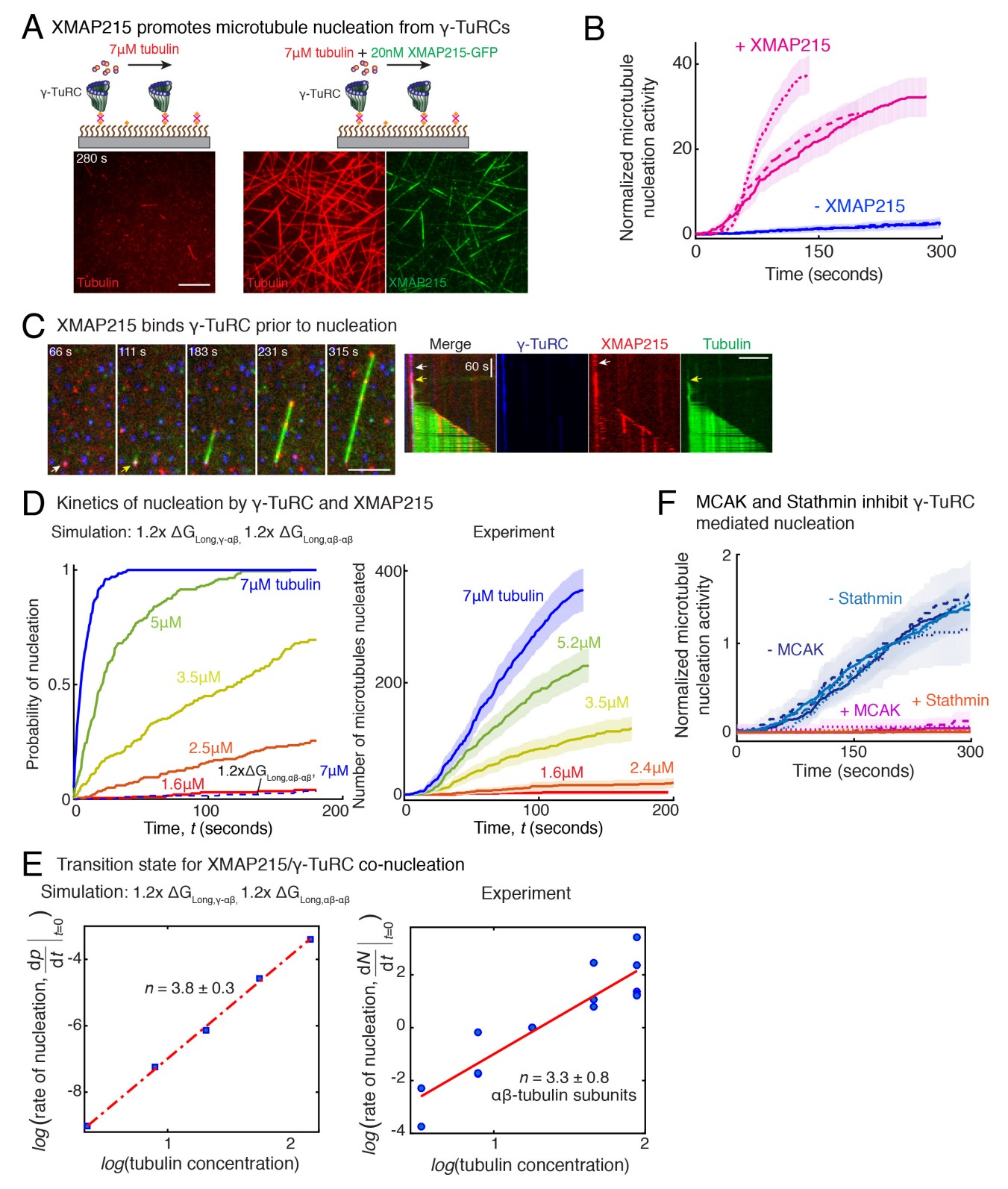

**Figure 7.** Role of XMAP215 and microtubule-associated proteins in microtubule nucleation with γ-TuRC. (**A**) γ-TuRCs were attached and 7μM tubulin (pseudo-colored in red) ± 20nM XMAP215-GFP (pseudo-colored in green) was added. Scale bar, 10 μm. Experiments and analyses in (**A–B**) were repeated thrice with independent γ-TuRC preparations. (**B**) Number of MTs nucleated ($N(t)$) over time (**t**) was measured and control reactions at 120 s was set to 1 to account for variable γ-TuRC concentration across purifications, all data were pooled and reported. Individual datasets with ±XMAP215 is
*Figure 7 continued on next page*

*Figure 7 continued*

represented with solid or dashed curves. Shaded regions represent 95% confidence interval ($n \pm 2\sqrt{n}$) from each dataset in the number of nucleated MTs ($n$) assuming a Poisson distribution as described in Materials and methods. See also *Figure 7—figure supplement 1A-B*. (C) Sequence of events during cooperative MT nucleation by γ-TuRC and XMAP215 was visualized using labeled γ-TuRC (blue), XMAP215 (red) and tubulin (green) represented in a time sequence and kymograph. γ-TuRC and XMAP215 form a complex prior to MT nucleation. XMAP215 molecules reside on γ-TuRC for before MT nucleation. The experiment was repeated a total of eight times with two independent γ-TuRC preparations and independent XMAP215 purifications. Scale bar, 5μm. (D–E) Number of MTs nucleated ($N(t)$) over time ($t$) was measured after titrating tubulin with constant γ-TuRC and XMAP215 concentration. XMAP215/γ-TuRC molecules nucleate MTs from 1.6 μM tubulin. Shaded regions represent 95% confidence interval ($n \pm 2\sqrt{n}$) in the number of nucleated MTs ($n$) assuming a Poisson distribution as described in Materials and methods. (E) Number of tubulin dimers ($n$) in the critical nucleus on cooperative nucleation by γ-TuRC/XMAP215 was obtained as 3.3 ± 0.8 from the equation $\frac{dN}{dt}\big|_{t \to 0} = kC_{tub}^n$ displayed on a log-log axis as detailed in Materials and methods. The rate of nucleation at 3.5μM was set to 1 to normalize differences in γ-TuRC concentration from individual experiments. Experiment and analyses in (D–E) was repeated thrice over the entire concentration range with independent γ-TuRC preparations, and fewer concentration points were repeated another two times. All five datasets were pooled and data points from a total of 18 nucleation-time curves are reported in (E). Simulations were adapted to understand how XMAP215 changes the thermodynamics of γ-TuRC-mediated nucleation. Parameter values used: $k_{on} = 1.3 \times 10^6 \ (M^{-1}s^{-1}pf^{-1})$, $G_{Long,\alpha\beta-\alpha\beta} = -8.64 k_B T$, $G_{Lat,\alpha\beta-\alpha\beta} = -6.2 k_B T$, $G_{Long,\gamma-\alpha\beta} = -9.5 k_B T$, $k_{\gamma TuRC-conf} = 0.01 s^{-1}$ and $G_{\gamma TuRC-conf} = 10 k_B T$. Compared to simulations for γ-TuRC alone (Figure S6A), either $G_{Long,\alpha\beta-\alpha\beta}$ was increased 1.2-fold, as proposed previously (*VanBuren et al., 2002*), or both $G_{Long,\alpha\beta-\alpha\beta}$ and $G_{Long,\alpha\beta-\alpha\beta}$ were increased 1.2-fold. 200 simulations each were performed for a range of tubulin concentration 1.6-7 μM. Probability of MT nucleation ($p(t)$) versus time ($t$) is displayed in (D). The initial rate of nucleation $\frac{dp}{dt}\big|_{t \to 0}$ was measured at each tubulin concentration and plotted against concentration on a log-log axis in (E). Linear curve was fit for $n=5$ simulated data points, and critical nucleus of 3.8 ± 0.3 αβ-tubulins. Increasing all longitudinal bond energies reproduces the effect of XMAP215 on γ-TuRC-mediated nucleation. (F) Number of MTs nucleated was measured to assess the effect of inhibitory MAPs MCAK or Stathmin on γ-TuRC-mediated nucleation. 10.5μM tubulin ± 10nM MCAK, or 7-10.5μM tubulin ± 2-5μM Stathmin was added to attached γ-TuRC- molecules, and MCAK and Stathmin were both found to inhibit γ-TuRC-mediated nucleation. Experiments and analyses for both MAPs were repeated thrice individually with independent γ-TuRC preparations. Number of MTs nucleated in control reactions at 200 seconds was set to 1 to account for variable γ-TuRC concentration across purifications, all data was pooled and reported. Individual dataset with ± MCAK are reported with solid, dashed or dotted curves. For Stathmin, two datasets for 10.5 μM tubulin ± 5 μM Stathmin are reported with solid and dashed lines, and one dataset for 7 μM tubulin ± 2 μM Stathmin in dotted line. Shaded regions represent 95% confidence interval ($n \pm 2\sqrt{n}$) from each dataset in the number of nucleated MTs ($n$) assuming a Poisson distribution as described in Materials and methods. See *Figure 7—figre supplement 1*, *2* and *Videos 7*, *8*, *9*.

The online version of this article includes the following source data and figure supplement(s) for figure 7:

**Source data 1.** Source data for *Figure 7* panels B, D, E, F and *Figure 7—figure supplement 1* panels B, E.
**Figure supplement 1.** Role of XMAP215 on γ-TuRC-mediated microtubule nucleation.
**Figure supplement 2.** MCAK and Stathmin inhibit γ-TuRC-mediated nucleation.

tubulin concentration with a resulting cooperative assembly of 3.3 ± 0.8 αβ-tubulin dimers occurs (*Figure 7E*). This suggests that XMAP215 does not lower the thermodynamic barrier to nucleation by altering the geometry of γ-TuRC. Further, neither the N-terminus, containing TOG1-4 domains, nor the C-terminus of XMAP215, containing the TOG5 and C-terminal domain that directly interact with γ-tubulin (*Thawani et al., 2018*), stimulate additional nucleation from γ-TuRC (*Figure 7—figure supplement 1D–E*).

Finally, we used our simulations to understand the thermodynamics underlying the MT nucleation activity of XMAP215. Based on its role in accelerating both MT polymerization and nucleation (*Thawani et al., 2018*; *Gunzelmann et al., 2018*; *Flor-Parra et al., 2018*), we implicitly modeled the thermodynamic effect of XMAP215's activity by strengthening the longitudinal tubulin bonds, as described previously (*VanBuren et al., 2002*). The simulation where only the longitudinal αβ-/αβ-tubulin bond is strengthened does not capture the enhancement of MT nucleation by XMAP215 (*Figure 7D*, left). Instead, simulations where both the longitudinal γ-/αβ-tubulin and αβ-/αβ-tubulin bond energies are increased by 1.2-fold captures the accelerated kinetics of MT nucleation at low αβ-tubulin concentrations. These simulations also predict a similar transition state composition as measured experimentally (*Figure 7D–E*, left), supporting XMAP215's role in strengthening γ-/αβ-tubulin interactions at the nucleation interface. Altogether, our results confirm that XMAP215 indeed functions synergistically with γ-TuRC, in agreement with recent works (*Consolati et al., 2020*; *Thawani et al., 2018*; *Gunzelmann et al., 2018*). Most importantly, our results show that, while the transition state is defined by γ-TuRC's conformation, XMAP215 strengthens the longitudinal γ-/αβ-tubulin bond to function as a bona-fide nucleation factor.

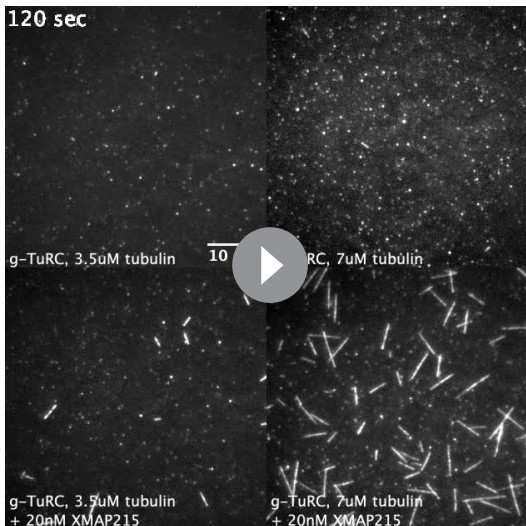

**Video 7.** XMAP215 increases microtubule nucleation activity of γ-TuRC. γ-TuRC was immobilized on coverslips and MT nucleation was assayed with low concentration of fluorescent αβ-tubulin (3.5 μM and 7 μM) without (top panels) or with 20 nM XMAP215-GFP (bottom panels). XMAP215 induces MT nucleation from γ-TuRC. Elapsed time is shown in seconds, where time-point zero represents the start of reaction. Scale bar, 10 μm.

https://elifesciences.org/articles/54253#video7

## Inhibition of γ-TuRC-mediated nucleation by specific microtubule-associated proteins

Finally, we asked whether specific MAPs could have an inhibitory effect on MT nucleation from γ-TuRC. The two most abundant inhibitory MAPs in the cytosol, MCAK and Stathmin function by removing αβ-tubulin dimers from the MT lattice (*Hunter et al., 2003*; *Howard and Hyman, 2007*) or sequestering αβ-tubulin dimers (*Jourdain et al., 1997*; *Belmont and Mitchison, 1996*), respectively. We find that addition of either sub-endogenous concentration of MCAK, or near-endogenous Stathmin concentration (*Figure 7F*, *Figure 7—figure supplement 2* and *Video 9*) was sufficient to nearly abolish MT nucleation from all γ-TuRC molecules. Thus, γ-TuRC-mediated nucleation is inhibited by MAPs that inhibit MT polymerization.

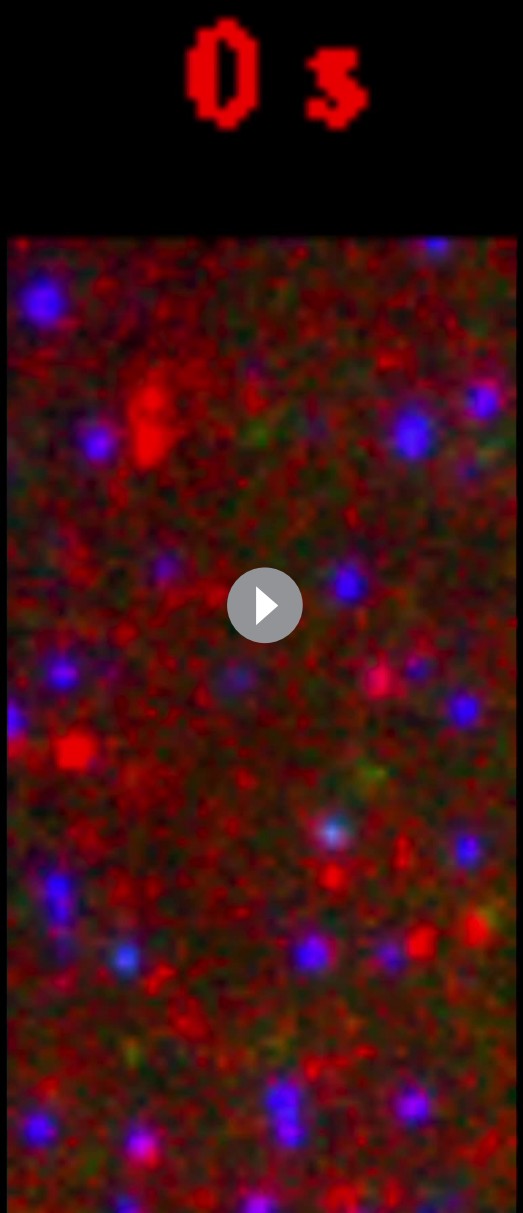

**Video 8.** Synergistic microtubule nucleation by γ-TuRC and XMAP215. Triple-color fluorescence microscopy was performed to observe the molecular sequence of events during MT nucleation from γ-TuRC and XMAP215. γ-TuRC (blue) and XMAP215 (red) formed a complex before MT nucleation occurred (pseudo-colored as green). For 50% of these events, XMAP215 remains on the nucleated minus-end. Elapsed time is shown in seconds, where time-point zero represents the start of reaction. Scale bar, 10 μm.

https://elifesciences.org/articles/54253#video8

## Discussion

Decades after the discovery of MTs, their αβ-tubulin subunits and the identification of γ-TuRC as the universal MT nucleator (*Oakley and Oakley, 1989*; *Moritz et al., 1995*; *Moritz et al., 1998*; *Zheng et al., 1995*; *Keating and Borisy, 2000*; *Wiese and Zheng, 2000*), it has remained poorly understood how MTs are nucleated and how this process is regulated in the cell (*Kollman et al., 2011*; *Roostalu and Surrey, 2017*; *Tovey and*

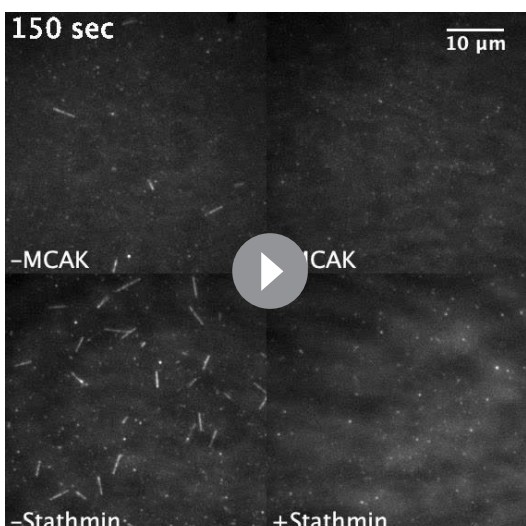

**Video 9.** MCAK and Stathmin inhibit γ-TuRC-mediated microtubule nucleation. Top panels: γ-TuRC was immobilized on coverslips and MT nucleation was observed upon introducing fluorescent 10.5 μM αβ-tubulin without (left) or with 10 nM MCAK (right). Bottom panels: γ-TuRC was immobilized on coverslips and MT nucleation was observed upon introducing fluorescent 10.5 μM αβ-tubulin without (left) or with 5 μM Stathmin (right). Elapsed time is shown in seconds, where time-point zero represents the start of reaction. Scale bar, 10 μm.
https://elifesciences.org/articles/54253#video9

*Conduit, 2018*). Here, we establish a single-molecule assay to study MT nucleation and combine it with computational modeling to identify the rate-limiting, transition state of γ-TuRC-mediated nucleation. We examine how biochemical features of γ-TuRC contribute to its the nucleation activity and regulation.

New methods and direct measurements developed in this study reconcile several prior observations for γ-TuRC-mediated MT nucleation. First, the nucleation activity of γ-TuRC has been found as variable and often low and similar to spontaneous MT assembly (*Moritz et al., 1995*; *Zheng et al., 1995*; *Kollman et al., 2015*; *Liu et al., 2020*; *Wieczorek et al., 2020*; *Consolati et al., 2020*; *Kollman et al., 2011*; *Roostalu and Surrey, 2017*), and γ-TuRC's requirement in the cell has been debated (*Hannak et al., 2002*; *Roostalu et al., 2015*; *Rogers et al., 2008*; *Raff, 2019*; *Woodruff et al., 2017*). Low concentration of γ-TuRC molecules obtained from endogenous purifications and lack of live observation of a growing or capped MT minus-end, which is needed to distinguish between γ-TuRC-mediated and spontaneous nucleation, could affect the assessment of γ-TuRC's nucleation activity. Second, because of technical challenges in the traditional setup where MTs are nucleated, fixed and spun down onto a coverslip (*Moritz et al., 1995*; *Zheng et al., 1995*; *Kollman et al., 2010*; *Kollman et al., 2015*; *Liu et al., 2020*; *Choi et al., 2010*; *Liu et al., 2014*; *Thawani et al., 2018*), variable assessment of the role of accessory factors (*Liu et al., 2020*; *Choi et al., 2010*; *Liu et al., 2014*) has been reported. Here, by developing a high-resolution assay that provides specific live information to visualize MT nucleation events from γ-TuRC and distinguish between non-γ-TuRC nucleated MTs, analyses system to measure its nucleation activity independent of concentration, as well as direct visualization of MAPs bound to γ-TuRC or the MT lattice allows us to unambiguously study γ-TuRC-mediated nucleation and its regulation by MAPs.

While the molecular architecture of γ-TuRC was revealed by recent cryo-EM structures (*Kollman et al., 2010*; *Liu et al., 2020*; *Wieczorek et al., 2020*; *Consolati et al., 2020*), the dynamics of MT nucleation from γ-TuRC and how it relates to γ-TuRC's specific biochemical features has remained unknown. By combining biochemical investigation with computational modeling, we show that 4 αβ-tubulin heterodimers on neighboring sites form the critical nucleus, that is the rate-limiting transition state on γ-TuRC. A model, in which γ-TuRC stochastically changes its conformation from an *open* to *closed* state, where the latter is stabilized by lateral αβ-tubulin interactions, comprehensively explains our experimental measurements. While native γ-TuRC purified from cytosol was used here, further activated γ-TuRC isolated from MTOCs may result in cooperativity between fewer αβ-tubulin dimers for successful nucleation. Likewise, MT assembly from pre-assembled, blunt seeds, could resemble nucleation from already closed γ-TuRCs. We find that the subsequent transition of the growing MT end from blunt- to tapered one, is not the major, rate-limiting step during nucleation from γ-TuRC. Notably, a parallel work also reported MT nucleation from single, human γ-TuRC molecules recently (*Consolati et al., 2020*). While the majority of findings agree with our work, 6.7 dimers were required in the critical nucleus and an overall lower activity of γ-TuRC (0.5%) was found (*Consolati et al., 2020*). Low structural integrity of purified γ-TuRC from incorporation of BFP-

tagged GCP2 and a higher ratio of γ-tubulin sub-complexes, or species-specific variation in γ-TuRC properties could explain these differences.

Our simulations further predict that a hypothetical low affinity between γ-/αβ-tubulin (*Rice et al., 2019*) is insufficient to induce any MT nucleation because αβ-tubulins that bind to γ-TuRC dissociate rapidly. Instead, our biochemical investigation show that the high affinity of γ-/αβ-tubulin interaction increases the dwell time of αβ-tubulin dimers on γ-TuRC and promotes γ-TuRC's MT nucleation activity, as predicted by our modeling. Finally, this net mechanism is thermodynamically favorable compared to spontaneous MT nucleation as the free energy of longitudinal γ-/αβ-tubulin interactions, $13\left(\Delta G_{Long,\gamma-\alpha\beta}\right)$, exceeds the energy penalty from conformational rearrangement of γ-TuRC, $\Delta G_{\gamma TuRC-conf}$. In sum, building on the recent structural work (*Liu et al., 2020*; *Wieczorek et al., 2020*; *Consolati et al., 2020*), our results show that the open γ-TuRC conformation and its transition to a closed one defines γ-TuRC's nucleation activity and transition state. In the future, it will be important to study how γ-TuRC transitions to a closed conformation with high-resolution structural studies, as well as how other biochemical properties, in addition to those modeled here, govern its nucleation activity. Our single molecule assay, kinetic analyses and computational modeling will be essential to complement and place atomic structures into a mechanism that explains how MT nucleation γ-TuRC occurs and how it is regulated.

Although spatial regulation of MT nucleation is achieved by localizing γ-TuRC to specific MTOCs as shown previously (*Kollman et al., 2011*; *Wiese and Zheng, 2006*; *Tovey and Conduit, 2018*; *Petry and Vale, 2015*), temporal regulation of MT nucleation had been proposed to occur through activation factors that modify γ-TuRC's conformation and upregulate its activity (*Kollman et al., 2011*; *Choi et al., 2010*; *Liu et al., 2014*; *Tovey and Conduit, 2018*; *Petry and Vale, 2015*). While several putative activation factors do not significantly enhance of γ-TuRC's nucleation activity as shown here, new factors, that are yet to be identified, may serve this role to alter γ-TuRC's conformation at MTOCs. Alternatively, we postulate another mechanism for temporal control governing the availability and localization of αβ-tubulin. In this model, locally concentrating soluble αβ-tubulin could upregulate the levels of γ-TuRC-mediated MT nucleation, for example as recently shown through accumulation of high concentration of tubulin dimers at the centrosome by MAPs (*Woodruff et al., 2017*; *Baumgart et al., 2019*) and by co-condensation of tubulin on MTs by TPX2 during branching MT nucleation (*King and Petry, 2020*), and finally via specific recruitment of tubulin on γ-TuRC through the binding of XMAP215 as shown here (*Thawani et al., 2018*; *Gunzelmann et al., 2018*; *Flor-Parra et al., 2018*).

## Supplementary materials

Supplementary Materials includes nine figures, nine videos, MATLAB code for simulations and source data.

## Materials and methods

### Purification of recombinant proteins

Full-length TPX2 with N-terminal Strep II-6xHis-GFP-TEV site tags was cloned into pST50Tr-STRHISNDHFR (pST50) vector (*Tan et al., 2005*) using Gibson Assembly (New England Biolabs). N-terminal 6xHis-tagged, *Xenopus laevis* Stathmin 1A was a gift from Christiane Wiese (University of Madison). N-terminal tagged 6xHis-TEV MCAK plasmid was a gift from *Ohi et al., 2004*. Wild-type XMAP215 with C-terminal GFP-7xHis plasmid was a gift from *Reber et al., 2013* and was used to clone XMAP215 with C-terminal SNAP-TEV-7xHis-StrepII tags as well as with C-terminal TEV-GFP-7xHis-StrepII tags, first into pST50 vector and further into pFastBac1 vector. TOG5-CT truncation of XMAP215 was produced by cloning amino acids 1091–2065 into pST50 vector with C-terminal GFP-7xHis-Strep tags. Human γ-tubulin TEV-Strep II-6xHis tags was codon-optimized for Sf9 expression, synthesized (Genscript), and further cloned into pFastBac1 vector. 6xHis tagged γ-TuNA (N-terminal aa 56–89 of *Xenopus laevis* CDK5RAP2) was also cloned into pST50 and expressed in *E. coli* Rosetta2 cells. Dual StrepII-6xHis-tagged *Xenopus laevis* NME7 was cloned into pFastBac1 vector, expressed, and purified from Sf9 cells.

TPX2, Stathmin and truncations of XMAP215 (TOG5-CT and TOG1-4) used in this study were expressed in *E. coli* Rosetta2 cells (EMD Millipore) by inducing with 0.5–1 mM IPTG for 12–18 hr at

16°C or 7 hr at 25°C. Wild-type XMAP215, MCAK and γ-tubulin were expressed and purified from Sf9 cells using Bac-to-Bac system (Invitrogen). The cells were lysed (EmulsiFlex, Avestin) and *E. coli* lysate was clarified by centrifugation at 13,000 rpm in Fiberlite F21-8 rotor (ThermoFisher) and Sf9 cell lysate at 50,000 rpm in Ti70 rotor (Beckman Coulter) for 30–45 min.

TPX2 was first affinity purified using Ni-NTA beads in binding buffer (50 mM Tris-HCl pH 8.0, 750 mM NaCl, 15 mM Imidazole, 2.5 mM PMSF, 6 mM BME) and eluted with 200 mM Imidazole. All proteins were pooled and diluted four-fold to 200 mM final NaCl. Nucleotides were removed with a Heparin column (HiTrap Heparin HP, GE Healthcare) by binding protein in 250 mM NaCl and isocratic elution in 750 mM NaCl, all solutions prepared in Heparin buffer (50 mM Tris-HCl, pH 8.0, 2.5 mM PMSF, 6 mM BME). Peak fractions were pooled and loaded on to Superdex 200 pg 16/600, and gel filtration was performed in CSF-XB buffer.

XMAP215-GFP-7xHis was purified using His-affinity (His-Trap, GE Healthcare) by binding in buffer (50 mM NaPO$_4$, 500 mM NaCl, 20 mM Imidazole, pH 8.0) and eluting in 500 mM Imidazole. Peak fractions were pooled and diluted 5-fold with 50 mM Na-MES pH 6.6, bound to a cation-exchange column (Mono S 10/100 GL, GE Healthcare) with 50 mM MES, 50 mM NaCl, pH 6.6 and eluted with a salt-gradient up to 1M NaCl. Peak fractions were pooled and dialyzed into CSF-XB buffer. XMAP215-SNAP-TEV-7xHis-StrepII or XMAP215-TEV-GFP-7xHis-StrepII was first affinity purified with StrepTrap HP (GE Healthcare) with binding buffer (50 mM NaPO$_4$, 270 mM NaCl, 2 mM MgCl$_2$, 2.5 mM PMSF, 6 mM BME, pH 7.2), eluted with 2.5 mM D-desthiobiotin. Peak fractions were pooled, concentrated and further purification via gel filtration (Superdex 200 10/300 GL) in CSF-XB buffer containing 150 mM KCl. For fluorescent labeling of SNAP-tag in XMAP215-SNAP-TEV-7xHis-StrepII, StrepTrap elution was cation-exchanged (Mono S 10/100 GL), peak fractions pooled and reacted with two-molar excess SNAP-substrate Alexa-488 dye (S9129, NEB) overnight at 4°C, followed by purification via gel filtration (Superdex 200 10/300 GL) in CSF-XB buffer. Approximately 70% labeling efficiency of the SNAP-tag was achieved.

γ-tubulin was purified by binding to HisTrap HP (GE Healthcare) in binding buffer (50 mM KPO$_4$ pH 8.0, 500 mM KCl, 1 mM MgCl$_2$, 10% glycerol, 5 mM Imidazole, 0.25 μM GTP, 5 mM BME, 2.5 mM PMSF), washing first with 50 mM KPO$_4$ pH 8.0, 300 mM KCl, 1 mM MgCl$_2$, 10% glycerol, 25 mM imidazole, 0.25 μM GTP, 5 mM BME), and then with 50 mM K-MES pH 6.6, 500 mM KCl, 5 mM MgCl$_2$, 10% glycerol, 25 mM imidazole, 0.25 μM GTP, 5 mM BME) and eluted in 50 mM K-MES pH 6.6, 500 mM KCl, 5 mM MgCl$_2$, 10% glycerol, 250 mM imidazole, 0.25 μM GTP, 5 mM BME. Peak fractions were further purified with gel filtration (Superdex 200 10/300 GL) in gel filtration buffer (50 mM K-MES pH 6.6, 500 mM KCl, 5 mM MgCl$_2$, 1 mM K-EGTA, 1 μM GTP, 1 mM DTT). For covalent labeling of γ-tubulin with Alexa-568 or Alexa-488 dye, peak gel filtration fractions were pooled and dialyzed into labeling buffer (50 mM KPO$_4$ pH 8.0, 500 mM KCl, 1 mM MgCl2, 2% glycerol, 25 μM GDP, 5 mM BME), reacted with 5 to 20-fold excess of Alexa-568 or Alexa-488 NHS ester (catalog # A20003, A20000, GE Healthcare) for 1 hr at 4°C, and unreacted dye was separated with size exclusion Superdex 200 10/300 GL in gel filtration buffer as above. 7% labeling of γ-tubulin was achieved.

γ-TuNA motif from CDK5RAP2 was purified by binding to Ni-NTA resin in binding buffer buffer (50 mM Tris-HCl pH 8, 500 mM NaCl, 20 mM Imidazole), eluted with 250 mM Imidazole and further purified by gel filtration into storage buffer (50 mM Tris-HCl pH 7.5, 200 mM NaCl). NME7 was purified similar to γ-TuNA by first Ni-NTA affinity followed by size exclusion, as described for γ-TuNA, except with salt concentration of 150 mM NaCl and additional 0.05% Tween-20, and further dialyzed into BRB80 for storage.

MCAK was first affinity purified by binding to His-Trap HP (GE Healthcare) in binding buffer (50 mM NaPO4, 500 mM NaCl, 6 mM BME, 0.1 mM MgATP, 10 mM Imidazole, 1 mM MgCl2, 2.5 mM PMSF, 6 mM BME, pH to 7.5), eluting with 300 mM Imidazole, followed by gel-filtration (Superdex 200 10/300 GL, GE Healthcare) in storage buffer (10 mM K-HEPES pH 7.7, 300 mM KCl, 6 mM BME, 0.1 mM MgATP, 1 mM MgCl$_2$, 10% w/v sucrose).

Stathmin was purified using His-affinity (His-Trap HP, GE Healthcare) by first binding in binding buffer (20 mM NaPO$_4$ pH 8.0, 500 mM NaCl, 30 mM Imidazole, 2.5 mM PMSF, 6 mM BME) and eluting with 300 mM Imidazole, followed by gel filtration (HiLoad 16/600 Superdex, GE Healthcare) into CSF-XB buffer (100 mM KCl, 10 mM K-HEPES, 5 mM K-EGTA, 1 mM MgCl$_2$, 0.1 mM CaCl$_2$, pH 7.7 with 10% w/v sucrose).

All recombinant proteins were flash-frozen and stored at −80°C, and their concentration was determined by analyzing a Coomassie-stained SDS-PAGE against known concentration of BSA (A7906, Sigma).

Bovine brain tubulin was labelled with biotin-, Cy5-, Alexa-488 or Alexa-568 NHS esters (GE Healthcare) as described previously (*Thawani et al., 2019*).

## Purification, biotinylated and fluorescent labeling of γ-TuRC

Endogenous γ-TuRC was purified from *Xenopus* egg extracts and labeled with the following steps at 4°C. 7–8 ml of meiotic extract from *Xenopus* laevis eggs, prepared as described previously (*Hannak and Heald, 2006*; *Murray and Kirschner, 1989*), was first diluted 5-fold with CSF-XBg buffer (10 mM K-HEPES, 100 mM KCl, 1 mM MgCl$_2$, 5 mM K-EGTA, 10% w/v sucrose, 1 mM DTT, 1 mM GTP, 10 µg/ml LPC protease inhibitors, pH 7.7), centrifuged to remove large aggregates at 3500 rpm (Thermo Sorvall Legend XTR) for 10 min, and the supernatant filtered sequentially with 1.2 µm and 0.8 µm Cellulose Acetate filters (Whatman) followed by 0.22 µm PES filter (ThermoFisher). γ-TuRC was precipitated by incubating with 6.5% w/v PEG-8000k (Sigma) for 30 min and centrifuged at 17,000 rpm (SS-34 rotor, ThermoScientific) for 20 min. γ-TuRC-rich pellet was resuspended in CSF-XB buffer with 0.05% v/v NP-40 using a mortar and pestle homogenizer, PEG was removed via centrifugation at 136,000 xg for 7 min in TLA100.3 (Beckman Ultracentrifuge), and supernatant was pre-cleared by incubating with Protein A Sepharose beads (GE LifeSciences #17127901) for 20 min. Beads were removed, γ-TuRC was incubated with 4–5 mg of a polyclonal antibody custom-made against C-terminal residues 413–451 of *X. laevis* γ-tubulin (Genscript) for 2 hr on gentle rotisserie, and further incubated with 1 ml washed Protein A Sepharose bead slurry for 2 hr. γ-TuRC-bound beads were washed sequentially with 30 ml of CSF-XBg buffer, 30 ml of CSF-XBg buffer with 250 mM KCl (high salt wash), 10 ml CSF-XBg buffer with 5 mM ATP (removes heat-shock proteins), and finally 10 ml CSF-XBg buffer before labeling. For biotinylation of γ-TuRC, beads were incubated with 25 µM NHS-PEG4-biotin (A39259, ThermoFisher) in CSF-XBg buffer for 1 hr at 4°C, and unbound biotin was removed by washing with 30 ml CSF-XBg buffer prior to elution step. For combined fluorescent and biotin labeling of γ-TuRC, the wash step after ATP-wash consisted of 10 ml of labelling buffer (10 mM K-HEPES, 100 mM KCl, 1 mM MgCl$_2$, 5 mM K-EGTA, 10% w/v sucrose, 0.5 mM TCEP, 1 mM GTP, 10 µg/ml LPC, pH 7.1) and fluorescent labelling was performed by incubating the beads with 1 µM Alexa-568 C$_5$ Maleimide (A20341, ThermoFisher). Unreacted dye was removed with 10 ml CSF-XBg buffer, beads were incubated with 25 µM NHS-PEG4-biotin (A39259, ThermoFisher) in CSF-XBg buffer for 1 hr at 4°C, and unreacted biotin removed with 30 ml CSF-XBg buffer. Labeled γ-TuRC was eluted by incubating 2–3 ml of γ-tubulin peptide (residues 413–451) at 0.4–0.5 mg/ml in CSF-XBg buffer with beads overnight. After 10–12 hr, γ-TuRC was collected by adding 1–2 ml CSF-XBg buffer to the column, concentrated to 200 µl in 30 k NMWL Amicon concentrator (EMD Millipore) and layered onto a continuous 10–50 w/w % sucrose gradient prepared in a 2.2 ml ultra-clear tube (11 × 34 mm, Beckman Coulter) using a two-step program in Gradient Master 108 machine. Sucrose gradient fractionation of γ-TuRC was performed by centrifugation at 200,000xg in TLS55 rotor (Beckman Coulter) for 3 hr. The gradient was fractionated from the top in 11–12 fractions using wide-bore pipette tips and peak 2–3 fractions were identified by immunoblotting against γ-tubulin with GTU-88 antibody (Sigma). γ-TuRC was concentrated to 80 µl in 30 k NMWL Amicon concentrator (EMD Millipore) and fresh purification was used immediately for single molecule assays. Cryo-preservation of γ-TuRC molecules resulted in loss of ring assembly and activity.

## Assessment of γ-TuRC with protein gel, immunoblot and negative stain electron microscopy

To assess the purity of γ-TuRC, 3–5 µl of purified γ-TuRC was visualized on an SDS-PAGE with SYPRO Ruby stain (ThermoFisher) following the manufacturer's protocol. Biotinylated subunits of γ-TuRC were assessed by immunoblotting with Streptavidin-conjugated alkaline phosphatase (S921, ThermoFisher). For further conjugation of Alexa-568 dye to γ-TuRC, fluorescently labeled subunits were assessed by visualizing an SDS-PAGE gel with Typhoon FLA 9500 (GE Healthcare) with LPG filter and 100 µm pixel size. γ-TuRC purification was also assessed by visualizing using electron microscopy. 4 µl of peak sucrose gradient fraction of γ-TuRC was pipetted onto CF400-Cu grids (Electron Microscopy Sciences), incubated at room temperature for 60 s and then wicked away. 2% uranyl

acetate was applied to the grids for 30 s, wicked away, and the grids were air-dried for 10 min. The grids were imaged using Phillips CM100 TEM microscope at 64,000x magnification.

## Preparation of functionalized coverslips

22 × 22 mm, high precision coverslips (170 ± 5 μm, Carl Zeiss, catalog # 474030-9020-000) were functionalized for single molecule assays based on a recent protocol (*Roostalu et al., 2015*; *Bieling et al., 2010*) with specific modifications. Briefly, coverslips were labeled on the surface to be functionalized by scratching 'C' on right, bottom corner, placed in Teflon racks, sonicated with 3N NaOH for 30 min, rinsed with water and sonicated in piranha solution (2 parts of 30 w/w % hydrogen peroxide and three parts sulfuric acid) for 45 min. Coverslips were rinsed thrice in water, and all water was removed by spin drying completely in a custom-made spin coater. Pairs of coverslips were made to sandwich 3-glycidyloxypropyl trimethoxysilane (440167, Sigma) on the marked sides, placed in glass petri dishes, and covalent reaction was performed in a lab oven at 75°C for 30 min. Coverslips were incubated for 15 min at room temperature, the sandwiches were separated, incubated in acetone for 15 min, then transferred to fresh acetone and quickly dried under nitrogen stream. Coverslip sandwiches were prepared with a small pile of well mixed HO-PEG-NH$_2$ and 10% biotin-CONH-PEG-NH$_2$ (Rapp Polymere) in glass petri dishes, warmed to 75°C in the lab oven until PEG melts, air bubbles were pressed out and PEG coupling was performed at 75°C overnight. The following day, individual coverslips were separated from sandwiches, sonicated in MilliQ water for 30 min, washed further with water until no foaming is visible, dried with a spin dryer, and stored at 4°C. Functionalized coverslips were used within 1 month of preparation.

Imaging chambers were prepared by first assembling a channel on glass slide with double sided tape strips (Tesa) 5 mm apart, coating the channel with 2 mg/ml PLL(20)-g[3.5]- PEG(2) (SuSOS) in dH$_2$O, incubating for 20 min, rinsing out the unbound PEG molecules with dH$_2$O and drying the glass slide under the nitrogen stream. A piece of functionalized coverslip was cut with the diamond pen and assembled functionalized face down on imaging chamber. The prepared chambers were stored at 4°C and used within a day of assembly.

## Microtubule nucleation assay with purified γ-TuRC

The imaging channel was prepared as follows. First, 5% w/v Pluronic F-127 in dH$_2$O was introduced in the chamber (1 vol = 50 μl) and incubated for 10 min at room temperature. The chamber was washed with 2 vols of assay buffer (80 mM K-PIPES, 1 mM MgCl$_2$, 1 mM EGTA, 30 mM KCl, 0.075% w/v methylcellulose 4000 cp, 1% w/v D-(+)-glucose, 0.02% w/v Brij-35, 5 mM BME, 1 mM GTP) with 0.05 mg/ml κ-casien (casein buffer), followed by 1 vol of 0.5 mg/ml NeutrAvidin (A2666, Thermo-Fisher) in casein buffer, incubated on a cold block for 3 min, and washed with 2 vols of BRB80 (80 mM K-PIPES, 1 mM MgCl$_2$, 1 mM EGTA pH 6.8). Five-fold dilution of γ-TuRC in BRB80 was introduced in the flow chamber and incubated for 10 min. Unattached γ-TuRC molecules were washed with 1 vol of BRB80.

During the incubations, nucleation mix was prepared containing desired concentration of αβ-tubulin (3.5–21 μM) purified from bovine brain with 5% Cy5-labeled tubulin along with 1 mg/ml BSA (A7906, Sigma) in assay buffer, centrifuged for 12 min in TLA100 (Beckman Coulter) to remove aggregates, a final 0.68 mg/ml glucose oxidase (SERVA, catalog # SE22778), 0.16 mg/ml catalase (Sigma, catalog # SRE0041) was added, and reaction mixture was introduced into the flow chamber containing γ-TuRC.

## Total internal reflection fluorescence (TIRF) microscopy and analysis of microtubule nucleation from γ-TuRC

Nucleation of MTs was visualized with inverted Nikon TiE TIRF microscope using a 100X, 1.49 NA TIRF objective. An objective heater collar was attached (Bioptechs, model 150819–13) and the temperature set-point of 33.5°C was used for experiments. Time-lapse videos were recorded for 10 min at 0.5–1 frame per second using Andor iXon DU-897 camera with EM gain of 300 and exposure time of 50–200 ms each frame. Reference time-point zero (0 s) refers to when the reaction was incubated at 33.5°C on the microscope, and for most reactions, imaging was started within 30 s.

Growth speed of the plus-ends of MTs nucleated by γ-TuRC was measured by generating kymographs in ImageJ. For few specific datasets with notable in-plane drift, an ImageJ plugin, StackReg

(*Thévenaz et al., 1998*), was to correct a minor translational drift before proceeding with the analysis. Region of interest (ROI) for individual MTs were selected and resliced to generate a length-time plot and a line was fit to the growing MT plus-end. The slope of this line represents growth speed. The kinetics of MT nucleation from γ-TuRC was measured as follows. A kymograph was generated for every MT nucleated in the field of view. For most nucleation events, the time of nucleation of the MT was obtained from observing the kymograph and manually recording the initiation time point (see *Figure 1C* for examples). For MTs where nucleation occurred before the timelapse movie began or where the initiation was not clearly observed in the kymograph, the shortest length of the MT that was clearly visible in the timelapse was measured and measured average growth speed of MTs was used to estimate the time of nucleation. We verified that this procedure accurately estimates the nucleation time for test case MTs where the nucleation event was visible. The measurement of number of MTs ($N(t)$) nucleated versus time was generated from a manual log containing the nucleation time for all MTs observed in the field of view. To represent the theoretical field-to-field heterogeneity in the number of MTs nucleated, we assumed that binding of γ-TuRC and subsequent nucleation follows a Poisson distribution with mean $n$ MTs and standard deviation $\sqrt{n}$ MTs. 95% confidence interval in the nucleation measurements, $n \pm 2\sqrt{n}$ is displayed on each nucleation time course.

To calculate the percentage of γ-TuRCs that nucleate a MT, we visualized MT nucleation from Alexa-568 labeled γ-TuRC in the presence of 21 μM tubulin and 100 nM XMAP215, or with 10.5 μM tubulin. We counted the number of labeled γ-TuRC molecules attached in the field of view and counted the number of MTs nucleated specifically from these molecules but excluded spontaneous MT nucleation. For the reaction with 21 μM tubulin and 100 nM XMAP215, we directly measured that 15% of γ-TuRC molecules nucleated a MT. For the reaction with 10.5 μM tubulin, a similar calculation was performed and using measured curves (*Figure 2C*), we estimated the percentage of γ-TuRC that will nucleate with 21 μM tubulin as 11%.

## Power-law analysis of critical nucleus size on γ-TuRC

We consider the following simplified model to determine the number of αβ-tubulin dimers in the rate-limiting, transition state on γ-TuRC that is the critical nucleus. We consider a total number of γ-TuRC molecules $N_0$ available to nucleate MTs at a specific αβ-tubulin concentration $C$. The total number of MTs nucleated $N(t)$ from a total $N_0$ γ-TuRCs is a function of time $t$. If $n$ tubulin dimers assemble cooperatively on γ-TuRC for a successful MT nucleation, the rate of MT nucleation from γ-TuRC molecules available to nucleated at time $t$, $N_0 - N(t)$ reads,

$$\frac{dN(t)}{dt} = k_{nucleate}(N_0 - N(t))C^n \tag{1}$$

Here, we assume that tubulin does not get significantly depleted over time in the course of our reactions as shown by previous calculations (*Zanic, 2016*). At the start of the reaction $t = 0$, no MTs have nucleated $N(t = 0) = 0$, therefore at early times we assume $N_0 - N(t) \approx N_0$ to simplify the calculation of the critical MT nucleus,

$$\left.\frac{dN}{dt}\right|_{t \to 0} = k_{nucleate}N_0 C^n \tag{2}$$

Converting into log scale,

$$ln\left(\left.\frac{dN}{dt}\right|_{t \to 0}\right) = n\,ln(C) + a \tag{3}$$

To obtain the number of αβ-tubulin dimers in the critical nucleus on γ-TuRC, a straight line was fit to the initial, linear region of each nucleation curve $N(t)$ versus $t$ curve for every tubulin concentration $C$ and the rate of nucleation $\left.\frac{dN}{dt}\right|_{t \to 0}$ was obtained from slope of this fit. A straight line was then fit to $ln\left(\left.\frac{dN}{dt}\right|_{t \to 0}\right)$ versus $ln(C)$ for all concentrations, the slope of which provides the size of critical nucleus $n$. Finally, the measured rate of nucleation depends on the total number of γ-TuRC molecules available. As the total number of γ-TuRC molecules obtained from different days purifications changes,

the rate of nucleation from γ-TuRCs at 10.5µM tubulin was set to 1 (normalization factor) to allow pooling of all datasets for γ-TuRC-mediated nucleation.

## Spontaneous microtubule nucleation and data analysis

Spontaneous MT assembly was visualized similar to γ-TuRC-mediated nucleation with the following changes. The Pluronic, casein and NeutrAvidin incubations were performed identical to γ-TuRC nucleation assay but instead of attaching γ-TuRCs, sucrose-based buffer (of the same composition as used for γ-TuRC elution) was diluted 5-fold with BRB80, introduced in the flow chamber and incubated for 10 min. Washes were performed with 1 vol of BRB80, nucleation mix was added, and imaging was performed as described above. MTs nucleate spontaneously in solution fall down on the coverslip due to depletion forces during the 10 min of visualizing the reaction. The number of MTs nucleated in the field of view were counted manually and plotted in *Figure 3B*. 95% confidence interval is displayed assuming a Poisson distribution for theoretical field-to-field heterogeneity as described above.

In the absence of any attached nucleation site, the spontaneously nucleated MTs are usually not visualized from the time of their nucleation and the analysis used for γ-TuRC mediated nucleation was adapted. Integrating the *Equation (2)* above

$$N(t) = k_{nucleate} N_0 C^n t \tag{4}$$

Converting into log scale at time $t = \tau$,

$$ln(N(t = \tau)) = n\,ln(C) + b \tag{5}$$

To obtain the number of αβ-tubulin dimers in the critical nucleus in spontaneous assembly, the number of MTs at a specified time $t$ =7.5 min was measured, a straight line was then fit to $ln(N(t = \tau))$ versus $ln(C)$ for all concentrations, the slope of which provides the size of critical nucleus $n$. All datasets were pooled and reported.

## Preparation, microtubule assembly from blunt microtubule seeds and data analysis

Blunt MTs were prepared with GMPCPP nucleotide in two polymerization cycles as described recently (*Wieczorek et al., 2015*). Briefly, a 50 µl reaction mixture was prepared with 20 µM bovine brain tubulin with 5% Alexa-568 labeled tubulin and 5% biotin-labeled tubulin, 1 mM GMPCPP (Jena Bioscience) in BRB80 buffer, incubated on ice for 5 min, then incubated on 37°C for 30 min to polymerize MTs, and MTs were pelleted by centrifugation at 126,000 xg for 8 min at 30°C in TLA100 (Beckman Coulter). Supernatant was discarded, MTs were resuspended in 80% original volume of BRB80, incubated on ice for 20 min to depolymerize MTs, fresh GMPCPP was added to final 1 mM, incubated on ice for 5 min, a second cycle of polymerization was performed by incubating the mixture at 37°C for 30 min, and MTs were pelleted again by centrifugation. Supernatant was discarded and MTs were resuspended in 200 µl warm BRB80, flash frozen in liquid nitrogen in 5 µl aliquots, stored at −80°C and found to be stable for months. To verify that these MT seeds have blunt ends, frozen aliquots were quickly thawed at 37°C, diluted 20-fold with warm BRB80, and incubated at room temperature for 30 min to ensure blunt ends as described previously (*Wieczorek et al., 2015*). MTs were pipetted onto CF400-Cu grids (Electron Microscopy Sciences), incubated at room temperature for 60 s and then wicked away. 2% uranyl acetate was applied to the grids for 30 s, wicked away, and the grids were air-dried for 10 min. The grids were imaged using Phillips CM100 TEM microscope at 130,000 x magnification and most MT ends were found to be blunt.

To assay MT assembly from blunt MT seeds, MT assembly experiments similar to γ-TuRC nucleation assays were performed with the following variation. A lower concentration 0.05 mg/ml NeutrAvidin (A2666, ThermoFisher) was attached, and washes were performed with warm BRB80 prior to attaching MTs. One aliquot of MT seeds was thawed quickly, diluted to 100-fold with warm BRB80, incubated in the chamber for 5 minutes, unattached seeds were washed with 1 vol of warm BRB80, and the slide was incubated at room temperature for 30 min to ensure blunt MT ends. Wide bore pipette tips were used for handling MT seeds to minimize the shear forces that may result in breakage of MTs. Nucleation mix was prepared as described above and a low αβ-tubulin concentration (1.4-8.7 µM) was used. MT assembly from blunt seeds was observed immediately after incubating

the slide on the objective heater. Imaging and analysis were performed as described above for to γ-TuRC nucleation assays. The probability curves $p(t)$ for MT assembly were obtained by normalizing for the total number of seeds observed in the field of view $N(t)/N_0$, which allow for direct comparison across datasets. 95% confidence interval represents the theoretical variation in the number of MTs assembled from seeds across fields of view as described above. Rate of nucleation $\frac{dp}{dt}\big|_{t\to 0}$ was obtained as the slope of a straight line fit to the initial region of $p(t)$ versus $t$ curve for every tubulin concentration $C$. Power-law analysis was performed similar to γ-TuRC nucleation assays described above. However, as assembly from seeds occur near minimal tubulin concentration needed for polymerization of the plus end $C_*$, the governing equation reads,

$$\frac{dp(t)}{dt} = k_{nucleate}(1-p(t))C^{n-1}(C-C_*) \tag{6}$$

At the start of the reaction $t=0$, no MTs have nucleated $p(t=0)=0$, therefore at early times we assume $1-p(t)\approx 1$ to simplify the calculation of the critical MT nucleus. Converting equation (6) in log scale with these simplifications,

$$ln\left(\frac{dp}{dt}\Big|_{t\to 0}\right) = (n-1)ln(C) + ln(C-C_*) + a \tag{7}$$

Critical tubulin concentration for polymerization $C_*$ was obtained from the x-intercept of the growth speed curve ($C_* = 1.4\mu M$) as described previously. Finally, observing the total number of MT seeds for assembly allows for direct pooling of all datasets for MT assembly from seeds. From fitting a straight line between $ln\left(\frac{dp}{dt}\Big|_{t\to 0}\right)$ versus $ln(C-C_*)$ for all concentrations, we found the slope $n\approx 1$, which satisfies the above equation and provides the size of critical nucleus for MT assembly from seeds $n\approx 1$.

## Size exclusion chromatography of γ-tubulin and αβ-tubulin

Size exclusion chromatography of γ-tubulin and αβ-tubulin was performed as follows at 4°C. Purified, human γ-tubulin was diluted to 300 nM in γ-TB buffer (defined 50 mM K-MES pH6.6, 5 mM MgCl2, 1 mM EGTA, 10 mM thioglycerol, 10 μM GDP) with additional 250 mM KCl, and αβ-tubulin individually diluted to 20 μM or 70 μM with BRB80 buffer. Protein aggregates were pelleted by ultracentrifugation of the proteins individually at 80,000 rpm in TLA 100 (Beckman Coulter) for 15 min. γ-tubulin and αβ-tubulin were mixed in 1:1 vol ratio to achieve final concentrations 150 nM γ-tubulin to 10 μM or 35 μM αβ-tubulin and incubated on ice for 10 min. 500 μl of the mixture was loaded onto Superdex 200 Increase 10/300 column (GE Healthcare). The column was equilibrated with γ-TB buffer containing 90 mM KCl and chromatography was performed in this buffer. For control chromatography runs, equal volume of corresponding buffer was used. Absorbance at 214 nm was recorded. 0.3 ml fractions were collected and alternate fractions eluted between 8.5 ml and 16.6 ml were analyzed via immunoblot against γ-tubulin, αβ-tubulin and StrepII tag on γ-tubulin. Secondary antibody conjugated to 800 nm IRDye (LI-COR) was used and imaged with Odyssey CLx imaging station (LI-COR). High-molecular-weight gel filtration standards (Thyroglobulin, Aldolase and Ovalbumin) were purchased from GE Healthcare (Catalog #28403842) and used to estimate the Stokes' radii of eluted proteins in the same buffer as used for corresponding SEC run (*Le Maire et al., 1986*).

## Measurement of affinity between γ-tubulin and αβ-tubulin with single molecule microscopy

γ-TuRC nucleation assay was adapted as follows to measure the interaction affinity between γ-tubulin and αβ-tubulin. The imaging channel was prepared by sequentially with 5% w/v Pluronic F-127 incubation, casein buffer washes, 0.05 mg/ml NeutrAvidin incubation in casein buffer, BRB80 washes as described above. 100–200 nM of biotinylated αβ-tubulin or BRB80 buffer was introduced in the flow chamber and incubated for 5 min on a cold block, and unbound αβ-tubulin was washed with 50 μl of BRB80. During the incubations, binding mix was prepared with 100 nM of Alexa-568 or Alexa-488 labeled αβ-tubulin (24–50% labeling percentage) or with 100 nM of labelled γ-tubulin with identical fluorophore (7% labeling percentage) in 1x assay buffer, ultracentrifuged for 12 min in TLA100, oxygen scavengers were added, and reaction mixture was introduced into the flow chamber.

Single molecule binding of fluorescent γ-tubulin or αβ-tubulin with biotinylated αβ-tubulin was visualized with TIRF microscopy using the setup described above at 33.5°C. Images were collected at 2–5 fps with EMCCD gain of 300 and exposure time of 200 ms each frame, and data acquisition was started within 60–90 s after flowing fluorescent γ-tubulin or αβ-tubulin. Minimal photobleaching was observed for the first 15 s of time series acquired, which was used to extract the number by molecules bound by analyzing with the single molecule analysis software ThunderSTORM (*Ovesný et al., 2014*). Specifically, images were filtered with wavelet B-spline filter (scale 2–3 and order 3), molecules localized with 8-connected local maximum approach, threshold selected as the standard deviation of the first wavelet level, and suggested settings for sub-pixel localization by fitting an integrated Gaussian PSF model with maximum likelihood estimation was performed. The number of single molecules identified for each frame were recorded. The results from Thunder-STORM analysis were verified against manually identified molecules with a sample dataset. To obtain how many molecules bind to biotin-αβ-tubulin for every frame, the number of molecules of γ-tubulin or αβ-tubulin bound inspecifically to the coverslip were independently subtracted from the number of molecules bound to biotin-αβ-tubulin, and this value was divided by the known fluorescent labelling percentage. The calculated number of γ-tubulin or αβ-tubulin bound were averaged for the first 14 s (28 frames) for each dataset, and their mean and standard deviation was reported.

Interaction assays between αβ-tubulin and γ-tubulin were confirmed with biolayer interferometry using Octet RED96e (ForteBio) instrument in an eight-well plate. The plate temperature was held at 33°C and the protein samples were shaken at 400 rpm during the experiment. First, Streptavidin coated biosensors (ForteBio) were rinsed in interaction buffer (50 mM K-MES pH 6.6, 100 mM KCl, 5 mM MgCl$_2$, 1 mM EGTA, 0.05% Tween20, 1 mM GTP). 100–400 nM biotin-labeled αβ-tubulin, or blank buffer, was bound to Streptavidin sensor until loaded protein results in a wavelength shift (Δλ) of 3 nm. Unbound protein was removed by rinsing the sensor in interaction buffer, and interaction with αβ-tubulin was measured by incubating the sensor containing biotinylated αβ-tubulin with 0–35 μM unlabeled αβ-tubulin or 0–1 μM unlabeled γ-tubulin in interaction buffer for 5 min. Δλ (nm) was recorded as a measure of the amount of unlabeled αβ-tubulin that binds to the sensor.

## Nucleation of microtubules from purified γ-tubulin

MT assembly experiments from purified γ-tubulin was performed similar to γ-TuRC nucleation assays described above with following variation. No avidin was attached to the coverslips, and varying concentration of γ-tubulin was prepared by diluting purified γ-tubulin in a high salt buffer (50 mM K-MES pH 6.6, 500 mM KCl, 5 mM MgCl$_2$, 1 mM EGTA), centrifuging to remove aggregates separately for 12 min in TLA100 before adding to the nucleation mix containing 15 μM αβ-tubulin (5% Cy5-labeled) with BSA, glucose oxidase and catalase as described above to a final salt concentration of 44 mM KCl. The reaction mixture was introduced into the flow chamber and imaged via TIRF microscopy. A large number of MTs get nucleated immediately in the presence of 250 nM-1000 nM γ-tubulin.

## Negative stain electron microscopy of γ-tubulin filaments

Purified γ-tubulin was observed to form higher order oligomers previously using analytical gel filtration (*Thawani et al., 2018*). γ-tubulin filaments were prepared by diluting pure γ-tubulin to 1 μM to the buffer 50 mM K-MES pH 6.6, 5 mM MgCl$_2$, 1 mM EGTA, 100 mM KCl. 5 μl of γ-tubulin mixture was pipetted onto EM grids (Electron Microscopy Sciences, Catalog number: CF400-Cu), which were glow discharged for 25 s. 5 μl sample was incubated on the grid at room temperature for 60 s and wicked away with Whatman filter paper. Grids were washed with 5 μl of dH$_2$O 3 times, stained three times with 0.75% Uranyl formate, where the first two incubations were wicked away immediately while the last was incubated for 30 s. The grids were air-dried for 10 min. Data were collected on a Talos L120C TEM (FEI) equipped with a BM Ceta CCD camera, at a nominal magnification of 74,000x corresponding to a pixel size of 2.03 A°/pixel on the specimen with 1 s integration time, and a defocus range of 1–2 μm underfocus. Micrographs were acquired both in-plane with +0 degree tilt.

Micrographs were converted to mrc file format with IMOD package and imported into RELION-3.0.6 (*Punjani et al., 2017*) where the data analysis was performed. Contrast transfer function (CTF) estimation of 370 micrographs performed using Gctf (*Zhang, 2016*). Segments along the length of

thin filaments were picked manually. Filaments were boxed into helical segments with 50 Å rise, and subjected to two rounds of 2D classification and particle selection. 1001 particles were selected and were used to generate an ab-initio 3D model. One round of refinement using 3D auto-refine was performed with all particles, followed by one round of 3D classification. 659 particles from the most populated 3D class were selected and another round of refinement was performed to generate a final map with the solvent mask. Analysis was performed in UCSF Chimera (*Pettersen et al., 2004*). Longitudinal arrangement of αβ-tubulins (pink filament, *Figure 4—figure supplement 2E*) was generated by isolating one protofilament from PDB: 6DPU (*Zhang and Nogales, 2018*) and elongating the protofilament with the super-position function in Coot. Lateral arrangement of γ-tubulin array (blue filament, *Figure 4—figure supplement 2E*) was generated from the crystal contacts observed in the published P21 crystal array (PDB: 1Z5W [*Aldaz et al., 2005b*]), as described previously (*Aldaz et al., 2005a*). An alternate γ-tubulin arrangement was also generated by isolating the other possible filament from this P21 symmetry group, where neighboring γ-tubulins neither arrange linearly nor show lateral contacts (green filament, *Figure 4—figure supplement 2E*). Simultaneous docking of four copies each of longitudinal αβ-tubulin array, lateral γ-tubulin array, or alternate arrangement of γ-tubulin array, was performed by fitting each copy at 15 Å resolution in UCSF Chimera using the fitmap function. Lateral γ-tubulin arrays, but not other filament arrangements, display good fit where the γ-tubulin spacing closely matches that of the reconstructed filaments.

## Monte Carlo simulations of microtubule nucleation by γ-TuRC

### Simulation procedure

Kinetic Monte Carlo simulations for MT nucleation and assembly by γ-TuRC were coded and run in MATLAB and were based on a previous stochastic model for the plus-end dynamics of a MT (*VanBuren et al., 2002*). A type-B MT lattice geometry with 13-protofilaments and a pitch of 3 tubulins at the seam was assumed, and a similar γ-TuRC geometry was encoded. On the blunt plus-end geometry, αβ-tubulin dimers in the MT lattice may have no neighbors, one or half a neighbor at the seam. Once the MT growth occurs into a tapered one, αβ-tubulin dimers can also have one or two neighbors.

New αβ-tubulin dimers arrive with a constant on rate, $k_{on}$ ($M^{-1}s^{-1}$) on each protofilament. This on rate is equal for each protofilament on the plus-end or on γ-TuRC and remains constant during the simulation. An input concentration of αβ-tubulin dimers was assumed to be constant and not be depleted as shown by previous calculations (*Zanic, 2016*). Therefore, the net on-rate at each time step is, $k_{on}C$ ($s^{-1}$), where $C$ is the concentration of αβ-tubulin dimers. The interactions between αβ-tubulins was assumed to occur with longitudinal and lateral bond energies, $G_{Long,\alpha\beta-\alpha\beta}$ and $G_{Lat,\alpha\beta-\alpha\beta}$, respectively. All αβ-tubulin dimers recruited to the MT lattice or γ-TuRC have a longitudinal bond, and the lateral bond energy depends on the arrangement of neighboring αβ-tubulin dimers. The longitudinal bond energy between γ-/αβ-tubulin on γ-TuRC is $G_{Long,\gamma-\alpha\beta}$. As a result, the dissociation rate (off-rate) of individual tubulin dimers from the lattice differs and is a function of total bond energy $G_{tot}$. $G_{tot}$ is a sum of the longitudinal bond energy, $G_{Long,\alpha\beta-\alpha\beta}$ or $G_{Long,\gamma-\alpha\beta}$, plus the total lateral bond energy from all the neighbors, $m \times G_{Lat,\alpha\beta-\alpha\beta}$. Based on previous works, we also posit that when a tubulin dimer dissociates, all dimers above it in the protofilament dissociate as well. The off-rate of each dimer was then calculated from the following equation as derived previously (*VanBuren et al., 2002*),

$$lnK = ln\left(\frac{k_{on}}{k_{off}\ (s^{-1})}\right) = -\frac{G_{tot}}{k_BT} \tag{8}$$

An open conformation of native γ-TuRC was assumed as observed in recent cryo-EM structures (*Liu et al., 2020*; *Wieczorek et al., 2020*; *Consolati et al., 2020*). The αβ-tubulins assembled on neighboring sites do not form lateral interactions in the open conformation. A possible transition to a closed γ-TuRC state was allowed with a thermodynamic penalty of $\Delta G_{\gamma TuRC-conf}$. However, if $n$ lateral bonds form upon this transition from αβ-tubulins assembled on neighboring sites, that net energy for an open-to-closed transition is $\Delta G_{close} = \Delta G_{\gamma TuRC-conf} - n\Delta G_{Lat,\alpha\beta-\alpha\beta}$. At each time step in the simulation, the rate of this transition is calculated as, $k_{\gamma TuRC-conf} \times \exp\left(\frac{-G_{close}}{k_BT}\right)$, where

$k_{\gamma TuRC-conf}$ $(s^{-1})$ is the pre-factor of the Arrhenius equation. Hydrolysis of incorporated tubulin dimers was ignored because few catastrophe events were observed in our experiments.

To execute the stochastic simulations, we formulate a list of possible events at every time step, including association of a αβ-tubulin dimer, dissociation of a αβ-tubulin dimer, or transition of γ-TuRC to closed state. The forward rate of each event is calculated as described above. A uniform random number $(R_i)$ from 0 to 1 is generated for each possible event in the list and a single realization of the exponentially distributed time is obtained for each event,

$$t_i = \frac{-\ln(R_i)}{k_i \ (s^{-1})} \tag{9}$$

The event with the shortest execution time is implemented and time elapsed during the simulation is advanced by $t_i$ seconds. Each simulation was run with a maximum defined time, usually between 100 and 500 s, or were stopped once the MT grew a total of 2-5 µm in length. The MATLAB code for simulations is provided in the Supplementary Materials.

## Parameter estimation

MT growth parameters were determined by fitting to experimental growth speed curves. Briefly, 20 simulations were performed for each concentration from 2 to 20 µM tubulin for 100 s each. MT length versus time was plotted. Growth speed was obtained from the slope of a linear curve fit of the polymerizing stretch of the length versus time plot. Parameter values of $k_{on} = 1.3 \times 10^6 \ (\mu M^{-1} s^{-1})$, $\Delta G_{Long, \alpha\beta-\alpha\beta} = -7.2 k_B T$, $\Delta G_{Lat, \alpha\beta-\alpha\beta} = -6.5 k_B T$ resulted in the best fit for all tubulin concentrations. These parameter values are similar to those obtained in previous reports (*Mickolajczyk et al., 2019*; *VanBuren et al., 2002*). With these polymerization parameters fixed, we varied the remaining parameters. $\Delta G_{Long, \gamma-\alpha\beta}$ was varied from $(0.7 - 1.3 \times \Delta G_{Long, \alpha\beta-\alpha\beta}$. $G_{\gamma TuRC-conf}$ was varied from $+(0 - 30)k_B T$ and $k_{\gamma TuRC-conf}$ from $(1 - 0.001) \ s^{-1}$. For each parameter set, we performed 200-500 simulations each at specific tubulin concentrations between 2 and 50 µM. For each simulation, the time of γ-TuRC ring closure was recorded as the nucleation time as it represents the transition from zero MT length to a continuously growing MT. For the simulation where no MT nucleation occurred, a nucleation time of infinity was recorded. Cumulative probability distribution of nucleation $(p(t))$ versus time was generated from the log of nucleation times for each tubulin concentration. Rate of nucleation $\frac{dp}{dt}\big|_{t \to 0}$ was obtained by a linear fit from the initial part of each nucleation fit, as described above. The slope of a straight line was fit to $ln\left(\frac{dp}{dt}\big|_{t \to 0}\right)$ versus $ln(C)$, as outlined in *Equation 2,3* above, provide the size of critical nucleus $n$. The nucleation curves and power-law analysis was compared with experimental data for γ-TuRC-mediated nucleation. The best agreement was found with $\Delta G_{Long, \gamma-\alpha\beta} = 1.1 \times \Delta G_{Long, \alpha\beta-\alpha\beta}$, as supported by our biochemical measurements, $\Delta G_{\gamma TuRC-conf} = 10 k_B T$ and $k_{\gamma TuRC-conf}$ = 0.01 $s^{-1}$.

To analyze the arrangement of αβ-tubulins in the transition state, the state of γ-TuRC with αβ-tubulin dimers was recorded at the time of γ-TuRC ring closure for 2119 simulations. 3D-dimensional probability distribution of total number of αβ-tubulin dimers and number of lateral αβ-tubulin bonds was generated. The arrangement of αβ-tubulin dimers in the most frequently occurring transition states were displayed with schematics.

To capture the dynamics of MT assembly from blunt seeds, we simulated nucleation assuming a closed γ-TuRC geometry as follows. Lateral bonds between αβ-tubulins assembled on the neighboring sites on γ-TuRC were allowed and $G_{Long, \gamma-\alpha\beta}$ was set equal to $G_{Long, \alpha\beta-\alpha\beta}$. Simulations were performed as described above with the following change. The time when the MT in each simulation grew to 50 nm length was recorded to generate the probability distribution. Nucleation curves and power-law analysis was compared with experimental data for seed-mediated MT assembly.

## Measuring the effect of microtubule associated proteins on γ-TuRC-mediated nucleation

Effect of microtubule associated proteins (MAPs) was measured on γ-TuRC's nucleation activity. γ-TuRC was attached on the coverslips using the setup described above and a control experiment was performed with identical reaction conditions for each protein tested. Because CDK5RAP2's γ-TuNA

motif and NME7 bind γ-TuRC, to test their activity γ-TuRC was additionally incubated 6 µM γ-TuNA motif or 6 µM NME7 to γ-TuRC for 5 min prior to attachment to coverslips to maximize their likelihood of binding and the control γ-TuRC reaction was treated identically with the storage buffer for each protein. Nucleation mix was then prepared containing 10.5 µM αβ-tubulin concentration (5% Cy5-labeled tubulin) as specified along with 1 mg/ml BSA and oxygen scavengers, and either buffer (control), 10 nM GFP-TPX2, 3 µM γ-TuNA motif from CDK5RAP2, 6 µM NME7, 5 µM Stathmin or 10 nM MCAK was added. To test NME7 or MCAK's effect, the assay buffer additionally contained 1 mM ATP. The reaction mixture containing tubulin and MAP at specified concentration was introduced into the flow chamber containing γ-TuRC, and MT nucleation was visualized by imaging the Cy5-fluorescent channel at 0.5–1 frames per second. For TPX2, fluorescence intensity of the protein was simultaneously acquired.

The number of MTs nucleated over time was measured as described above and the effect of protein on γ-TuRC's nucleation activity was assessed by comparing nucleation curves with and without the MAP. In order to normalize for the total number of γ-TuRC molecules obtained from different purifications and enable pooling results from all datasets, the number of MTs nucleated at a specified time point, mentioned in each figure legend, was set to 1 for γ-TuRC only (no MAP) control reactions. As before the shaded region represents 95% confidence interval $(n \pm 2\sqrt{n})$ in the number of MTs, $n$ assuming a Poisson distribution that determines binding and subsequent nucleation from γ-TuRCs and was calculated and displayed on each nucleation time-course.

## Cooperative microtubule nucleation assay with purified XMAP215 and γ-TuRC

A similar set of experiments as above to characterize the effect of MAPs was performed to study the effect of XMAP215 on γ-TuRC-mediated nucleation with the single molecule assays with the following differences. 20 nM of XMAP215-GFP-7xHis was added to nucleation mix prepared with 3.5–7 µM αβ-tubulin concentration (5% Cy5-label) in XMAP assay buffer (80 mM K-PIPES, 1 mM MgCl$_2$, 1 mM EGTA, 30 mM KCl, 0.075% w/v methylcellulose 4000 cp, 1% w/v D-(+)-glucose, 0.007% w/v Brij-35, 5 mM BME, 1 mM GTP). MTs nucleated from attached γ-TuRC with and without XMAP215 were measured to assess the efficiency of nucleation induced by XMAP215. To assess if N- or C-terminal domains of XMAP215 increases nucleation efficiency, wild-type XMAP215 was replaced with a C-terminal construct of XMAP215 (TOG5-Cterminus-GFP) or an N-terminal construct (TOGs1-4-GFP) in the described experiment.

To measure the kinetics of cooperative nucleation XMAP215 and γ-TuRC, a constant density of γ-TuRC was attached as described above and nucleation mix nucleation mix was prepared with a range of αβ-tubulin concentration between 1.6 and 7 µM (5% Cy5-label) with 20–25 nM of XMAP215-GFP-7xHis or XMAP215-TEV-GFP-7xHis-StrepII in XMAP assay buffer, introduced into reaction chamber and MT nucleation was imaged immediately by capturing dual color images of XMAP215 and tubulin intensity at 0.5 frames per second.

Data analysis was performed as above for γ-TuRC mediated nucleation, theoretical field-to-field heterogeneity in the number of MTs nucleated was represented with a Poisson distribution as before and 95% confidence interval. Critical tubulin nucleus for cooperative nucleation from XMAP215 and γ-TuRC was obtained as described for γ-TuRC alone (*Equations 1, 2, 3*). A straight line was fit to log rate of nucleation $ln\left(\frac{dN}{dt}\Big|_{t\to 0}\right)$ versus log tubulin concentration $ln(C)$ and its slope provides the size of critical nucleus $n$. Finally, to normalize for the total number of γ-TuRC molecules obtained from different purifications, the rate of cooperative nucleation from XMAP215 and γ-TuRC at 3.5µM tubulin was set to 1. All datasets were pooled and reported.

## Triple-color imaging of XMAP215, γ-TuRC and microtubules

For triple-color fluorescence assays, Alexa-568 and biotin-conjugated γ-TuRC was first attached to coverslips as described above with the following variation: 0.05 mg/ml of NeutrAvidin was used for attaching γ-TuRC. Nucleation mix was prepared with 7 µM αβ-tubulin (5% Cy5-label), 10 nM Alexa-488 XMAP215-SNAP or XMAP215-GFP with BSA and oxygen scavengers in XMAP assay buffer (80 mM K-PIPES, 1 mM MgCl$_2$, 1 mM EGTA, 30 mM KCl, 0.075% w/v methylcellulose 4000 cp, 1% w/v D-(+)-glucose, 0.007% w/v Brij-35, 5 mM BME, 1 mM GTP) and introduced into the reaction chamber

containing attached γ-TuRC. Three-color imaging per frame was performed with sequential 488, 568 and 647 nm excitation and images were acquired with EMCCD camera at 0.3 frames per second.

## Acknowledgements

We thank Brian Mahon, Brianna Romer and Sophie Travis for advice on collecting and processing of electron microscopy data, as well as Petry lab members for discussions. We thank David Agard and Michelle Moritz for sharing unpublished data and for discussions. This work was supported by an American Heart Association predoctoral fellowship 17PRE33660328 and a Princeton University Honorific Fellowship (both to AT), a Howard Hughes Medical Institute Gilliam fellowship and a National Science Foundation graduate research fellowship (both to MJR), NIGMS R00GM112982 (to GB), NIH New Innovator Award 1DP2GM123493, Pew Scholars Program in the Biomedical Sciences 00027340, David and Lucile Packard Foundation 2014–40376 (all to SP), and the Center for the Physics of Biological Function sponsored by the National Science Foundation grant PHY-1734030.

## Additional information

### Funding

| Funder | Grant reference number | Author |
| --- | --- | --- |
| American Heart Association | 17PRE33660328 | Akanksha Thawani |
| Princeton University | Charlotte Elizabeth Procter Honorific Fellowship | Akanksha Thawani |
| Howard Hughes Medical Institute | Gilliam fellowship | Michael J Rale |
| National Science Foundation | Graduate Student Fellowship | Michael J Rale |
| National Institute of General Medical Sciences | R00GM112982 | Gira Bhabha |
| National Institute of General Medical Sciences | 1DP2GM123493 | Sabine Petry |
| Pew Charitable Trusts | 00027340 | Sabine Petry |
| David and Lucile Packard Foundation | 2014-40376 | Sabine Petry |
| National Science Foundation | PHY-1734030 | Joshua W Shaevitz |

The funders had no role in study design, data collection and interpretation, or the decision to submit the work for publication.

### Author contributions

Akanksha Thawani, Conceptualization, Software, Validation, Investigation, Visualization, Methodology, Writing - original draft, Writing - review and editing; Michael J Rale, Investigation, Purified NME7 and CDK5RAP2 fragments and contributed to the related experiments; Nicolas Coudray, Gira Bhabha, Methodology, Provided advice on EM reconstruction of αβ-tubulin filaments; Howard A Stone, Joshua W Shaevitz, Supervision, Writing - review and editing; Sabine Petry, Conceptualization, Supervision, Funding acquisition, Project administration, Writing - review and editing

### Author ORCIDs

Akanksha Thawani https://orcid.org/0000-0003-4168-128X
Howard A Stone http://orcid.org/0000-0002-9670-0639
Joshua W Shaevitz http://orcid.org/0000-0001-8809-4723
Sabine Petry https://orcid.org/0000-0002-8537-9763

## Ethics

Animal experimentation: This study was performed in strict accordance with the recommendations in the Guide for the Care and Use of Laboratory Animals of the National Institutes of Health. All of the animals were handled according to approved Institutional Animal Care and Use Committee (IACUC) protocol # 1941-16 of Princeton University.

## Decision letter and Author response

Decision letter https://doi.org/10.7554/eLife.54253.sa1
Author response https://doi.org/10.7554/eLife.54253.sa2

## Additional files

### Supplementary files

- Source code 1. MATLAB code for simulations of γ-TuRC-mediated microtubule nucleation.

- Transparent reporting form

### Data availability

All data generated or analysed during this study are included in the manuscript and supporting files. Source data files have been provided for Figures 2, 3, 4, 6, 7 and related supplements.

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
