## [Decision Letter]

**Acceptance summary:**

This study establishes TIRF-based single molecule imaging of microtubule nucleation from immobilized, purified γ-TuRCs, which function as nucleation templates. The authors provide first insight into the molecular events that occur at the γ-TuRC during nucleation and establish the critical size that the α-β-tubulin nucleus requires to be converted into a growing microtubule.

**Decision letter after peer review:**

[Editors’ note: the authors submitted for reconsideration following the decision after peer review. What follows is the decision letter after the first round of review.]

Thank you for submitting your work entitled "Molecular mechanism of microtubule nucleation from γ-tubulin ring complex" for consideration by *eLife*. Your article has been reviewed by three peer reviewers, including Jens Lüders as the Reviewing Editor and Reviewer #3, and the evaluation has been overseen by a Senior Editor.

Our decision has been reached after consultation between the reviewers. Based on these discussions and the individual reviews below, we regret to inform you that your work will not be considered further for publication in *eLife*.

Establishing an in vitro assay that allows to study microtubule nucleation events from purified gTuRC at the single molecule level is overdue. Such an assay is required to answer fundamental and long-standing questions regarding the mechanism of nucleation. In this regard the assay using immobilized gTuRC and single molecule TIRF imaging, as presented in the current manuscript, is an important achievement. Unfortunately, it has not yet been used to its full potential. Several interesting observations are made, but the chosen path and derived conclusions in this manuscript are challenged by three recently published Cryo-EM structures of native gTuRC, including gTuRC purified from *Xenopus* egg extract as in the present study (Liu et al., 2019). The structures show that native gTuRC does not match the symmetry of a microtubule. It is in an 'inactive' conformation, one that is not well suited to promote lateral tubulin interactions, questioning the validity of the presented model. Considering this and other issues raised by the reviewers the manuscript does not provide the conceptual advance required by *eLife*.

Reviewer #1:

The authors use γ-TuRCs purified from *Xenopus* egg extracts in combination with single molecule microtubule nucleation assays to gain insight into the process of microtubule nucleation. The key findings are that γ-TuRCs enhance microtubule nucleation by promoting the lateral association of a/β-tubulin molecules and that nucleation from γ-TuRCs is further promoted by XMAP-215. The experiments are well-executed and the data is nicely presented (although I have some questions about N numbers – see below). Overall, I think there are some interesting insights into γ-TuRC-mediated nucleation and the experiments are at the cutting edge, but I feel that we do not learn a great deal more about the role of γ-TuRC and XMAP-215 in microtubule nucleation to justify publication within a broad spectrum high-impact journal such as *eLife*. Several results or notions are already known e.g. γ-TuRCs and XMAP-215 promote microtubule nucleation, a higher concentration of tubulin is required for nucleation than for microtubule elongation, γ-tubulin forms filaments, XMAP-215 promotes nucleation, XMAP-215 associates with γ-TuRCs. Moreover, the main message, that γ-TuRCs promote lateral association of tubulin dimers, is widely assumed based on γ-TuRC structure and because longitudinal a/β-tubulin contacts are believed to be stronger than lateral contacts. As the authors themselves state, "the insights on MT nucleation by γ-TuRC and XMAP215 provide an essential basis to build upon". I therefore feel that, without further advances, such as how γ-TuRCs can be activated to better promote microtubule nucleation, the paper would be better suited to a more specialised journal, such as JBC.

More specific comments:

1) The title should be more specific to the work done, as the current title indicates that the authors have solved the entire mechanism of nucleation.

2) Abstract: "the underlying mechanism largely remains a mystery". This is a difficult statement to swallow. I would argue that the underlying mechanism (templating the addition of tubulin dimers into a tube-like structure) has been widely assumed (although not directly proved) for a long time, in particular after the 2010 Nature paper from David Agard's group that showed the structure of the yeast "γ-TuRC". This was good proof that the template model was correct, rather than the protofilament model. I think the authors should therefore modify their statement to something like: "While it has been assumed that…., here we provide direct evidence for….".

) Abstract: "…we uncover that γ-TuRC nucleates a MT more efficiently than spontaneous assembly". This was already known – many studies have shown (using more standard microtubule nucleation assays) that microtubules form more readily in the presence of γ-TuRC.

4) Figure 1B: Three γ-TuRC nucleation sites are indicated by arrows – is this all of the γ-TuRCs present within this field (Can the authors show a two-colour image overlay)? If there were only three γ-TuRCs, it would not fit with the finding that spontaneous nucleation does not occur at 10.5μM tubulin….

5) Subsection “Defining the microtubule nucleus on γ-TuRC”: "Surprisingly, γ-TuRC nucleated MTs starting from 7 μM tubulin (Figure 1D), which is higher than the minimum tubulin concentration (C*) needed for growth at pre-formed MT plus-ends (C* = 1.4μM, Figure 1E)". I don't think this is surprising, given the results of Wieczorek et al., 2015 – these authors used solution exchange to show that, while nucleation from a microtubule seed required ~6μM tubulin, microtubules could continue to grow at ~1μM tubulin. This statement should be toned down and the authors should refer to Wieczorek et al. results.

6) Figure 1G: The authors use power-law calculations to calculate that ~3.7 tubulin dimers are required for nucleation. It is not clear exactly how they did this. Given that this power analysis underlies the main finding of the paper, I think it deserves a full explanation.

7) Subsection “Does γ-TuRC nucleate a microtubule via the blunt plus-end model?”: The authors use microtubule seeds to compare their nucleation efficiency to γ-TuRCs and find that microtubules can assemble from these seeds at only 2.45μM tubulin. This is significantly lower than that found by Wieczorek et al., 2015 (6μM tubulin) – can the authors comment on this and suggest an explanation?

8) Subsection “Molecular insight into microtubule nucleation by γ-TuRC”: Experiments to nucleate microtubules from γ-tubulin oligomers: I don't think this adds much to the paper. It is already known that γ-tubulin can form filaments and it is hard to understand how these filaments would nucleate microtubules, given that the filaments are presumably made of laterally contacting γ-tubulin molecules, thus creating a linear, rather than circular, template for the addition of a/β-tubulin dimers. It was also not clear from the methods what the final salt concentration was during the nucleation reactions, given that the g-tubulin was diluted in high salt buffer before being added to the reaction buffer. This is important as the authors show that tubulin oligomers do not form at high salt concentrations. If the authors want to keep these experiments in the paper, they need to cite work from the groups of Alvarado-Kristensson, Binarova and Monasterio who have shown that γ-tubulin can oligomerise into filaments.

9) Subsection “Molecular insight into microtubule nucleation by γ-TuRC”: The stochastic models, used to predict how many γ-tubulin sites must to be occupied to have the best chance of 2 adjacent tubulin dimers being present, are based on a perfectly circular arrangement of γ-tubulin molecules (Figure 2H). Liu et al., 2019 have just shown the cryo-EM structure of the *Xenopus* γ-TuRC. It is in an open confirmation, where the first and last γ-tubulin molecules are not adjacent to each other. I presume this would change (albeit perhaps only slightly) the calculations from the stochastic model simulations, so the authors should repeat them using an open ring where there is no lateral contact if positions 1 and 13 are occupied.

10) For quantifying the numbers of microtubules nucleated in e.g. Figure 1F, Figure 3A,3C it is not clear how many times the experiments were performed and whether the graphs represent mean or median values. Given the shakiness of the lines in the graph in Figure 3A, the N numbers seem low. I would expect that the experiments (being quick to perform) would have been performed several times (perhaps 5 to 10 times) and that a mean or median value was used to represent microtubule numbers at each timepoint. Can the authors confirm this, or do more repeats?

11) Figure 3A: the authors show that TPX2 has no role in microtubule nucleation from γ-TuRCs, but previous studies (Wieczorek et al., 2015, Roostalu et al., 2015, Woodruff et al., 2017) have shown that TPX2 does promote nucleation. Do the authors believe there is a difference in the requirement of TPX2 when considering γ-TuRCs versus microtubule seeds as templates? Can the authors repeat previous results showing that TPX2 promotes nucleation from microtubule seeds, as a positive control to show that their purified TPX2 is functional?

12) Subsection “How do γ-TuRC and XMAP215 synergistically nucleate microtubules?”: "XMAP215 effectively decreases the minimal tubulin concentration necessary for MT nucleation from γ-TuRC to 1.6 μM". How was this calculated? Also, does adding XMAP215 also increase the probability of spontaneous nucleation? In a model where XMAP215 binds γ-tubulin and uses TOG domains to add tubulin dimers for the initial nucleation event, the addition of XMAP215 should not increase the probability of spontaneous nucleation.

13) Figure S4B: the authors test whether the C-term γ-tubulin-interacting region of XMAP215 promotes microtubule nucleation, and show that it does not. It would be informative to also test whether the rest of XMAP-215 alone (i.e. XMAP-215 minus the γ-tubulin binding region) can promote microtubule nucleation from a γ-TuRC. Presumably not, but this truncated form of XMAP-215 should in theory be able to bind to microtubule plus ends via the TOG domains. It would therefore be a nice experiment to show that it is not plus-end bound XMAP-215 that is promoting nucleation, but it is the XMAP-215 that binds to the γ-TuRC.

Reviewer #2:

The present manuscript uses purified γ-TuRC, tubulin, and different MAPs, to reconstitute microtubule nucleation in vitro, and to quantify nucleation rates, growth speed, and cooperativity of tubulin dimers. Using coated coverslips to anchor tagged γ-TuRCs, in combination with TIRF microscopy, several important observations are made:

- Nucleation of microtubules occurs from γ-TuRCs de novo, and this is more efficient than spontaneous assembly.

- The critical nucleus for nucleation from γ-TuRCs comprises only 3-4 tubulin dimers, which is less than half the size required for spontaneous assembly.

- γ-TuRC-mediated nucleation differs from plus-end growth at blunt microtubule ends, where a single tubulin dimer suffices as a nucleus.

- γ-TuRC-mediated nucleation is not influenced by TPX2. By contrast, XMAP215 decreased the minimal tubulin concentration and increased the nucleation rate, by binding first to the γ-TuRC, yet requiring a similar critical nucleus of 3-4 tubulin dimers.

Altogether, these data provide extremely valuable insight into the mechanisms of microtubule nucleation, since they clarify multiple major issues: γ-TuRCs act directly as templates for microtubule nucleation, ruling out other hypotheses that propose spontaneous nucleation, followed by capping of the free minus-ends by γ-TuRCs. Most importantly, we have for the first time a concrete idea about the critical size of the tubulin nucleus, and about the involvement (or non-involvement) of XMAP215 and TPX2 in the nucleation process.

The manuscript is timely, since several other studies on the γ-TuRC (e.g. structure of the native full complex) are likely to be published by other groups in the very near future.

Reviewer #3:

The manuscript by Thawani et al., together with a recently posted study by the Surrey group on the bioRxiv preprint server, is the first to study microtubule nucleation from purified, immobilized gTuRC at the single molecule level by TIRF microscopy. Using this approach the authors investigate and compare nucleation from gTuRC with spontaneous MT assembly and nucleation from preformed MT seeds. They find that nucleation from gTuRC is more efficient than spontaneous assembly, but less efficient than nucleation from MT seeds. They argue that this is due to relatively weak longitudinal affinity between g-tubulin and ab-tubulin based on measurements of ab-tubulin interaction with purified g-tubulin and ab-tubulin in biolayer interferometry assays. XMAP215 significantly increases gTuRC nucleation activity. They estimate that nucleation from gTuRC alone requires 3-4 ab-tubulin dimers to bind to gTuRC before the kinetic nucleation barrier is overcome and argue that this is facilitated by the lateral array of g-tubulins in the gTuRC.

The establishment of TIRF imaging of nucleation from gTuRC at single molecule level is overdue and the presented work is a great achievement in this regard, providing interesting new ideas and experimental possibilities. However, the authors base their conclusions on various assumptions that cannot be made without further evidence and control experiments. A major issue is the fact that native gTuRC is not a perfect template that matches the symmetry of a microtubule, as revealed by three independent recent Cryo-EM studies by the Kapoor, Schiebel and Surrey labs. Overall, the study requires significant additional work that in my opinion would go beyond what would be reasonable for a revision in *eLife*.

Major points:

1) Subsection “Reconstituting and visualizing microtubule nucleation from γ-TuRC”: "MT nucleation events occurred specifically from single gTuRC molecules" – gTuRCs are not labelled in this assay, how do the authors get to this conclusion? How can they exclude that some of the MTs are from spontaneous nucleation?

2) The main conclusions of this study regarding the nucleation mechanism are based on the assumption that gTuRC is a perfect template, which has recently been disproven. Cryo-EM data of purified native human and *Xenopus* gTuRC clearly show that the g-tubulin array in gTuRC does not match the symmetry of a MT, by displaying lateral gaps between g-tubulin molecules, in particular in the part of gTuRC that contains GCP4, 5, and 6. This would be expected to impact its ability to promote lateral interactions between ab-tubulin dimers.

3) Figure 2C-E: It is assumed that γ-tubulin alone, through lateral interaction, forms a filament-like microtubule nucleation template. There are two issues here. First, there is no evidence that the g-tubulin filaments are indeed formed by lateral g-tubulin interactions. In fact, the EM image in S2C shows a fiber much thicker than expected. Second, the authors should present evidence that the formed ab-tubulin polymers are actually microtubules rather than other types of filaments (e.g. protofilaments or polymerization along g-tubulin filaments).

4) Fig2D: Are these really protofilaments? They seem much thicker. Also, can the authors exclude that g-tubulin polymerizes during coating of the chip?

5) Figure 2H: First, the simulation (i) assumes that gTuRC is perfectly ring-shaped (but it is not, see point 2 above) and ii) does not take into account binding affinity with tubulin dimers already bound to the template. After binding of the first dimer, binding of a second dimer at the neighbouring position may be promoted by lateral affinity. Also, I don't understand how the probability in calculated – should it not increase further after 4 sites are occupied? At this point it should be much more likely to have lateral contacts than not.

6) Considering the technical challenges with experimental repetitions and use of different, freshly prepared gTuRC preps for each repetition, it would be useful to repeat the nucleation assay to measure spontaneous and nucleation from seeds and from gTuRC side-by-side to properly compare the different conditions. It should be discussed that the minimal concentration for nucleation from seeds in a previous study by Wieczorek et al., (2015) was quite different and similar to nucleation from gTuRC in the present study.

7) It is not very clear how many times experiments were repeated and what the exact outcomes of each were. Sometimes data from different repetitions was "pooled", sometimes only one "representative dataset" of several repetitions is shown. Would it not be more appropriate to plot mean values from the multiple repetitions? The descriptions in all three figure legends are quite vague, for example Figure 2: "…were repeated at least twice with multiple supporting results…" – what exactly does this mean?

The same is true for the transparent reporting form:.…"confirmation and supporting experiments were performed with slightly different conditions and complete agreement was found amongst all measurements. The datasets were sufficient to give confidence on the measured value of interest…". What are the slightly different conditions? How are "complete agreement" and "confidence" determined?

Source data is not provided because "all data generated in this study was pooled (no data was left out)", but if only one "representative dataset" is shown, my understanding is that some data was in fact left out.

8) What is the percentage of gTuRCs that nucleate microtubules in the TIRF assay? This number should be provided.

[Editors’ note: further revisions were suggested prior to acceptance, as described below.]

Thank you for submitting your article "The transition state and regulation of microtubule nucleation from γ-TuRC revealed by single molecule microscopy" for consideration by *eLife*. The evaluation of your paper has been overseen by a Reviewing Editor and Anna Akhmanova as the Senior Editor.

The reviewers have discussed the reviews with one another and the Reviewing Editor has drafted this decision to help you prepare a revised submission.

Summary:

The assembly and organization of the microtubule cytoskeleton requires nucleation of microtubules by the nucleator γ-TuRC, but the underlying mechanism has remained obscure. Recent Cryo-EM structures of γ-TuRC have raised additional questions by showing that native γ-TuRC is in a conformation that is not well suited to template microtubule nucleation. This study provides first insight by establishing TIRF imaging from purified, immobilized γ-TuRCs in vitro. Using a combination of biochemistry and modeling the study explains how an "imperfect" nucleation template can function as nucleation platform and identifies several crucial determinants of this activity. Rather than activation of γ-TuRC by a conformational switch prior to nucleation, the authors propose that the conformational switch occurs as a result of nucleation.

The authors have provided very thorough revisions, presenting and including a substantial amount of new data. The manuscript has been significantly improved and despite the lack of structural support data by EM in the manuscript, the main conclusions are well supported and we consider it now suitable for publication in *eLife*. There are, however, a few remaining issues that we would like the authors to address.

Essential revisions:

1) We appreciate that you have modified the title, but it now seems to not make complete sense. We would suggest "The transition state and regulation of γ-TuRC-mediated microtubule nucleation revealed by single molecule microscopy".

2) "Notably, the distinct minimal tubulin concentration needed for seeds to assemble a MT as compared to a previous report 24 results from the differences in assay conditions." The authors should be clearer here, explaining in more detail (as they did in the response to reviewers) that they got the same result as previous authors when they used their particular conditions.

3) "In summary, because γ-TuRC positions an array of γ-tubulins at its nucleation interface that are thought to stabilize intrinsically weak, lateral αβ/αβ-tubulin interaction 9,10,13,14,17,20,22147, MT nucleation by γ-TuRC has been proposed to function similar to polymerization of a MT end. Here we show several lines of evidence that γ-TuRC-mediated nucleation has distinct characteristics from MT polymerization and assembly from blunt MT seeds. While growth speed of MTs nucleated from γ-TuRC or templated from MT seeds is similar (Figure 3—figure supplement 1D), γ-TuRC molecules do not nucleate MTs at low tubulin concentration where MT polymerization can occur." It is important to point out here that this relates specifically to the purified γTuRCs that nucleate microtubules in the absence of other cellular factors. While the authors do show that some of the proposed activators do not activate the γ-TuRC in their assays, they cannot yet rule out that other proteins do function in vivo to promote the correct configuration of the γTuRC. They mention this in the Discussion section, but they are making the statement in the Results section, and it is a bit misleading to group all γTuRCs into this poor-nucleation bracket.

4) Figure 6B, D – the authors state that there is no significant difference in nucleation rate when they add γTuNA or TPX2, but the graphs do show differences and no statistical test is provided.

5) "Recently, XMAP215 was discovered to be a nucleation factor that synergizes with γ-TuRC". There is a new pre-print from Trisha Davis' group showing that nucleation by γTuRC and XMAP215 is additive, not synergistic http://biorxiv.org/content/early/2020/05/23/2020.05.21.109561. It would be useful for the reader if the authors would discuss how this compares to their work here.

6) The authors should discuss in more detail recently published work by the Surrey group (2020), which also shows, as the current manuscript, TIRF imaging of nucleation from immobilized γ-TuRCs. There are some differences e.g. regarding the critical number of dimers that need to assemble on gTuRC and regarding overall gTuRC activity in the assay. It would be very useful for the reader to know how the observations/conclusions in the two studies agree or differ.

---

## [Author Response]

[Editors’ note: The authors appealed the original decision. What follows is the authors’ response to the first round of review.]

Our decision has been reached after consultation between the reviewers. Based on these discussions and the individual reviews below, we regret to inform you that your work will not be considered further for publication in eLife.Establishing an in vitro assay that allows to study microtubule nucleation events from purified gTuRC at the single molecule level is overdue. Such an assay is required to answer fundamental and long-standing questions regarding the mechanism of nucleation. In this regard the assay using immobilized gTuRC and single molecule TIRF imaging, as presented in the current manuscript, is an important achievement. Unfortunately, it has not yet been used to its full potential. Several interesting observations are made, but the chosen path and derived conclusions in this manuscript are challenged by three recently published Cryo-EM structures of native gTuRC, including gTuRC purified from *Xenopus* egg extract as in the present study (Liu et al., 2019). The structures show that native gTuRC does not match the symmetry of a microtubule. It is in an 'inactive' conformation, one that is not well suited to promote lateral tubulin interactions, questioning the validity of the presented model. Considering this and other issues raised by the reviewers the manuscript does not provide the conceptual advance required by eLife.

Although all reviewers recognized this work as far-reaching in studying MT nucleation from γ-TuRC at the single molecule level and a necessary step forward, the results from the original manuscript did not account for key information on the structure and conformation of γ-TuRC described in the works that were published in the last months (Liu et al., 2020; Wiezcorek et al., 2020) while our original manuscript was in review. We took this feedback seriously and performed new experiments and computational modeling to address these open questions. We now provide a more comprehensive understanding of how MT nucleation from γ-TuRC occurs as described below and believe that the revised work provides the conceptual advance for the microtubule and the broader cytoskeleton fields.

Re-examination of g-/αβ-tubulin interaction

We re-examined the biochemical interaction between g-tubulin and αβ-tubulin to address the comments raised by reviewer 3. First, we measured the interaction of purified g-tubulin and αβ-tubulin via size exclusion chromatography (SEC) experiments (Figure 4A and Figure 4—figure supplement 1A). We performed SEC of g-tubulin (150nM) with αβ-tubulin at two different concentrations, 10μM (low) and 35μM (high), as well as of g-tubulin and αβ-tubulin alone. At low concentration of αβ-tubulin alone, we observed a single species for αβ-tubulin heterodimer eluting in fractions H-K (Figure 4—figure supplement 1A(i)). At high concentration of αβtubulin alone, we additionally observed αβ-tubulin oligomers in our setup (Figure 4—figure supplement 1A(ii)), red arrows marked in fractions B-C) that correspond to two αβ-/αβ-tubulin in complex from estimated molecular weight using the gel filtration standards. While, g-tubulin alone runs as a broad peak in fractions I-N (Figure 4A(i)), upon incubation with αβ-tubulin, g-tubulin additionally elutes at an earlier elution volume in fraction H ((Figure 4A(ii-iii)) and its elution profile is modified to following αβ-tubulin. Surprisingly, this was observed with both low and high αβ-tubulin concentration, showing that the g-/αβ-tubulin interaction occurs even at low concentrations and suggests that this interaction is higher affinity than that between two αβ-/αβ-tubulins, which requires higher concentrations.

Second, we turned to single molecule binding assays as an orthogonal approach (Figure 4B). We attached buffer or biotin-αβ-tubulin molecules on the coverslip at low concentration of 100-200nM and introduced fluorescent αβ-tubulin (Figure 4B(i)) or fluorescent g-tubulin molecules (Figure 4B(ii)) in solution also at low concentrations (50-100nM) to avoid formation of MTs or g-tubulin assemblies. In both cases, specific binding of αβ-tubulin and g-tubulin molecules binding to biotin-αβ-tubulin on the coverslip was observed (compare left- and right-panels in Figure 4B(i-ii)), yet more fluorescent g-tubulin molecules than fluorescent αβ-tubulin molecules bound. Performing single molecule analysis while accounting for total labelled percentage of αβ- and g-tubulin in our purifications as detailed in the Methods section, revealed that nearly 15-fold more g-tubulin molecules than αβ-tubulin bind to the attached biotin-αβ-tubulin molecules (Figure 4B).

Lastly, we verified this interaction with biolayer interferometry (Figure 4—figure supplement 1B), which corroborated the SEC and single molecule assays. We would like to note that a reverse configuration was used for interferometry assays in the previous version of the manuscript where g-tubulin was attached on the probe via its His-tag and αβ-tubulin was introduced in solution, as compared to the revised version.

Upon further investigation, we found that the proximity of the His-tag, used as the site of attachment to the probe, to the αβ-tubulin’s binding surface occluded the binding of αβ-tubulin, both in the interferometry and single molecule assays (data not shown). Further, g-tubulin shows significant non-specific background binding to the probe in interferometry experiments (Figure 4—figure supplement 1B(ii)). To overcome these drawbacks in the revised manuscript, we developed new binding assays that either do not involve any immobilization of proteins (SEC) or and where attached molecules could be directly visualized (single molecule assay). These biochemical measurements, therefore, serve as most direct assessment of binding and supersede the interferometry experiments now presented in the supplement.

In sum, we find that a high affinity g-/αβ-tubulin interaction at the nucleation interface promotes gTuRC’s MT nucleation activity. We are grateful to reviewer 3 for their feedback that prompted us to assay g-/αβ-tubulin interactions with alternate methods which resulted in a more comprehensive biochemical understanding.

Monte Carlo simulations of microtubule nucleation from γ-TuRC

We next developed a detailed mathematical model to capture the dynamics of MT nucleation from γ-TuRC. Our Monte Carlo simulations were based on previous models for polymerization and depolymerization of the MT plus-end, first developed by Odde and Cassimeris (VanBuren et al., 2002). We assumed a 13-protofilament (pf) geometry of γ-TuRC and the MT lattice with a pitch of 3 tubulin monomers (Figure 5A). The assembly of αβ-tubulin on γ-TuRC or the MT lattice occurred with a constant on-rate *k*_on_ (μM^-1^s^1^pf^-1^) with longitudinal and lateral Gibbs free energy of interaction with other αβ-tubulins, ∆*G_Long,αβ-αβ_* and ∆*G_Lat,αβ-αβ_*, respectively, and with longitudinal free energy of interaction with gtubulin at the interface of γ-TuRC, ∆*_Long,γ-αβ_*. To incorporate the data obtained from recent cryo-EM structures (Liu et al., 2020; Wiezcorek et al., 2020), we assumed that native γ-TuRC has an open conformation where interactions between tubulins on neighboring sites could not occur. A thermodynamic penalty,∆*G_γTuRC-conf_*, and a rate constant *k_γTuRC-conf_*(s^-1^) determine the transition from this open to closed γ-TuRC conformation where lateral tubulin interactions can occur. We ran stochastic simulations with these assumptions and determined the input parameters from our experimental data.

We determined the values for parameters *k*_on_, ∆*G_Long,αβ-αβ_* and ∆*G_Lat,αβ-αβ_* by fitting our experimental measurements of growth speed (Figure 5—figure supplement 1A). The resulting values are in agreement with previous estimates for these parameters (VanBuren et al., 2002; Mickolajczyk et al., 2019). ∆*G_Long,γ-αβ_* was estimated to be 1.1 times ∆*G_Long,αβ-αβ_* based on our experimental binding affinity measurements (Figure 4). Finally, we explored the behavior for different values of the two unknown parameters ∆*G_γTuRC-conf_* and *k_γTuRC-conf_*. We note that the total energy of transition to a closed γ-TuRC state is affected by the number of tubulin dimers that bind on the neighboring sites on the open γ-TuRC. Specifically, the free energy of all (*n*) lateral contacts that can form upon closure adds to the total transition energy between the start and end state, ∆*G_γTuRC-conf_* + *n*∆*G_Lat,αβ-αβ_*.

With this model, we could incorporate features of γ-TuRC that contribute to its nucleation activity including its conformation, blunt-end geometry and differential biochemical interactions. To our surprise, the simulations recapitulate our experimental measurements in detail. First, we observe a sharp transition from zero MT length on γ-TuRC to a continuously growing MT (Figure 5B). Second, this transition occurs at variable time for each γ-TuRC and the resulting nucleation curves obtained from hundreds of simulations match our experimental curves (Figure 5—figure supplement 1B(i)-C(i)). Analogous to our experimental results, we measured the cooperativity of αβ-tubulin dimers in the rate-limiting transition state and found that it is determined by the magnitude of ∆*G_γTuRC-conf_* (Figure 5C) but not affected significantly by changing *k_γTuRC-conf_* or ∆*G_Long,αβ-αβ_* (Figure 5—figure supplement 1B(ii)-C(ii)). For the specific parameter values of ∆*G_γTuRC-conf_* = +10*k_b_T* and *k_γTuRC-conf_*= 0.01s^−1^, the nucleation kinetics strikingly agree with our experimental data for γ-TuRC-mediated MT nucleation (Figure 5—figure supplement 2A). Hundreds of these simulations do not result in successful MT nucleation at αβ-tubulin concentration less than 7μM, and increasing MT nucleation events occur with 10-25μM tubulin (Figure 5—figure supplement 2A(i)). The cooperativity of 4 αβ-tubulin dimers is predicted in the transition state (Figure 5—figure supplement 2A(ii)), in agreement with our experimental data.

To gain deeper understanding of how γ-TuRC nucleates a MT, we characterized the dynamics of αβ-tubulins from individual model simulations. When the simulation begins, we find that many αβ-tubulins interact with the open γ-TuRC longitudinally, or with an existing αβ-tubulin in a pf after its assembly onto γ-TuRC (Figure 5B, left insets). While many of these association events happen rapidly, these αβ-tubulins are only bound by the longitudinal bond energy and then dissociate from γ-TuRC or the γ-TuRC bound αβtubulin. On the other hand, αβ-tubulin dimers persist when assembled on a corner site in a MT lattice that has both longitudinal and lateral contacts. These events drive MT assembly, as has been shown by many works in the last decade (Ayaz et al., 2014; Mickolajczyk et al., 2019; Rice et al., 2019). At the sharp transition when MT nucleation occurs (Figure 5B, right insets), many αβ-tubulin dimers in the simulations stochastically assemble on neighboring sites on γ-TuRC and the transition to a closed γTuRC state becomes favorable. In other words, upon γ-TuRC closure, the favorable free energy from lateral interaction between αβ-tubulins overcomes the energy penalty from its conformational change. Combining the results from 2,119 simulations reveals that on average 5.2 ± 1 total αβ-tubulin dimers are present with γ-TuRC at this transition state (Figure 5D, left). Most strikingly, these αβ-tubulins arrange on neighboring sites in laterally-arranged groups of 2-5 αβ-tubulins (Figure 5D, right). The most probable transition state has 4 αβ-tubulin arranged on neighboring sites that form 3 lateral bonds, and this physically represents the power-law exponent measured from the average kinetics of MT nucleation.

Finally, our simulations are also able to predict the kinetics of MT nucleation from a closed γ-TuRC conformation and quantitatively predicts the measured MT nucleation dynamics from blunt seeds (Figure 5—figure supplement 2B). In summary, our computational model provides essential insights into the reaction scheme and transition state of MT nucleation from γ-TuRC.

Probing the conformation of γ-TuRC before and after microtubule nucleation

In agreement with the recent cryo-EM structures, our revised manuscript supports a model in which a conformational change in γ-TuRC from an open to a closed state occurs during the rate-limiting transition step for MT nucleation.

A high resolution cryoEM structure of the γ-TuRC-nucleated MT-end will be needed to specifically demonstrate what conformational differences in γ-TuRC occur during MT nucleation. Further capturing the transition state with a few tubulin dimers will require the development of new methods that can resolve at high resolution how tubulins assemble on γ-TuRC, and will constitute the next decades of work from the field. Despite tremendous progress in cryo-EM of the MT lattice (work by Nogales lab and others), high resolution structure of intermediates either in the MT plus-end dynamics and in MT nucleation is yet to be reached. While we are very interested in pursuing this, such an endeavor is outside the scope of this work.

Finally, we believe that the mechanism we describe is necessary to place the molecular structure of a postnucleated γ-TuRC and any transition state into a mechanistic scheme. As such, our biochemical investigation stands independently as a valuable and timely contribution to the current literature. The work presented here is a critical step toward understanding MT nucleation, which we expect the field will address in depth in the next decade.

Role of putative activators in γ-TuRC mediated nucleation

Previous methods used to identify factors that alter the function of γ-TuRC have been indirect and could not distinguish between γ-TuRC-mediated and spontaneous nucleation. These experiments also show a large amount of variability and were not well suited to determine the role of factors that have small effects on MT nucleation. E.g. published results on the effect of NME7 reported a minor, 2.5-fold, increase in activity (Liu et al., 2014), while a high variability in the role of CDK5RAP2’s g-TuNA motif has been reported: 1.7-fold in Liu et al., 2020 and 7-fold in Choi et al., 2010. Further, γ-TuRC purified with CDK5RAP2’s g-TuNA motif and NME7 still forms an open configuration (Liu et al., 2020; Wiezcorek et al., 2020; Consolati et al., 2019), raising the question how this structural data can be reconciled with the previous biochemical measurements.

Here, we used an experimental setup that allows direct visualization and measurement of γ-TuRCmediated MT nucleation, as well as the ability to distinguish these nucleation events from spontaneous ones, and assessed the role of these putative factors. We examined the role of several putative activation factors in regulating γ-TuRC’s conformation and nucleation activity. First, the γ-TuRC nucleation activation domain (g-TuNA) from *Xenopus laevis* protein CDK5RAP2 does not significantly increase MT nucleation activity from γ-TuRC (Figure 6A-B). NME7, another proposed activator of γ-TuRC-mediated nucleation (Figure 6B and Figure 6—figure supplement 1) did not significantly affect γ-TuRC-mediated nucleation either. Similar results were observed for TPX2 (Figure 6C-D).

Discussion of the regulation of γ-TuRC-mediated nucleation

Here we would like to address the final point raised in response to our appeal letter by the reviewers.

The current view is that there is a good reason for having γ-TuRC in an inactive conformation: to spatially restrict nucleation. Otherwise γ-TuRC would nucleate MTs anywhere in the cell, once tubulin starts binding. […] This would still not be compatible with the fact that in cells nucleation occurs from a minority of γ-TuRCs at MTOCs and not the large majority of γ-TuRCs in the cytosol.

We thank the reviewers for raising this important point. First, we note that despite their open conformation, purified γ-TuRCs nucleate MTs efficiently as shown here. Similarly, γ-TuRCs present in the cytosol in vivo also nucleate MTs efficiently. This is best exemplified by branching MT nucleation that occurs in the cytoplasm, generates thousands of MTs (Petry et al., 2013; Alfaro-Aco et al., 2017; Thawani et al., 2018), constitutes a major pathway for MT nucleation and drives spindle assembly in many organisms compared to the centrosomes (Decker et al., 2018; David et al., 2019). We propose that native, open γ-TuRC could function and be regulated in an alternative way by tuning the availability of αβ-tubulin subunits. Concentrating tubulin on MTOCs such as centrosomes (400μM tubulin, Baumgart et al., 2019),on branched MTs by TPX2 (King and Petry, 2019), or directly on γ-TuRC via XMAP215 as shown here and in a previous work (Thawani et al., 2018) could regulate MT nucleation. Cytosolic γ-TuRC’s activity could also be regulated by sequestering αβ-tubulin by Stathmin or MCAK in the cytosol and later releasing it during the cell cycle through phosphorylation. This could allow for spatiotemporal control of MT nucleation.

Second, there is little experimental evidence supporting the widely-believed hypothesis that gTuRC conformation is modified to a closed one by activation factors at the MTOCs. While g-TuSC rings on the spindle pole body of a yeast cell have a smaller diameter (Kollman et al., 2015), this was observed after the g-TuSC rings have nucleated *MTs*, which is consistent with the proposition we outlined above. Further, several factors proposed as putative activators of γ-TuRC have since been shown to either play a role in the assembly of γ-TuRC or to function as localization factors (Kollman et al., 2010; Leung et al., 2019; Liu et al., 2020).

Notably, the possibility of γ-TuRC closure by αβ-tubulin and its implications on MT nucleation have also been discussed in the recent cryo-EM papers (Liu et al., 2020; Consolati et al., 2019), in consensus with our model. We discuss this points in our revised manuscript in the Discussion section.

“In this model, locally concentrating soluble αβ-tubulin could upregulate the levels of γ-TuRC-mediated MT nucleation, e.g. as recently shown through accumulation of high concentration of tubulin dimers at the centrosome by MAPs^55,58^ and by co-condensation of tubulin on MTs by TPX2 during branching MT nucleation^46^, and finally via specific recruitment of tubulin on γ-TuRC through the binding of XMAP215 as shown here^29,48^.”

Finally, in light of several comments from the reviewers, we have drastically re-structured the manuscript and modified the Discussion section. We now first present our findings on characterization of γ-TuRCmediated, spontaneous and seed-mediated nucleation (Figure 1, Figure 2, Figure 3 and related supplements). We then discuss biochemical features of γ-TuRC that determine its nucleation activity and develop simulations that explain how γ-TuRC nucleates a MT (Figure 4, Figure 5 and related supplements). Lastly, we examine the role of several MT-associated proteins in γ-TuRC-mediated nucleation (Figure 6, Figure 7 and related supplements).

Reviewer #1:The authors use γ-TuRCs purified from *Xenopus* egg extracts in combination with single molecule microtubule nucleation assays to gain insight into the process of microtubule nucleation. The key findings are that γ-TuRCs enhance microtubule nucleation by promoting the lateral association of a/β-tubulin molecules and that nucleation from γ-TuRCs is further promoted by XMAP-215. The experiments are well-executed and the data is nicely presented (although I have some questions about N numbers – see below). Overall, I think there are some interesting insights into γ-TuRC-mediated nucleation and the experiments are at the cutting edge, but I feel that we do not learn a great deal more about the role of γ-TuRC and XMAP-215 in microtubule nucleation to justify publication within a broad spectrum high-impact journal such as eLife. Several results or notions are already known e.g. γ-TuRCs and XMAP-215 promote microtubule nucleation, a higher concentration of tubulin is required for nucleation than for microtubule elongation, γ-tubulin forms filaments, XMAP-215 promotes nucleation, XMAP-215 associates with γ-TuRCs. Moreover, the main message, that γ-TuRCs promote lateral association of tubulin dimers, is widely assumed based on γ-TuRC structure and because longitudinal a/β-tubulin contacts are believed to be stronger than lateral contacts. As the authors themselves state, "the insights on MT nucleation by γ-TuRC and XMAP215 provide an essential basis to build upon". I therefore feel that, without further advances, such as how γ-TuRCs can be activated to better promote microtubule nucleation, the paper would be better suited to a more specialised journal, such as JBC.

We thank the reviewer for their insightful comments that helped us improve the manuscript significantly by performing several new experiments and interpreting our results in light of the cryo-EM structures of γtubulin ring complex. Here, we clarify several points noted by the reviewer to reflect the advance provided by our work.

First, while γ-TuRC has been widely assumed as the MT nucleator, its activity has been reported to be low, and often comparable to spontaneous MT nucleation (Kollman et al., 2010; Choi et al., 2010; Liu et al., 2014; Thawani et al., 2018). However, to directly compare the activity of γ-TuRC-mediated and spontaneous nucleation, it is necessary to assay these activities side-by-side using the same experimental setup, as well as live visualization where any spontaneously nucleated MTs in γTuRC-reaction can be distinguished from γ-TuRC-nucleated ones. Additionally, while this observation has been made for MT assembly from seeds (Wieczorek et al., 2013), it has never been demonstrated that MT nucleation from γ-TuRC required higher tubulin concentration than MT polymerization before our work. Instead, a number of reviews (Kollman et al., 2011; Roostalu and Surrey, 2017) had suggested that the dynamics of MT nucleation from γ-TuRC template could resemble the plus-end of a MT template. In our work, we developed sensitive assays and quantitative metrics to define the nucleation efficiency and rate-limiting step of γ-TuRC at the single molecule level, which for the first time could be directly compared to spontaneous and MT seed based nucleation. The results obtained and reported in our manuscript regarding this comparison are novel. Most importantly, our data on the composition of the ratelimiting, transition state consisting of γ-TuRC and four tubulin dimers. Together with the computational modeling, our works provides gives first insight into the dynamics and transition state of γ-TuRC-mediated MT nucleation, where laterally-associated tubulin dimers promote an open to closed transition in γ-TuRC conformation.

Second, as described in the common response above, we comprehensively investigated how γTuRC nucleation occurs by pairing experimental measurements and computational modelling. We characterized the biochemical features that define γ-TuRC-mediated nucleation, and find that the longitudinal γ-/αβ-tubulin interaction affinity is high. This high affinity promotes γ-TuRC’s nucleation activity by increasing the dwell time of αβ-tubulin dimers.

Third, we directly test the effect of specific MAPs on γ-TuRC-mediated nucleation. We believe that testing the activity of two previously proposed putative activation factors, CDK5RAP2’s γ-TuNA motif and NME7, with a sensitive, direct measurement in this paper is important. The finding that these factors do not significantly affect MT nucleation by γ-TuRC is significant. This result clarifies many contradicting results for the roles of these factors from previous works, where γ-TuNA was both found to affect nucleation by 7-fold (Choi et al., 2010) or minimally (Liu et al., 2020), while neither γ-TuNA nor NME7 present in complex with γ-TuRC altered the structure of the γ-TuRC from an open conformation to closed one (Liu et al., 2020; Wieczorek et al., 2020; Consolati et al., 2019). These results suggest that these MAPs do not alone function as activators, but may instead support the assembly or localization of γ-TuRC. They also highlight that MT nucleation in the cell may be regulated primarily through the availability and distribution of αβ-tubulin dimers, as is discussed in the reviewed manuscript. Only XMAP215 functions as the bona-fide co-nucleation factor with γ-TuRC and increases MT nucleation efficiency by 30-fold. These measurements had to be made side-by-side with our sensitive, direct assay to assess γ-TuRC’s nucleation activity in order to reveal the role of any of these factors in MT nucleation.

Finally, we emphasize that this information could not be obtained simply from solving EM structures, as recently achieved, as they only captured the pre-nucleation state and did not reveal anything about the nucleation reaction. All these insights could only be obtained from single molecule light microscopy paired with reaction kinetics analyses, which now constitute the state-of-the-art measurements. The transition state of γ-TuRC with four αβ-tubulin dimers represents an important milestone in understanding the γ-TuRC-mediated MT nucleation reaction scheme and its steps.

More specific comments:1) The title should be more specific to the work done, as the current title indicates that the authors have solved the entire mechanism of nucleation.

We thank the reviewer for this comment and have modified the title of the paper to, “The transition state and regulation of microtubule nucleation γ-TuRC revealed by single molecule microscopy” to specifically reflect the work done, and additionally welcome any alternate suggestions for the title.

2) Abstract: "the underlying mechanism largely remains a mystery". This is a difficult statement to swallow. I would argue that the underlying mechanism (templating the addition of tubulin dimers into a tube-like structure) has been widely assumed (although not directly proved) for a long time, in particular after the 2010 Nature paper from David Agard's group that showed the structure of the yeast "γ-TuRC". This was good proof that the template model was correct, rather than the protofilament model. I think the authors should therefore modify their statement to something like: "While it has been assumed that…., here we provide direct evidence for….".

We thank the reviewer for this suggestion. We made major changes to the Abstract to incorporate the specific suggestion. Also, the Abstract is now more focused on the new quantitative measurements made in this work that constitute the major advance compared to the previous literature.

3) Abstract: "…we uncover that γ-TuRC nucleates a MT more efficiently than spontaneous assembly". This was already known – many studies have shown (using more standard microtubule nucleation assays) that microtubules form more readily in the presence of γ-TuRC.

While some previous works (Zheng et al., 1995; Oegema et al., 1999) reported γ-TuRC’s nucleation activity to be higher than spontaneous nucleation, a number of recent works have described γTuRC-mediated nucleation to be inefficient and similar to the levels of spontaneous nucleation in their assays (Kollman et al., 2010; Choi et al., 2010; Liu et al., 2014; Thawani et al., 2018; Liu et al., 2020). Some proportion of the nucleated MTs in all reports could be due to spontaneous assembly, which cannot be distinguished from γ-TuRC-nucleated MTs in the traditional, endpoint MT nucleation assays. In contrast, we precisely measured the nucleation activity for γ-TuRC-mediated and spontaneously nucleated MTs, which can be further distinguished by observing the growth of MT minus-end for the latter case but not the former. Such a side-by-side comparison in a sensitive assay was missing and provides insight into how γ-TuRC nucleates a MT. To highlight the most critical result from our measurements, we now state in the Abstract, “Whereas spontaneous nucleation requires assembly of 8 αβ-tubulins, nucleation from γ-TuRC occurs efficiently with a cooperativity of 4 αβ-tubulin dimers.”

4) Figure 1B: Three γ-TuRC nucleation sites are indicated by arrows – is this all of the γ-TuRCs present within this field (Can the authors show a two-colour image overlay)? If there were only three γ-TuRCs, it would not fit with the finding that spontaneous nucleation does not occur at 10.5μM tubulin….

We thank the reviewer for pointing this out and realized that this representation could be improved. All MTs nucleated in the field of view in Figure 1B were nucleated from γ-TuRC molecules. We now represent all MTs nucleated already in the first frame with yellow arrows and all new nucleation events between first and last frames with blue arrows. We additionally compared MT nucleation with and without γ-TuRC sideby-side in an experiment requested by Reviewer 3 (revised Figure 3—figure supplement 1B-C). This confirmed that γ-TuRC-mediated nucleation occurs efficiently at 10.5μM tubulin, while no MTs are nucleated spontaneously (i.e. without γ-TuRC).

5) Subsection “Defining the microtubule nucleus on γ-TuRC”: "Surprisingly, γ-TuRC nucleated MTs starting from 7 μM tubulin (Figure 1D), which is higher than the minimum tubulin concentration (C*) needed for growth at pre-formed MT plus-ends (C* = 1.4μM, Figure 1E)". I don't think this is surprising, given the results of Wieczorek et al., 2015 – these authors used solution exchange to show that, while nucleation from a microtubule seed required ~6μM tubulin, microtubules could continue to grow at ~1μM tubulin. This statement should be toned down and the authors should refer to Wieczorek et al. results.

We appreciate this point by the reviewer and have removed the word “surprisingly” from this statement in the manuscript. In our work, we directly compared γ-TuRC-mediated MT nucleation and seed-mediated assembly. Our results for seed-mediated MT nucleation differ from those of Wiezcorek et al., such that MT seeds nucleate near the critical tubulin concentration (Figure 3C-E). The differences in results from those of Wiezcorek et al., are due to different assay conditions, as detailed in the response to point (7) by the reviewer. The tubulin concentration needed for nucleation from γ-TuRC in this work cannot, therefore, be directly compared with Wiezcorek et al. results where a different assay condition produces the kinetic barrier and requirement of high tubulin concentration. For this reason, we performed the seed-mediated nucleation assay is our assay conditions to compare side-by-side with γ-TuRC-mediated nucleation in the same conditions. We request the reviewer to refer to our response for point (7) below for an in-depth explanation.

6) Figure 1G: The authors use power-law calculations to calculate that ~3.7 tubulin dimers are required for nucleation. It is not clear exactly how they did this. Given that this power analysis underlies the main finding of the paper, I think it deserves a full explanation.

We thank the reviewer for these suggestions. In the revised manuscript, we included a new subsection, “Power-law analysis of critical nucleus size on γ-TuRC” with a detailed explanation for the power-law analysis including the governing equations for the reaction kinetics balance. We further included similar analysis details for spontaneous nucleation and nucleation from MT ends in subsection “Preparation, microtubule assembly from blunt microtubule seeds and data analysis” and subsection “Size exclusion chromatography of γ-tubulin and αβ-tubulin” respectively.

7) Subsection “Does γ-TuRC nucleate a microtubule via the blunt plus-end model?”: The authors use microtubule seeds to compare their nucleation efficiency to γ-TuRCs and find that microtubules can assemble from these seeds at only 2.45μM tubulin. This is significantly lower than that found by Wieczorek et al., 2015 (6μM tubulin) – can the authors comment on this and suggest an explanation?

We thank the reviewer for this question, which echoes the related point (5). The differences between our results and those of Wiezcorek et al., 2015 are due to the different assay conditions, which we elaborate on below.

This assay was performed under different conditions by Wiezcorek et al., 2015. First, the preparation of our coverslips relies on covalently attaching PEG molecules to the glass, which is superior than the non-specific passivation of coverslips used in other works including Wiezcorek et al., 2015 as well as our previous works (Thawani et al., 2018). Second, there are specific differences in the buffer composition used. As we were initially surprised by these findings as well, we repeated this experiment exactly as published (Author response image 1). Under the assay conditions reported in Wiezcorek et al., we observed similar results where most MT seeds do not assemble polymerizing MT at 5μM tubulin, while at 7μM tubulin, MTs assemble from seeds (Author response image 1). This suggests that different assay condition or coverslip preparation results in distinct nucleation kinetics from seeds in vitro. Further, we characterized which assay condition represents that the barrier in assembly from seeds exists in a physiological context. We examined how MT assembly from seeds occurs in *Xenopus* egg extracts. Using the same coverslip preparation and MT seed attachment as Wiezcorek et al., we added *Xenopus* egg extracts instead of purified tubulin to visualize MT assembly. Surprisingly, MTs assembled immediately from the seeds (Author response image 1). We note that MT nucleation by seeds in extract is in contrast with γ-TuRC-mediated nucleation in egg extract, where a large number of γ-TuRCs are present in the cytoplasm, yet MT nucleation from these molecules is not observed without the addition of MTOCs or branching effector RanGTP (Petry et al., 2013; Thawani et al., 2018; Alfaro-Aco et al., 2017). In sum, the nucleation barrier for a blunt-to-tapered transition does not appear as the major contributor to the kinetics of γ-TuRC-mediated assembly either our in vitro setup or in *Xenopus* egg extracts.

**Author response image 1. sa2fig1:** 

Therefore, when repeated side-by-side with the same assay conditions, MT seeds do not display a nucleation barrier while γ-TuRC does. To clarify this point in the manuscript, in subsection “Contribution of end architecture of γ-TuRC to microtubule nucleation”, we state, “Notably, the distinct minimal tubulin concentration needed for seeds to assemble a MT as compared to a previous report^24^ results from the differences in assay conditions.”

8) Subsection “Molecular insight into microtubule nucleation by γ-TuRC”: Experiments to nucleate microtubules from γ-tubulin oligomers: I don't think this adds much to the paper. It is already known that γ-tubulin can form filaments and it is hard to understand how these filaments would nucleate microtubules, given that the filaments are presumably made of laterally contacting γ-tubulin molecules, thus creating a linear, rather than circular, template for the addition of a/β-tubulin dimers. It was also not clear from the methods what the final salt concentration was during the nucleation reactions, given that the g-tubulin was diluted in high salt buffer before being added to the reaction buffer. This is important as the authors show that tubulin oligomers do not form at high salt concentrations. If the authors want to keep these experiments in the paper, they need to cite work from the groups of Alvarado-Kristensson, Binarova and Monasterio who have shown that γ-tubulin can oligomerise into filaments.

We now cited the works suggested by the reviewer in subsection “γ-tubulin has a high affinity for αβ-tubulin” and specified the final salt concentration in the assay in subsection “Nucleation of microtubules from purified γ-tubulin”, “varying concentration of γ-tubulin was prepared by diluting purified γ-tubulin in a high salt buffer (50mM K-MES pH 6.6, 500mM KCl, 5mM MgCl_2_, 1mM EGTA), *[…]* to a final salt concentration of 44mM KCl”.

With further analyses of our negative stain EM data (Figure 4—figure supplement 2C-E), we show that γ-tubulins within the filament arrange into laterally-associated arrays. We speculate that MT generation could occur from assembly of a αβ-tubulin dimers onto the γ-tubulin arrays, followed by further polymerization and closure of the tube to form a MT. Notably similar mechanisms involving a sheet-to-tube transition via a sheet-closure have been previously proposed for MT assembly (Wu et al., 2009; Vitre et al., 2008; Voter and Erickson, 1984). Finally, we agree with the reviewer that this is not a major point in the manuscript and have therefore, moved these data to the supplementary figures.

9) Subsection “Molecular insight into microtubule nucleation by γ-TuRC”: The stochastic models, used to predict how many γ-tubulin sites must to be occupied to have the best chance of 2 adjacent tubulin dimers being present, are based on a perfectly circular arrangement of γ-tubulin molecules (Figure 2H). Liu et al., 2019 have just shown the cryo-EM structure of the *Xenopus* γ-TuRC. It is in an open confirmation, where the first and last γ-tubulin molecules are not adjacent to each other. I presume this would change (albeit perhaps only slightly) the calculations from the stochastic model simulations, so the authors should repeat them using an open ring where there is no lateral contact if positions 1 and 13 are occupied.

We thank the reviewer for this comment. In the revised version, we developed Monte Carlo simulations to model the process of MT nucleation from γ-TuRC comprehensively. We describe the simulation methods and results in detail in the common response. Briefly, we incorporated γ-TuRC’s open conformation, the precise geometry of the seam, as well as stochastic association and dissociation of αβ-tubulin dimers to γTuRC’s sites or to other αβ-tubulins. This geometry, encoded in our detailed simulations, incorporates the reviewer’s suggestions. This model independently recapitulates the detailed dynamics of γ-TuRC-mediated MT nucleation of our measurements, and also identifies the arrangement of αβ-tubulin dimers in the transition state as described in the common response. We hope that the new results from the detailed simulation model addresses the points raised by the reviewer.

10) For quantifying the numbers of microtubules nucleated in e.g. Figure 1F, Figure 3A,3C it is not clear how many times the experiments were performed and whether the graphs represent mean or median values. Given the shakiness of the lines in the graph in Figure 3A, the N numbers seem low. I would expect that the experiments (being quick to perform) would have been performed several times (perhaps 5 to 10 times) and that a mean or median value was used to represent microtubule numbers at each timepoint. Can the authors confirm this, or do more repeats?

We appreciate this point raised by the reviewer and addressed it in the following way:

First, we performed and included additional experimental repeats and analyses. Between 3-5 repeats were performed for most MT nucleation experiments. We expanded on our description to also provide detailed information on the number of repeats performed in every figure legend. Notably our power-law analyses with 3-5 pooled replicates result in an estimate of 3.9 ± 0.5 tubulin dimers for γ-TuRCmediated nucleation, 8.1 ± 0.9 for spontaneous nucleation, 1 ± 0.3 for assembly from blunt MT end, and 3.3 ± 0.8 for XMAP215/γ-TuRC-mediated co-nucleation.

Second, each replicate for an experiment was performed with independent γ-TuRC purifications where the concentration of γ-TuRCs obtained varies between purifications. The variation in the absolute number of MTs nucleated between replicates mostly represents variance in number of γ-TuRCs. Therefore, reporting the mean or median value was not appropriate. In light of the reviewer’s comment, we instead developed new ways of combining data from independent replicates using the following two approaches. (i) For data where power-law analysis of nucleation versus concentration was performed (Figure 2C-D, Figure 3B, Figure 3D-E, Figure 7D-E), we reported a representative curve the nucleation kinetics to display the raw data and combined data from all repeats in the power-law analysis (Figure 2D, 2B, 2E, Figure 7E). (ii) To combine data in the power-law analysis from multiple γ-TuRC experiments, it was necessary to set a normalization factor that we specify in individual figure legend. For example, in the legend for Figure 2B, we state, “The rate of nucleation at 10.5μM was set to 1 to normalize differences in γ-TuRC concentration from individual experiments.” For experiments to study the effect of MAPs (Figure 6B, 6D, Figure 7B, 7F), we combined results from multiple experiments, as requested by the reviewer, by setting a normalization factor between different experimental repeats and displaying kinetics from all repeats on the plot. For example, in the legend for Figure 6D, we state, “To account for the variable γ-TuRC concentration across purifications, the number of MTs nucleated in control reactions at 150 seconds was set to 1. All data was pooled and reported.” We also report 95% confidence intervals on our data as detailed in Methods and individual figure legends.

Finally, we emphasize that these are the most challenging and time-intensive experiments we have witnessed in our careers. Getting these to work was a breakthrough in itself. We observed that some γTuRC’s dissociate and lose activity during freeze-thaw cycle, and therefore, we performed each assay with freshly purified γ-TuRC. Our most important metric was reproducibility, which was ensured with all experiments reported and we specify this in detail for individual figure panels.

11) Figure 3A: the authors show that TPX2 has no role in microtubule nucleation from γ-TuRCs, but previous studies (Wieczorek et al., 2015, Roostalu et al., 2015, Woodruff et al., 2017) have shown that TPX2 does promote nucleation. Do the authors believe there is a difference in the requirement of TPX2 when considering γ-TuRCs versus microtubule seeds as templates? Can the authors repeat previous results showing that TPX2 promotes nucleation from microtubule seeds, as a positive control to show that their purified TPX2 is functional?

We thank the reviewer for raising this important point and address it as follows. First, we tested the effect of TPX2 on γ-TuRC’s nucleation activity at a higher TPX2 concentrations (30-50nM; Author response image 1). While the TPX2 appeared to generate more MTs than γ-TuRC alone (Author response image 1), the formation of large tubulin clusters occurred at these higher TPX2 concentration and MTs were seen to be generated from these clusters (Author response image 1). The finding that TPX2 can co-condense with tubulin and generate MTs spontaneously is in agreement with previous work (King and Petry, Nat Comm 2020). Such a high concentrations of TPX2 (~50nM), above the co-condensation boundary, was also used in previous works where TPX2 was observed to nucleate MTs spontaneously when high TPX2 concentrations were used (Roostalu et al., 2015, Woodruff et al., 2017). However, a lower concentration of TPX2 (~1nM) is sufficient to bind to the MT lattice and does not form tubulin clusters (Roostalu et al., 2015; King and Petry, 2020). Because we wanted to measure the effect of soluble TPX2 on γ-TuRC’s nucleation activity and to avoid confounding measurements via tubulin clustering, we operated below this phase boundary at 10-20nM TPX2. At this concentration, TPX2 completely coated the MT lattice, yet did not have a significant effect on γ-TuRC-mediated nucleation (Figure 6C-D).

In summary, TPX2 does not significantly affect γ-TuRC-mediated nucleation, whereas its high concentrations are required for spontaneous MT assembly. Several studies from our lab have previously shown that in *Xenopus* egg extracts, TPX2 also does not directly increase MT nucleation from γ-TuRC. This is the case at TPX2’s physiological concentration (30nM) as well as at high concentration (1μM) (Alfaro-Aco et al., 2017; Thawani et al., 2019).

12) Subsection “How do γ-TuRC and XMAP215 synergistically nucleate microtubules?”: "XMAP215 effectively decreases the minimal tubulin concentration necessary for MT nucleation from γ-TuRC to 1.6 μM". How was this calculated? Also, does adding XMAP215 also increase the probability of spontaneous nucleation? In a model where XMAP215 binds γ-tubulin and uses TOG domains to add tubulin dimers for the initial nucleation event, the addition of XMAP215 should not increase the probability of spontaneous nucleation.

We appreciate these two questions. First, in Figure 7—figure supplement 1C, we display a set of experiment where constant γ-TuRC and XMAP215 concentration was used, while tubulin concentration was varied to allow a measurement of the critical nucleus size. XMAP215 and γ-TuRC together nucleate MTs starting from a tubulin concentration of 1.6μM, as quantified in Figure 7D. To address the second question, we compared side-by-side how many MTs are nucleated by XMAP215 alone, γ-TuRC alone and γ-TuRCs with XMAP215 (Figure 7—figure supplement 1A), which were quantified in Figure 7—figure supplement 1B. We find that at 7μM tubulin, neither γ-TuRC nor XMAP215 alone significantly nucleate MTs, yet robust MT nucleation only occurs when XMAP215 and γ-TuRC are present together. We hope this addresses the reviewer’s question.

13) Figure S4B: the authors test whether the C-term γ-tubulin-interacting region of XMAP215 promotes microtubule nucleation, and show that it does not. It would be informative to also test whether the rest of XMAP-215 alone (i.e. XMAP-215 minus the γ-tubulin binding region) can promote microtubule nucleation from a γ-TuRC. Presumably not, but this truncated form of XMAP-215 should in theory be able to bind to microtubule plus ends via the TOG domains. It would therefore be a nice experiment to show that it is not plus-end bound XMAP-215 that is promoting nucleation, but it is the XMAP-215 that binds to the γ-TuRC.

XMAP215 construct containing TOG domains 1-4, and further repeats for C-terminal truncation TOG5CT. The results are displayed in Figure 7—figure supplement 1D-E and we compared these side-by-side with the activity of full-length XMAP215 (positive control) and γ-TuRC alone (negative control). To summarize, we find that neither the C-terminal, which binds γ-tubulin directly (Thawani et al., 2018), nor the Nterminal truncations of XMAP215 increase γ-TuRC-mediated nucleation significantly, while wild-type XMAP215 drastically induces MT nucleation from γ-TuRC in vitro.

Reviewer #2:The present manuscript uses purified γ-TuRC, tubulin, and different MAPs, to reconstitute microtubule nucleation in vitro, and to quantify nucleation rates, growth speed, and cooperativity of tubulin dimers. Using coated coverslips to anchor tagged γ-TuRCs, in combination with TIRF microscopy, several important observations are made:- Nucleation of microtubules occurs from γ-TuRCs de novo, and this is more efficient than spontaneous assembly.- The critical nucleus for nucleation from γ-TuRCs comprises only 3-4 tubulin dimers, which is less than half the size required for spontaneous assembly.- γ-TuRC-mediated nucleation differs from plus-end growth at blunt microtubule ends, where a single tubulin dimer suffices as a nucleus.- γ-TuRC-mediated nucleation is not influenced by TPX2. By contrast, XMAP215 decreased the minimal tubulin concentration and increased the nucleation rate, by binding first to the γ-TuRC, yet requiring a similar critical nucleus of 3-4 tubulin dimers.Altogether, these data provide extremely valuable insight into the mechanisms of microtubule nucleation, since they clarify multiple major issues: γ-TuRCs act directly as templates for microtubule nucleation, ruling out other hypotheses that propose spontaneous nucleation, followed by capping of the free minus-ends by γ-TuRCs. Most importantly, we have for the first time a concrete idea about the critical size of the tubulin nucleus, and about the involvement (or non-involvement) of XMAP215 and TPX2 in the nucleation process.The manuscript is timely, since several other studies on the γ-TuRC (e.g. structure of the native full complex) are likely to be published by other groups in the very near future.

We thank the reviewer for recognizing the breakthrough and timeliness of our work.

Reviewer #3:The manuscript by Thawani et al., together with a recently posted study by the Surrey group on the bioRxiv preprint server, is the first to study microtubule nucleation from purified, immobilized gTuRC at the single molecule level by TIRF microscopy. Using this approach the authors investigate and compare nucleation from gTuRC with spontaneous MT assembly and nucleation from preformed MT seeds. They find that nucleation from gTuRC is more efficient than spontaneous assembly, but less efficient than nucleation from MT seeds. They argue that this is due to relatively weak longitudinal affinity between g-tubulin and ab-tubulin based on measurements of ab-tubulin interaction with purified g-tubulin and ab-tubulin in biolayer interferometry assays. XMAP215 significantly increases gTuRC nucleation activity. They estimate that nucleation from gTuRC alone requires 3-4 ab-tubulin dimers to bind to gTuRC before the kinetic nucleation barrier is overcome and argue that this is facilitated by the lateral array of g-tubulins in the gTuRC.The establishment of TIRF imaging of nucleation from gTuRC at single molecule level is overdue and the presented work is a great achievement in this regard, providing interesting new ideas and experimental possibilities. However, the authors base their conclusions on various assumptions that cannot be made without further evidence and control experiments. A major issue is the fact that native gTuRC is not a perfect template that matches the symmetry of a microtubule, as revealed by three independent recent Cryo-EM studies by the Kapoor, Schiebel and Surrey labs. Overall, the study requires significant additional work that in my opinion would go beyond what would be reasonable for a revision in eLife.

We thank the reviewer for the positive comments recognizing the breakthrough of this work and for providing critical feedback of how this work can be advanced further to provide deeper mechanistic insight and reconcile it with the recent cryo-EM structures. As described in the common response above, we reevaluated the previous assumptions, added several entirely new experiments and developed computational models to provide deeper insight how a MT is nucleated by γ-TuRC based on the recent cryo-EM structures. In the following, we directly address all comments raised by the reviewer.

Major points:1) Subsection “Reconstituting and visualizing microtubule nucleation from γ-TuRC”: "MT nucleation events occurred specifically from single gTuRC molecules" – gTuRCs are not labelled in this assay, how do the authors get to this conclusion? How can they exclude that some of the MTs are from spontaneous nucleation?

We thank the reviewer for raising this crucial point, which we address in the following response. First, we included new data in which purified γ-TuRCs were covalently attached to both biotin- and Alexa-568 fluorescent-dye (Figure 1C and Figure 1—figure supplement 1C). MT nucleation events were always observed from individual Alexa-568 labelled, green spots (Figure 1C) demonstrating that individual γ-TuRC molecules nucleate MTs and significant spontaneous nucleation does not occur. We note that many experiments in this work were not performed with fluorescent γ-TuRC because we empirically found that additional labelling of γ-TuRC with the dye using a cysteine reaction resulted in fewer γ-TuRCs and lower activity at the end of the purification, vis-a-vis from biotinylated γ-TuRCs only (data not shown). Therefore, we used biotinylated γ-TuRCs for most experiments which was sufficient to answer our questions. Second, and of primary importance, we note that the characteristics of MTs nucleated by γ-TuRC and those nucleated spontaneously can be clearly distinguished using our live assay. In Author response 4A, we show kymographs for γ-TuRC-nucleated MTs from Figure 1C and for spontaneously nucleated from Figure 3—figure supplement 1A side-by-side. We marked the minus- and plus-end of MTs with dashed and solid yellow lines, respectively to clearly demonstrate this. For γ-TuRC mediated nucleation, MTs emerge from zero length from the coverslips and pivot around the nucleation site due to the attachment. Most importantly, the minus-end of these MTs, marked by dashed yellow lines in the kymographs, does not grow while only the plus-end grows, as shown in solid yellow lines. In contrast, for spontaneously nucleated MTs, nucleation occurs from solution and significant translational and rotational motion of MTs is observed due to the lack of attachment.

Finally, both ends of MTs, minus- and the plus-ends shown with dashed and solid lines respectively, elongate. Another way to observe whether the minus-end grows or is capped is by following fiducial marks in the MT lattice that occur stochastically from to fewer fluorescent tubulin molecules incorporating into the lattice than average. In Author response image 2, we displayed some of these fiduciary marks with red dotted lines. For MTs nucleated from γ-TuRCs, the distance from these marks to the minus-end of the MT does not change over time, while that from the plus-end increases. In contrast, for spontaneously nucleated MTs, the distance of these fiduciary marks from both minus- and plus-end increases over time. This again shows that minus-end does not grow for γ-TuRC-nucleated MTs, while it does for spontaneously nucleated MTs. This was used to distinguish γ-TuRC-mediated and spontaneous nucleation events.

Finally, as requested by the reviewer in point (6) below, we compared γ-TuRC-mediated and spontaneous nucleation side-by-side (Figure 3—figure supplement 1B-C). As described in the manuscript, few spontaneous nucleation events occur in comparison to γ-TuRC mediated nucleation events with the tubulin concentrations used here. We elaborate on this further in our response to point (6).

2) The main conclusions of this study regarding the nucleation mechanism are based on the assumption that gTuRC is a perfect template, which has recently been disproven. Cryo-EM data of purified native human and *Xenopus* gTuRC clearly show that the g-tubulin array in gTuRC does not match the symmetry of a MT, by displaying lateral gaps between g-tubulin molecules, in particular in the part of gTuRC that contains GCP4, 5, and 6. This would be expected to impact its ability to promote lateral interactions between ab-tubulin dimers.

We thank the reviewer for raising this important point. In the revised manuscript, we consider the implications of the γ-TuRC’s conformation revealed by the cryo-EM structures, as described in the common response above. Altogether, our revised work shows that γ-TuRC’s nucleation properties are defined by the thermodynamic barrier from the open conformation of native γ-TuRC which does not allow complete lateral αβ-tubulin interactions.

3) Figure 2C-E: It is assumed that γ-tubulin alone, through lateral interaction, forms a filament-like microtubule nucleation template. There are two issues here. First, there is no evidence that the g-tubulin filaments are indeed formed by lateral g-tubulin interactions. In fact, the EM image in S2C shows a fiber much thicker than expected.

We thank the reviewer for raising this point and would like to clarify that arrangement of γ-tubulin into filaments. To characterize the arrangement of γ-tubulins within filaments, we optimized the negative staining of γ-tubulin filaments to minimize background, acquired 370 micrographs and obtained 3D reconstructions of γ-tubulin filaments in RELION (Figure 4—figure supplement 2C-E), as described in the figure legend and subsection “Negative stain electron microscopy of γ-tubulin filaments”. Reconstructions revealed that the filaments are made of linear arrays of laterally-interacting γ-tubulins with a constant subunit repeat of 54Å in each array. This arrangement of γ-tubulins was also observed in the crystal contacts across unit cells in the structure of γtubulin (Aldaz et al., 2005). This subunit repeat distance does not match that between α- and βtubulins interacting longitudinally in a protofilament (40Å). In sum, γ-tubulins arrange laterally along the length of γ-tubulin filaments, an arrangement that differs from γ-tubulin arranged helically along the cross-section of a MT. Here we acknowledge personal communication with Drs. Michelle Moritz and David Agard, who observed a similar arrangement of γ-tubulin filaments and MT nucleation from these assemblies using a light scattering assay (King et al., bioRxiv 2020).

Second, the authors should present evidence that the formed ab-tubulin polymers are actually microtubules rather than other types of filaments (e.g. protofilaments or polymerization along g-tubulin filaments).

We thank the reviewer for raising this important point. We have several lines of evidence that show that the αβ-tubulin forms MTs here. First, we measured the αβ-tubulin intensity along the length of polymers generated by γ-tubulin and of MTs nucleated by γ-TuRC. We find that the mean αβ-tubulin intensity along their lengths to be similar (Author response image 3) and polymers grow with similar plus-end growth speed (Author response image 3). Second, we visualized the nucleation reactions described in Figure 4—figure supplement 2A with negative stain electron microscopy (Author response image 3). While a lot of protein background was observed on the EM grids likely due to high concentration of unpolymerized αβ-tubulin, MTs were clearly seen on the grids. Third, we find that γ-tubulin oligomers attached to the coverslips are also sufficient to nucleate MTs from the coverslips (Author response image 3), and large filaments are not necessary for this activity. Finally, as these polymers are not curved and short (less than 100nm) as previous observed for protofilaments (Portran et al., 2017), but are indeed MTs formed from αβ-tubulin.

**Author response image 3. sa2fig3:** 

In summary, our data shows that MTs nucleate from high concentration of γ-tubulins where γtubulin assembles into filaments. While some of the data provided in Author response image 3 will be of general interest to the field, it deviates from the main focus of the current manuscript, and we will communicate this in an independent study. We believe that revealing the arrangement of γ-tubulins in the filament (Figure 4—figure supplement 2C-E) clarifies that γ-tubulin filaments and oligomers do not position MT-like plus-ends, but instead form lateral arrays that are sufficient to nucleated MTs from αβ-tubulin dimers.

4) Fig2D: Are these really protofilaments? They seem much thicker. Also, can the authors exclude that g-tubulin polymerizes during coating of the chip?

We thank the reviewer for this comment, which prompted us to comprehensively characterize γ-/αβ-tubulin interaction using a number of biochemical approaches. As described in the common response above, using size exclusion chromatography (SEC) and single molecule binding assays (Figure 4A-C), we find that the affinity between γ-tubulin and αβ-tubulin is higher than that between two αβ-tubulins. Our interferometry experiments performed with the new orientation where αβ-tubulin is bound to the probe and γ-tubulin is in solution corroborates this result (Figure 4—figure supplement 1B(ii)). As outlined in the common response, the reverse orientation used before likely resulted in an occlusion of αβ-tubulin binding site on γ-tubulin and no binding could be observed. Further, as the reviewer correctly points out, γ-tubulin associates non-specifically with the probe to a lower extent in the interferometry experiments. Thus, we think that SEC and single molecule binding assays supersede this experiment. Hence, we have moved the interferometry experiments to the supplement as supporting observations. With this revised picture, it was not necessary to display the protofilaments that form from αβ/αβ-tubulin as previous works clearly show that more favorable longitudinal αβ/αβ-tubulin interaction results in assembly of tubulin protofilaments (Portran et al., 2017).

5) Figure 2H: First, the simulation (i) assumes that gTuRC is perfectly ring-shaped (but it is not, see point 2 above) and ii) does not take into account binding affinity with tubulin dimers already bound to the template. After binding of the first dimer, binding of a second dimer at the neighbouring position may be promoted by lateral affinity. Also, I don't understand how the probability in calculated – should it not increase further after 4 sites are occupied? At this point it should be much more likely to have lateral contacts than not.

We thank the reviewer for this question. In the common response above, we outline the assumptions and results from the explicit Monte Carlo simulation model for MT nucleation from γ-TuRC (Figure 5 and related supplements). Our revised model incorporates all of the suggestions made by the reviewer. First, we assume an open conformation where lateral αβ-tubulin contacts are not promoted, as shown by the recent structural work. Second, in our stochastic simulations, we allow all possibilities of αβ-tubulin associations including those with γ-TuRC as well other assembled tubulin dimers at various sites. Third, tubulin dimers that arrive have either no lateral neighbors, half a neighbor at the seam, one or two lateral neighbors. The number of neighbors that contribute to the total lateral bond energy affect the dissociation rate of each dimer, as modeled previously for MT polymerization (VanBuren et al., 2002) and shown experimentally (Mickolajczyk et al., 2019) Finally, in Figure 5D, we characterize the arrangement of αβ-tubulin dimers at the transition state for each simulation where γ-TuRC nucleation occurs. On average 5.2**±**1 total αβtubulin dimers were present (n=2119 simulations). Here, many arrangements of αβ-tubulin dimers on γTuRC result in the same the total number of lateral αβ-tubulin contacts formed at the transition state, which is the important feature of our model (Figure 5D, right). Indeed, a few simulations showed a second αβ-tubulin dimer bound to one αβ-tubulin that was already assembled on γ-TuRC at the transition state. While many αβ-tubulin arrangements can occur in the transition state, the ones that exist most frequently have one layer of αβ-tubulins on γ-TuRC and are diagrammed in Figure 5D, right.

Notably these comprehensive simulations go beyond the previous model where we simply counted the number of sites occupied, and αβ-tubulin’s dissociation rate or local interactions from the neighbors were not incorporated in detail. In sum, our Monte Carlo simulations address the reviewer’s comments and comprehensively describe the dynamics of MT nucleation from γ-TuRC in detail.

6) Considering the technical challenges with experimental repetitions and use of different, freshly prepared gTuRC preps for each repetition, it would be useful to repeat the nucleation assay to measure spontaneous and nucleation from seeds and from gTuRC side-by-side to properly compare the different conditions.

We thank the reviewer for this comment and address it as follows. First, as suggested by the reviewer, we performed side-by-side comparison of γ-TuRC and spontaneous MT nucleation in Figure 3—figure supplement 1BC at 10.5μM αβ-tubulin. In Figure 3—figure supplement 1C, we display results from one repetition and supply the other one in Extra Figure 4b and in the Source data file. As described in the manuscript, γ-TuRC-mediated nucleation occurs efficiently at 10.5μM αβ-tubulin, but spontaneous MT nucleation does not. Because each experiment shown here is challenging and time intensive, it is not possible to side-by-side repeat all concentrations reported for γ-TuRC and spontaneous nucleation.

Second, due to the very nature of these in vitro experiments, they are highly reproducible from one experimental replicate to the next. In this work, we used the same batch of αβ-tubulin for comparison across experiments, buffer reagents prepared identically and state-of-the-art coverslip preparation which ensured high reproducibility for spontaneous nucleation and assembly from seeds. In particular for spontaneous MT nucleation, we have displayed the absolute number of spontaneous MTs nucleated in three independent repetitions in Figure 3B, inset. The number of MTs nucleated spontaneously with specified αβ-tubulin concentration is very similar from one replicate to the next. Similarly, reproducibility was observed acorss all three repetitions of MT assembly from seeds, where 2.45μM tubulin was sufficient for seeds to assemble MTs in each replicate. Further, the absolute rate of MT assembly was similar from one replicate to the next. Thus, in Figure 3E, we have displayed data from all three replicates without any normalization.

Notably, for γ-TuRC-mediated nucleation experiments, the variable concentration of purified γTuRC introduces some variability between individual repeats. Thus, for each replicate, that the control experiment with γ-TuRC alone at a specific concentration results in 2-4-fold variation in the number of MTs nucleation. Yet, the trends reported in the manuscript do not change from one-replicate to the next. To exemplify this, in Author response image 2, we present all three replicates for the data corresponding to Figure 2AC. Because the concentration of purified γ-TuRCs varies roughly 2-4-fold between replicates (data sets), the number of MTs nucleated (y-axis) between individual repeats varies but data within individual experimental set can be compared where we used the same γ-TuRC purification. Despite this variation from one replicate to the next, the trends reported in the manuscript remain invariant across replicates. Minimal nucleation was seen at 3.5μM and 7μM tubulin respectively and increasing level of nucleation observed from 10.5-21μM. Therefore, to pool data from multiple replicates, we set a normalization scale (Figure 2D), which we describe in the Materials and methods section and figure legends. E.g. in subsection “Power-law analysis of critical nucleus size on γ-TuRC”, we state, “As the total number of γ-TuRC molecules obtained from different days purifications changes, the rate of nucleation from γ-TuRCs at 10.5μM tubulin was set to 1 (normalization factor) to allow pooling of all datasets for γ-TuRC-mediated nucleation.”

Finally, the power-law analyses used here is insensitive to the γ-TuRC concentration as it measures the fold-change in MT nucleation for each unit change in tubulin concentration. Hence, this measurement was used to compare across different experiments with γ-TuRC, seeds and spontaneous nucleation.

It should be discussed that the minimal concentration for nucleation from seeds in a previous study by Wieczorek et al., (2015) was quite different and similar to nucleation from gTuRC in the present study.

We appreciate this comment from the reviewer that echoes reviewer 1’s comment. Note that there are specific differences between the assay conditions used by Wiezcorek et al., 2015 and this work including different preparation of coverslips and assay conditions. Our coverslip preparation relies on covalently attaching PEG molecules to the glass, which is superior than the non-specific passivation of coverslips used in other works including Wiezcorek et al., 2015 as well as by us previously to assay MT growth (Thawani et al., 2018). As we were initially surprised by the differences of our findings compared to published work, we repeated these experiments with the same coverslip preparation and assay condition as described by Wiezcorek et al. (Author response image 1). The resulting assay yields similar results as Wiezcorek et al. where most MT seeds do not assemble polymerizing MT at 5μM tubulin, but at 7μM tubulin, MTs assemble from seeds (Author response image 1). In sum, with our in vitro assay conditions where we compare nucleation from seeds and from γ-TuRC side-by-side, all seeds assemble MTs at 5μM tubulin. The differences with the published work stem from the specific assay condition.

To examine which of these two in vitro conditions better represents the endogenous conditions, we subjected the blunt seeds to *Xenopus* egg extracts instead of purified tubulin by adapting the assay preparation from Wiezcorek et al. We visualized MT assembly from seeds by spiking fluorescent tubulin in egg extracts. Surprisingly, MTs assembled from seeds immediately. We note that this result is in contrast with γ-TuRC-mediated nucleation in *Xenopus* egg extracts, where that a large number of γ-TuRCs are present in the cytoplasm, yet MT nucleation from these molecules is not observed without MTOCs or branching effector RanGTP (Petry et al., 2013; Thawani et al., 2018; Alfaro-Aco et al., 2017). This comparison corroborates our in vitro results and assay conditions.

Finally, as we aimed to compare their kinetics with γ-TuRC-mediated nucleation side-by-side in our work, we used the same assay condition and coverslip preparation that we used to visualize nucleation from single γ-TuRC molecules. Under these conditions, MT seeds require low tubulin concentration to nucleate. To clarify this in the manuscript, in subsection “Contribution of end architecture of γ-TuRC to microtubule nucleation”, we state, “Notably, the distinct minimal tubulin concentration needed for seeds to assemble a MT as compared to a previous report^24^ results from the differences in assay conditions.”

7) It is not very clear how many times experiments were repeated and what the exact outcomes of each were. Sometimes data from different repetitions was "pooled", sometimes only one "representative dataset" of several repetitions is shown. Would it not be more appropriate to plot mean values from the multiple repetitions? The descriptions in all three figure legends are quite vague, for example Figure 2: "…were repeated at least twice with multiple supporting results…" – what exactly does this mean?The same is true for the transparent reporting form:.…"confirmation and supporting experiments were performed with slightly different conditions and complete agreement was found amongst all measurements. The datasets were sufficient to give confidence on the measured value of interest…". What are the slightly different conditions? How are "complete agreement" and "confidence" determined?Source data is not provided because "all data generated in this study was pooled (no data was left out)", but if only one "representative dataset" is shown, my understanding is that some data was in fact left out.

We appreciate this point raised by the reviewer and have done the following to thoroughly address it. First, performed and included additional experimental repeats and analyses as requested by reviewer 1. Between 3-5 repeats were performed for most MT nucleation experiments. We have also now provided detailed information on the number of replicated performed for every experiment provided in individual figure legend. Notably with this additional data pooled, our power-law analyses results in cooperativity of 3.9 ± 0.5 tubulin dimers for γ-TuRC-mediated nucleation, 8.1 ± 0.9 for spontaneous nucleation, 1 ± 0.3 for assembly from blunt MT end, and 3.3 ± 0.8 for XMAP215/γ-TuRC-mediated co-nucleation.

Second, as described in the response to point (6) above, each replicate was performed with an independent γ-TuRC purification. As a result, the variation in the absolute number of MTs nucleated between replicates mostly represents variation in concentration of γ-TuRCs used. Thus, reporting the mean or median value did not appropriately represent the experimental data. Instead, to present data from all the replicates as suggested by the reviewer, we developed alternative ways of combining data using the following two approaches. For data where power-law analysis of nucleation versus concentration was performed (Figure 2C-D, Figure 3B, Figure 3D-E, Figure 7D-E), we reported a representative curve the nucleation kinetics to display the raw data and combined data from all repeats in the power-law analysis (Figure 2D, Figure 3B, 3E, Figure 7E). This was reported in the figure legend precisely and raw time-series data from the other replicates not displayed in Figure 2C, Figure 3D, Figure 7D is provided in the Source data file. While replicates from seed-templated and spontaneous nucleation can simply be combined as is, to combine data in the power-law analysis from multiple γ-TuRC experiments, it was necessary to set a normalization factor that accounts for variance in γ-TuRC concentration. We specify this normalization factor in individual figure legend. For example, in the legend for Figure 2B, we state, “The rate of nucleation at 10.5μM was set to 1 to normalize differences in γ-TuRC concentration from individual experiments.” For experiments where we study the effect of MAPs (Figure 6B, 6D, Figure 7B, 7F), we combined results from multiple experiments, as requested by the reviewer, by setting a normalization factor between different experimental repeats and displaying kinetics from all repeats on the plot. For example, in the legend for Figure 6D, we state, “Number of MTs nucleated in control reactions at 150 seconds was set to 1 to account for variable γ-TuRC concentration across purifications, all data was pooled and reported.”

Third, as we performed and included further replicates, it was not necessary to refer to supporting data in most places. In few cases where we felt the need to mention the supportive data in the revision manuscript, we specified the exact differences for those additional experiments. The differences were in concentration points used. To exemplify this for Figure 4B-C’s legend, we state, “(B) Single molecule microscopy was performed with γ-tubulin and αβ-tubulin. Control buffer (left panels, (i) and (ii)) or biotinylated αβ-tubulin (right panels, (i) and (ii)) was attached to coverslips, incubated with fluorescent αβ-tubulin (i) or γ-tubulin (ii) molecules, set as 0 seconds, and their binding at 60-90 seconds. Number of bound molecules were analyzed for the first 15 seconds of observation described in Methods. Experiments and analyses in (B-C) were repeated identically two times, pooled and reported. n=56 data points each were displayed as mean **±** std in the bar graph in (C). Further confirmed with a third supporting experimental set where the observation began later at 180 seconds and was therefore, not pooled.”

Fourth, we also report 95% confidence intervals on the number of MTs nucleated to quantitatively represent the error in our measurements. This is displayed with shaded regions for each measured curve. The confidence interval was calculated to represents the heterogeneity between various fields-of-view in a reaction and is detailed in Methods and individual figure legends.

Finally, we have provided source data for Figure 2B, 2C, 2D, Figure 3—figure supplement 1C, Figure 3—figure supplement 1D, Figure 3B, 3D, 3E, Figure 4C, Figure 6B, 6D, Figure 7B, 7D, 7E, 7F, Figure 7—figure supplement 1B, Figure 7—figure supplement 1E. We have also provided detailed information on sample size in the transparency reporting form, and thoroughly inspected for any other instance where information was left out.

8) What is the percentage of gTuRCs that nucleate microtubules in the TIRF assay? This number should be provided.

To address this question, we used our labelled γ-TuRC assays and either directly measured the number of MTs that nucleated from labelled γ-TuRCs in the presence of highest tubulin we tested at 21μM tubulin and additional 100nM XMAP215, or directly measured the number γ-TuRCs that nucleated with lower 10.5μM and further back-calculated the nucleation efficiency at the highest tubulin concentration tested (21μM) using previously measured curves. We find that approximately 10-15% of γ-TuRCs nucleate MTs. Notably this number is likely to be an underestimate as few contaminant molecules in the purification could also be labelled with Alexa-568 dye during purification or some γ-TuRCs could exist as incomplete rings as shown recently (Liu et al., 2020). We have now reported this in the text in subsection “Contribution of end architecture of γ-TuRC to microtubule nucleation”, “At the highest tubulin concentrations, approximately 10-15% of γ-TuRCs nucleate MTs in the TIRF assays”, and how this calculation was made is detailed in the Materials and methods section.

[Editors’ note: what follows is the authors’ response to the second round of review.]

Summary:The assembly and organization of the microtubule cytoskeleton requires nucleation of microtubules by the nucleator γ-TuRC, but the underlying mechanism has remained obscure. Recent Cryo-EM structures of γ-TuRC have raised additional questions by showing that native γ-TuRC is in a conformation that is not well suited to template microtubule nucleation. This study provides first insight by establishing TIRF imaging from purified, immobilized γ-TuRCs in vitro. Using a combination of biochemistry and modeling the study explains how an "imperfect" nucleation template can function as nucleation platform and identifies several crucial determinants of this activity. Rather than activation of γ-TuRC by a conformational switch prior to nucleation, the authors propose that the conformational switch occurs as a result of nucleation.The authors have provided very thorough revisions, presenting and including a substantial amount of new data. The manuscript has been significantly improved and despite the lack of structural support data by EM in the manuscript, the main conclusions are well supported and we consider it now suitable for publication in eLife. There are, however, a few remaining issues that we would like the authors to address.Essential revisions:1) We appreciate that you have modified the title, but it now seems to not make complete sense. We would suggest "The transition state and regulation of γ-TuRC-mediated microtubule nucleation revealed by single molecule microscopy".

That sounds good and we have now edited the manuscript’s title as requested.

2) "Notably, the distinct minimal tubulin concentration needed for seeds to assemble a MT as compared to a previous report 24 results from the differences in assay conditions." The authors should be clearer here, explaining in more detail (as they did in the response to reviewers) that they got the same result as previous authors when they used their particular conditions.

We agree with this point and clarified this point further in the manuscript. In subsection “Contribution of end architecture of γ-TuRC to microtubule nucleation”, we state, “Notably, when this experiment was replicated with the coverslip preparation and assay conditions reported previously^25^, a high concentration of tubulin was necessary for seeds to assemble MTs in agreement with the previous work^25^. However, our assay conditions, that were used to compare seed-templated MT assembly with γ-TuRC-mediated nucleation side-by-side, result in a low, minimal tubulin concentration that is needed for seed-mediated MT assembly.”

3) "In summary, because γ-TuRC positions an array of γ-tubulins at its nucleation interface that are thought to stabilize intrinsically weak, lateral αβ/αβ-tubulin interaction 9,10,13,14,17,20,22147, MT nucleation by γ-TuRC has been proposed to function similar to polymerization of a MT end. Here we show several lines of evidence that γ-TuRC-mediated nucleation has distinct characteristics from MT polymerization and assembly from blunt MT seeds. While growth speed of MTs nucleated from γ-TuRC or templated from MT seeds is similar (Figure 3—figure supplement 1D), γ-TuRC molecules do not nucleate MTs at low tubulin concentration where MT polymerization can occur." It is important to point out here that this relates specifically to the purified γTuRCs that nucleate microtubules in the absence of other cellular factors. While the authors do show that some of the proposed activators do not activate the γ-TuRC in their assays, they cannot yet rule out that other proteins do function in vivo to promote the correct configuration of the γTuRC. They mention this in the Discussion section, but they are making the statement in the Results section, and it is a bit misleading to group all γTuRCs into this poor-nucleation bracket.

We appreciate this point by the reviewers. In subsection “Contribution of end architecture of γ-TuRC to microtubule nucleation”, we appended the specified text with, “While these results were obtained with endogenous γ-TuRCs purified from cytosol, it remains possible that specific factors at MTOCs can modulate γ-TuRC’s conformation and kinetics.”

4) Figure 6B, D – the authors state that there is no significant difference in nucleation rate when they add γTuNA or TPX2, but the graphs do show differences and no statistical test is provided.

We thank the reviewers for raising this important point. We used the 95% confidence interval to indicate the high degree of similarity. At the same time, we agree that our data can be represented in another way that quantitatively reports the small effect of TPX2 and γ-TuNA. In accordance with how the effect of activation factors was reported in previous and recent works (Consolati et al., 2020; Liu et al., 2020; Choi et al., 2010; Kollman et al., 2010 and others), we have now presented this data as fold change. Specifically, CDK5RAP2’s γ-TuNA motif γ-TuRC-mediated nucleation by 1.4(**±** 0.02) -fold at *t* = 180 seconds (mean ± std for *n* = 2 biological replicates). TPX2 increases the nucleation activity of γTuRC by 1.2 (**±** 0.3) -fold (mean ± std for *n* = 3 biological replicates) at *t* = 180 seconds. We have also reported the effect of XMAP215 in a similar fashion in text in subsection “XMAP215 promotes microtubule nucleation by strengthening the longitudinal bond energy between γ-TuRC and αβ-tubulin” for the corresponding Figure 7B. Further, we have edited the sub-headings of Figure 6B and Figure 6D to reflect these modifications.

5) "Recently, XMAP215 was discovered to be a nucleation factor that synergizes with γ-TuRC". There is a new pre-print from Trisha Davis' group showing that nucleation by γTuRC and XMAP215 is additive, not synergistic http://biorxiv.org/content/early/2020/05/23/2020.05.21.109561. It would be useful for the reader if the authors would discuss how this compares to their work here.

We appreciate this comment by the reviewers. We will summarize the recent work from Davis lab below and then explain how we will incorporate it.

As detailed below, we have several concerns regarding the pre-print on bioRxiv including that it is incomplete. First, in King et al., 2020, the data for supplementary figures has not been provided that is necessary to understand how “additive” and “synergistic” effects of XMAP215 with γ-tubulin filaments were calculated with a theoretical model. Little detail for this is provided in the Materials and methods section. As a result, it is unclear to us how the additive effect of XMAP215 was established, an understanding which is necessary to comment on this finding in our manuscript. Second, and most importantly, we note that the new preprint used turbidity measurement to assay MT assembly, which is not a direct measurement of MT nucleation. Turbidity combines the measurement of number of MTs (nucleation) with the total length of MTs. The latter is an important effect that cannot be ignored with polymerases like XMAP215, which increase MT length too. Instead, a direct nucleation assay is needed to clearly measure XMAP215’s effect, and distinguish between synergistic or additive scenarios. Finally, from the limited information provided in King et al., 2020, we gather that the turbidity curves reported are a normalized (not absolute) measurement of the total MT mass assembled in the individual reaction. The 10% assembly time from normalized curves was compared or likely summed across conditions to assign a role to XMAP215. We believe that instead of this analysis, quantitatively comparing the total MT mass (prior to normalization) will be necessary to assess if XMAP215 has an additive or synergistic effect.

Indeed, work from two other labs alongside our previous work have established a synergy between XMAP215 homologs with γ-tubulin complexes using varied model systems (Gunzelmann et al., 2018; Consolati et al., 2020; Thawani et al., 2018). In all three works, when measured with an assay where number of MTs were directly counted, the two molecules alone are inefficient in MT nucleation but have a synergistic effect. Finally, our current work alongside Thomas Surrey’s recent work (Consolati et al., 2020) further supports a synergistic effect with single molecule assays.

Nevertheless, we cite and give credit the work on bioRxiv and incorporate the reviewers’ suggestion as best as possible. In Subsection “Role of putative activation factors in γ-TuRC mediated nucleation”, we changed the introductory sentence to the following one, “ Recently, XMAP215 was discovered to be a nucleation factor that synergizes with γ-TuRC in *X. laevis* and *S. cerevisiae*^29,50^, or works in an additive manner with γ-tubulin^51^.” Then, we resolve this paradox at the end of the XMAP215 section on page16-17 with “Altogether, our results confirm that XMAP215 indeed functions synergistically with γ-TuRC, in agreement with recent works^15,29,50^. Most importantly, our results show that, while the transition state is defined by γ-TuRC’s conformation, XMAP215 strengthens the longitudinal γ-/αβ-tubulin bond to function as a bona-fide nucleation factor.”

6) The authors should discuss in more detail recently published work by the Surrey group (2020), which also shows, as the current manuscript, TIRF imaging of nucleation from immobilized γ-TuRCs. There are some differences e.g. regarding the critical number of dimers that need to assemble on gTuRC and regarding overall gTuRC activity in the assay. It would be very useful for the reader to know how the observations/conclusions in the two studies agree or differ.

We have expanded on our Discussion of this recently published work with the sentences:

“Notably, a parallel work also reported MT nucleation from single, human γ-TuRC molecules recently^15^. While the majority of findings agree with our work, 6.7 dimers were required in the critical nucleus and an overall lower activity of γ-TuRC (0.5%) was found^15^. Low structural integrity of purified γ-TuRC from incorporation of BFP-tagged GCP2 and a higher ratio of γ-tubulin sub-complexes, or species-specific variation in γ-TuRC properties could explain these differences.”

In the revised manuscript file, we have also made changes to the bibliography to reflect the final, published versions of the cryo-EM works.